# Chronic social defeat stress induces meningeal neutrophilia via type I interferon signaling in male mice

Stacey L. Kigar [1,2,3,8] ✉, Mary-Ellen Lynall [3,4,8], Allison E. DePuyt[1], Robert Atkinson[1], Virginia H. Sun [1], Joshua D. Samuels[1], Nicole E. Eassa[1], Chelsie N. Poffenberger[1], Michael L. Lehmann[1], Samuel J. Listwak[1], Ferenc Livak[5], Abdel G. Elkahloun[6], Menna R. Clatworthy [4,7], Edward T. Bullmore[3] & Miles Herkenham[1]

Inflammation is increasingly recognized as a risk factor for psychiatric disorders. Animal models of stress and stress-related disorders are associated with blood neutrophilia. The mechanistic relevance of this to symptoms or behavior is unclear. We characterized the immune response to chronic social defeat (CSD) stress at brain border regions in male mice. Here we show that chronic, but not acute, stress causes neutrophil accumulation in the meninges —i.e., "meningeal neutrophilia"— but not the brain. CSD promotes neutrophil trafficking to meninges via vascular channels originating from skull bone marrow (BM). Transcriptional analysis suggests CSD increases type I interferon (IFN-I) signaling in meningeal neutrophils. Blocking this pathway via the IFN-I receptor (IFNAR) protects against the negative behavioral effects of CSD stress. Our identification of IFN-I signaling as a putative mediator of meningeal neutrophil recruitment may facilitlate development of new therapies for stress-related disorders.

In humans, chronic inflammation has been linked to both psychosocial stress[1] and major depressive disorder (MDD)[2], highlighting a possible role for the immune system as an intermediary between psychological risk factors and the development of mood disorders. Proinflammatory cytokines produced by innate immune cells have attracted attention, given strong evidence that they are associated with MDD[3] and depressive-like behavior in animal models[4]. In particular, proinflammatory type I interferons (IFN-I) are associated with development of MDD[5,6] and induce depressive symptoms in otherwise non-depressed people[7] and quasi-depressive behaviors in laboratory animals[8,9].

In parallel, multiple clinical studies have documented an increased ratio of neutrophils relative to lymphocytes in people with MDD[10,11]. Neutrophils are IFN-I-sensing phagocytes of the innate immune system that play a critical role as sentinels, acting as first responders to both pathogens and sterile injury[12,13]. Blood neutrophil levels rapidly elevate following acute stress via direct sympathetic innervation of bone marrow (BM)[14,15], and stress is strongly associated with the development of depression[16]. We have furthermore demonstrated in MDD patients that peripheral neutrophil levels are the major immune cell subset most predictive of symptom severity[17], suggesting a potential role for neutrophils in the pathogenesis of depression.

[1]National Institute of Mental Health, Bethesda, MD, USA. [2]Department of Medicine, University of Cambridge, Cambridge, UK. [3]Department of Psychiatry, University of Cambridge, Cambridge, UK. [4]Molecular Immunity Unit, University of Cambridge Department of Medicine, Cambridge, UK. [5]Laboratory of Genome Integrity, Flow Cytometry Core, National Cancer Institute, Bethesda, MD, USA. [6]Microarrays and Single-Cell Genomics, National Human Genome Research Institute, Bethesda, MD, USA. [7]Cellular Genetics, Wellcome Sanger Institute, Cambridge, UK. [8]These authors contributed equally: Stacey L. Kigar, Mary-Ellen Lynall. ✉e-mail: sk2128@cam.ac.uk

Mechanisms by which neutrophils contribute to depressive symptoms are currently unclear; however, we and others have demonstrated in mice that cells acting from within the meninges play a key role in the maintenance of behavioral homeostasis[18–20]. Specifically, we have shown that transgenic *Cd19*[−/−] mice, which are B cell deficient, are more anxious than wild type (WT) littermates at baseline[20]. Subjecting WT mice to chronic social defeat (CSD) stress–which reliably induces depressive and anxiety-like behaviors[21–24]–led to decreased B cell numbers in the meninges. Bulk transcriptomic analysis of meningeal tissue from CSD-stressed and *Cd19*[−/−] mice revealed a shared increase in IFN-I signaling. Moreover, *Cd19*[−/−] mice showed elevated levels of meningeal neutrophils[20].

Traditionally, neutrophils are thought to enter tissue from the bloodstream following BM egress. However, a brain-specific mechanism wherein neutrophils traffic directly via skull BM reservoirs to the meninges has recently been characterized[25]. These skull-derived neutrophils are preferentially recruited to the brain following tissue damage and in disease[26–28] but their role in stress-related processes is unknown.

We hypothesized that neutrophils within the meningeal space were related to stress-induced depressive symptoms, and that these neutrophils originate from skull BM. We characterized immune cell dynamics and phenotypic changes in the meninges and in a variety of peripheral tissues following CSD stress. Using a data-driven approach, we identified CSD meningeal neutrophils as the source of enhanced IFN-I signaling previously seen in CSD[20]. This led us to hypothesize that IFN-I signaling promotes neutrophil egress from skull BM to the meninges, and in turn, depressive- and anxious-like behaviors following CSD stress. Our preliminary evidence suggests that systemic IFN-I depletion normalizes the effects of CSD stress on meningeal neutrophils and behavior. This suggests that skull-to-meningeal trafficking of IFN-I sensing neutrophils is an important factor in the progression of CSD stress-induced behavioral changes.

## Results

### CSD causes expected depressive- and anxiety-like behavioral changes

We first confirmed that our well-established CSD paradigm[21–24] elicits expected depressive- and anxiety-like behavior in C57BL/6 J WT mice (Fig. 1a, S1A). Male CSD-treated mice exhibited sexual and social anhedonia in the urine scent marking (USM; Fig. 1b)[23] and social interaction (SI; Fig. 1c) tests, respectively. CSD stress also induced anxiety-like behavior in both the open field (OF; Fig. 1d) and the light/dark (LD; Fig. 1e) tests.

Approximately 30% of the CSD-treated mice were "resilient" to stress, as assessed using an SI quotient $\geq 2$ (time socially investigating/time on non-social investigation; Fig. S1B), comparable to other, similar social defeat stress models[29].

### CSD increases meningeal and blood neutrophil abundance

Pre-sacrifice intravenous anti-CD45 antibody injections (CD45iv) were used to allow post-mortem distinction of non-vascular (iv−) and blood-exposed (iv+) meningeal cells. Flow cytometric analysis of meningeal tissue and blood showed that CSD increases neutrophil abundance (Fig. 1f, g; see Fig. S2A, B for gating strategy). Specifically, there was a 1.3-fold increase in the relative percentage of iv− meningeal neutrophils, consistent with absolute count data (Fig. 1f). There was also a 1.7-fold increase in the relative percentage of iv+ meningeal neutrophils (Fig. 1g). In blood, there was a large, 5.6-fold increase in the proportion of neutrophils, consistent with absolute count data (Fig. 1h).

We also noted a CSD-related increase in meningeal monocytes and, as previously reported[20], a decrease in meningeal B cells (Fig. S3A, B). For sensitivity analyses, we regrouped mice into susceptible (SI < 2) and resilient (SI ≥ 2)[29]; this did not alter the main effects

of CSD on immune cell populations (Fig. S4A, B). We considered the potential impact of fight-related wounding on immune cell dynamics (Fig. S5A–F). Of all cell types and tissues examined, only iv− meningeal monocytes showed a significant relationship with wounding (Fig. S5A). Duration of time in our colony also did not impact blood or meningeal neutrophil levels (Fig. S5G).

### Increased neutrophil levels correlate with stress-related behavioral changes

Increased meningeal and blood neutrophils were associated with increased anhedonic- and anxiety-like behaviors (Fig. 2). For anhedonia, there were significant associations between reduced USM marking and increased neutrophil levels (Table S1) and a trending relationship between elevated iv+ meningeal neutrophils and fewer social approaches (Table S2). We also noted an increase in iv− meningeal neutrophils in HC mice that did not mark in the USM test (Fig. S5H).

For anxiety, there were significant associations between reduced exploration of a novel arena and higher neutrophil levels (Table S3) and trends between meningeal neutrophil elevation and reduced crosses to light in the light/dark box (Table S4).

### Chronic but not acute stress induces meningeal neutrophil accumulation

In both blood and meningeal tissue, we observed dynamic changes in neutrophil and other immune cell populations with increasing exposure to social defeat (Figs. 3a, S6A, S7A). No stress-induced elevation in meningeal neutrophils was observed until the day 14 time point (Table S5). In contrast, a single day of defeat stress was sufficient to elevate neutrophils in blood, and there was a significant increase in blood neutrophils at each time point overall when compared to HC (Table S6).

### Tissue-specific recovery dynamics following cessation of CSD stress

To assess the stability of the neutrophilia phenotype, mice were given varying amounts of time to recover from 14 days of CSD stress. We observed tissue- and immune cell-specific changes over the course of recovery (Figs. 3b, S6B, S7B). Meningeal neutrophil levels remained elevated up to 24 h after the final defeat encounter but decreased to control levels following 7 days of recovery (Table S7). Blood neutrophil levels had returned to control levels more quickly, within 8 h of stress cessation (Table S8).

### Replication of CSD-induced meningeal immune changes

LysM[gfp/+] mice, which strongly express GFP in neutrophils (Fig. 4a, b), recapitulated the core behavioral response to CSD stress, e.g., decreased USM preference and decreased OF crosses to center (Fig. S8A, B). Confocal microscopy of meningeal whole mounts showed broad distribution of LysM-GFP+ cells that appeared morphologically like neutrophils both inside and outside blood vessels (Fig. 4C, D). Quantitation revealed a 1.3x-fold increase in the number of LysM-GFP+ meningeal cells in CSD vs control (Fig. 4e) - consistent with the increase in Ly6G+ neutrophils determined by flow cytometry in C57BL/6 J mice (Fig. 1f, g).

There was a main effect of CSD on LysM-GFP+ cell counts across intravascular, abluminal (≤10μm away from a blood vessel) and non-vascular (>10μm away from a blood vessel) meningeal tissue compartments, but no significant post hoc effects (Fig. 4f). Like meningeal neutrophils in C57BL/6 J mice, LysM-GFP+ cell levels had not decreased within 16 h of recovery from CSD stress (Fig. S8C).

LysM[gfp/+] mice were otherwise similar to C57BL/6 J mice: flow cytometric analysis showed blood neutrophilia and decreased blood B cells with recovery to baseline 16 h post-CSD (Fig. S9A, B). Immunohistochemical (IHC) examination of brain tissue showed no evidence

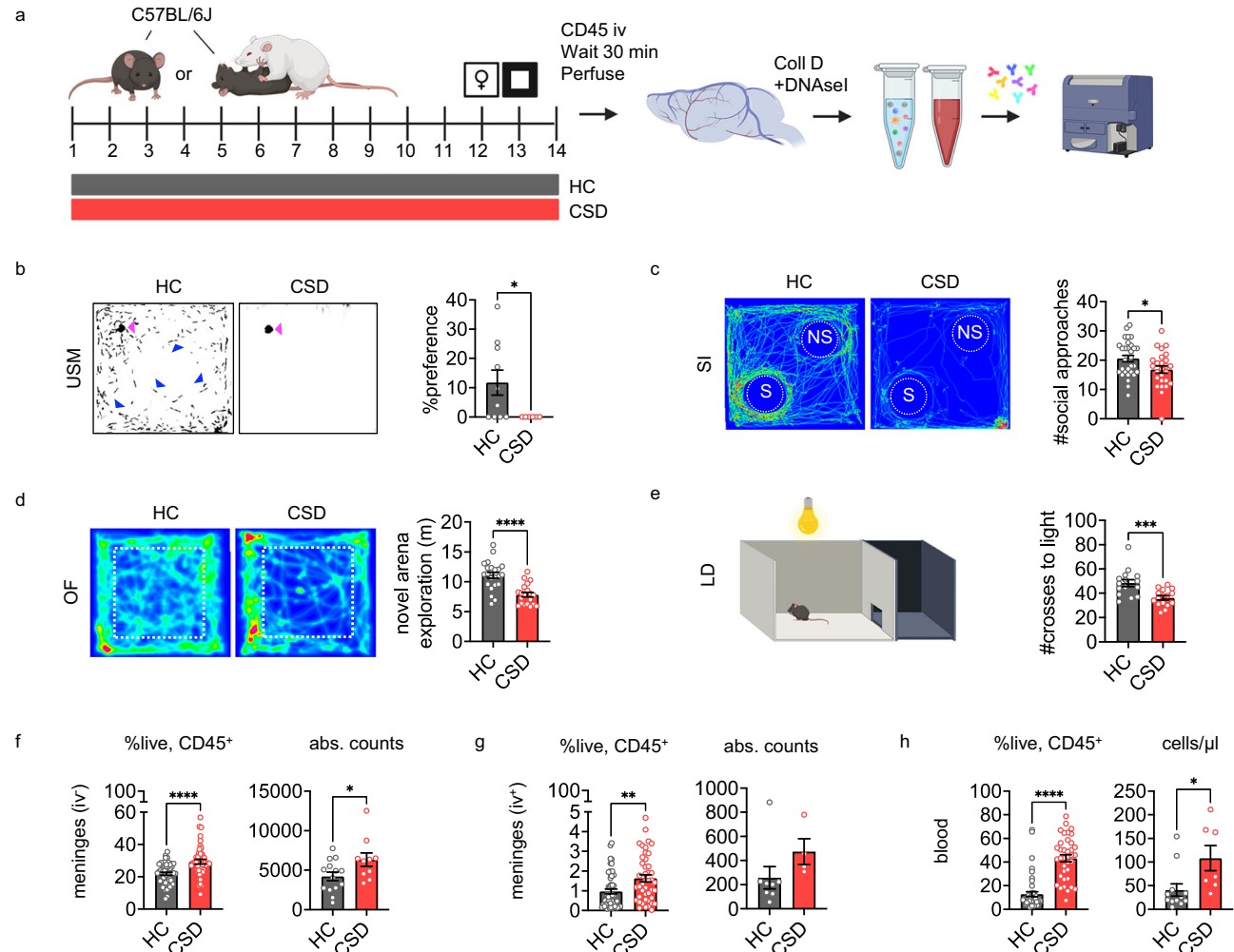

**Fig. 1 | Meningeal neutrophils are elevated following chronic social defeat (CSD) stress, which causes depressive-like behavioral changes. a** CSD mice were behaviorally tested on days 10-13, and tissue harvested at day 14. A fluorescently labeled CD45 antibody was retro-orbitally injected to discern blood-exposed meningeal cells. CSD causes expected sexual (**b**; ****$p$ < 0.0001, $U$ = 13, Hodges-Lehmann estimate = −24.5, $n_{HC}$ = 16, $n_{CSD}$ = 13. N = 5 experiments.) and social (**c**; *$p$ = 0.011, $U$ = 323, Hodges-Lehmann estimate = −4.0, $n_{HC}$ = 34, $n_{CSD}$ = 30. N = 10 experiments) anhedonia in USM and SI tests, respectively. In (**b**), pink arrowheads indicate female stimulus urine, blue arrowheads indicate representative urine scent marks from male test mice. Heatmaps (**c, d**) indicate animal tracking; red color indicates greater time spent in that location. CSD causes increased anxiety-like behavior in the OF (**d**; ****$p$ < 0.0001, $U$ = 65, Hodges-Lehmann estimate = −3.4, $n_{HC}$ = 23, $n_{CSD}$ = 21. N = 8 experiments) and LD box (**e**; ***$p$ = 0.0007, $U$ = 38, Hodges-Lehmann estimate = −11.0, $n_{HC}$ = 15, $n_{CSD}$ = 16. N = 7 experiments) tests. Flow cytometric analysis shows CSD causes an increase in: (**f**) iv⁻ meningeal neutrophils (CD45iv⁻; CD11b⁺; Ly6G⁺; Ly6Cᶦⁿᵗ), both as a percent of live CD45⁺ cells (****$p$ < 0.0001, $U$ = 573, Hodges-Lehmann estimate = −6.6, $n_{HC}$ = 52, $n_{CSD}$ = 44.

N = 19 experiments), and in absolute cell counts (Unpaired t test, *$p$ = 0.038, $t$ = 2.2, $df$ = 22, 95% CI [128.4, 4107], $n_{HC}$ = 14, $n_{CSD}$ = 10. N = 4 experiments); (**g**) iv⁺ meningeal neutrophils (CD45iv⁺) as a percentage (Unpaired t test, **$p$ = 0.0024, $t$ = 3.1, $df$ = 94, 95% CI [0.24, 1.08], $n_{HC}$ = 52, $n_{CSD}$ = 44), but not absolute counts ($n_{HC}$=8, $n_{CSD}$ = 4. N = 1 experiment); and (**h**) blood neutrophils (CD45iv⁺), in both percentage (****$p$ < 0.0001, $U$ = 146, Hodges-Lehmann estimate=34.2, $n_{HC}$ = 47, $n_{CSD}$ = 37. N = 17 experiments) and absolute counts (Unpaired t test, *$p$ = 0.021, $t$ = 2.6, $df$ = 17, 95% CI [11.6, 122.8], $n_{HC}$ = 12, $n_{CSD}$ = 7. N = 3 experiments). Data points represent individual mice. Two-tailed tests were used in (**b**–**h**); all tests were Mann-Whitney tests unless otherwise indicated. No adjustments for multiple comparisons were made. Additional data shown in Figs. S1–4. Gating strategy shown in Fig. S2. CSD = chronic social defeat stress, HC = home cage, LD = light/dark, OF = open field, SI = social interaction (S = social, NS = non-social), USM = urine scent marking. Data shown as mean ± SEM. *$p$ < 0.05, **$p$ < 0.01, ***$p$ < 0.001, ****$p$ < 0.0001. Source data provided in Source Data file. Schematics from BioRender. Kigar, S. (2025) https://BioRender.com/v98waxl.

of LysM-GFP+ cell infiltration into brain parenchyma, though there were significantly more cells 'stuck' in the neurovasculature of CSD mice (Fig. S10A, B). As in blood, neurovascular 'sticking' of LysM-GFP+ cells had returned to baseline within 16 h of recovery from CSD, though we could not determine the cause of this phenomenon (Fig. S10C–F).

**CSD increases neutrophil trafficking between skull BM and meninges**

We used tissue clearing to visualize LysM-GFP⁺ cells in the channels connecting skull BM to the meninges (Fig. 4g). There were more

LysM-GFP⁺ cells per vascular channel in CSD mice compared to HC (Fig. 4h, i), suggesting this route may be important for myeloid cell trafficking into the meninges following psychosocial stress.

We next tested the extent to which elevated meningeal neutrophil numbers reflect neutrophil levels in peripheral tissues, which were also increased by CSD stress in C57BL/6 J mice (Fig. 5a; Table S9). Pearson correlations and unsupervised hierarchical clustering across tissues revealed that iv⁻ meningeal neutrophil levels were most similar to skull and tibial BM neutrophils and showed no relationship with blood neutrophils (Fig. 5b; Table S10). In contrast, iv⁺ meningeal neutrophils correlated with blood neutrophils.

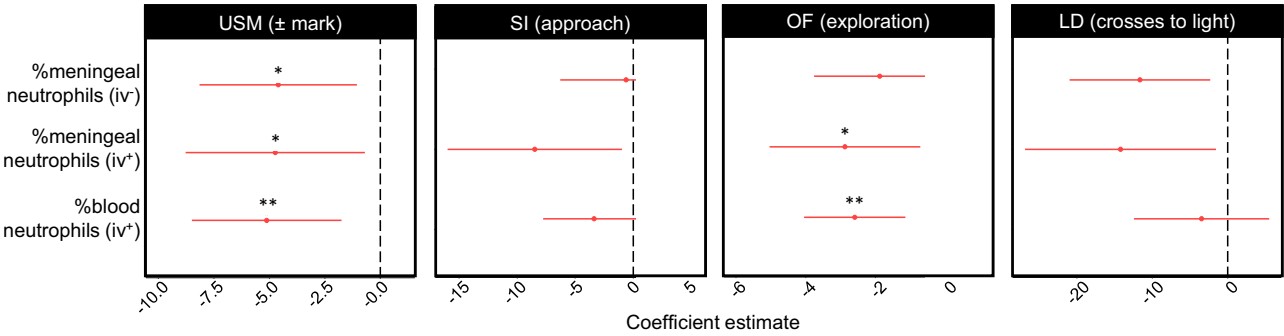

**Fig. 2 | Elevated neutrophil levels are associated with the negative behavioral sequelae of CSD stress.** Plots show standardized effect sizes for neutrophil levels on behavioral outcomes shown in Fig. 1. *Left to right*: USM marking ($n_{HC}$ = 16, $n_{CSD}$ = 17. N = 5 experiments), SI social approaches ($n_{HC}$ = 22, $n_{CSD}$ = 26. N = 10 experiments), OF novel arena exploration ($n_{HC}$ = 21, $n_{CSD}$ = 23. N = 9 experiments), and L/D box crosses to light ($n_{HC}$ = 15, $n_{CSD}$ = 17. N = 7 experiments). Mixed generalized linear regression modeling with maximum likelihood estimation (MLE) revealed negative relationships between USM marking behavior and blood (**FDR = 0.0087, β = -5.12, $z$ = -2.98, 95% CI [-4.47, -1.12]), iv⁻ (*FDR = 0.016, β = -4.60,

$z$ = -2.55, 95% CI [-4.50, -0.85]), and iv⁺ (*FDR = 0.021, β = -4.74, $z$ = -2.30, 95% CI [-4.74, −0.65]) meningeal neutrophils. Blood (**FDR = 0.0053, β = −2.59, $t$ = −3.50, 95% CI [−0.82, −0.24]) and iv⁺ (*FDR = 0.020, β = −2.88, $t$ = −2.61, 95% CI [−1.06, −0.13]) meningeal neutrophils are significantly associated with OF anxiety-like behavior. There were no significant relationships between neutrophils and SI or LD. See Tables S1–S4 for statistics. Unless otherwise indicated, ordinary least squares (OLS) regression was used for modeling. Significance levels reflect FDR-adjusted p values. LD light/dark, OF open field, SI social interaction, USM urine scent marking. *$p < 0.05$, **$p < 0.01$. Source data provided in Source Data file.

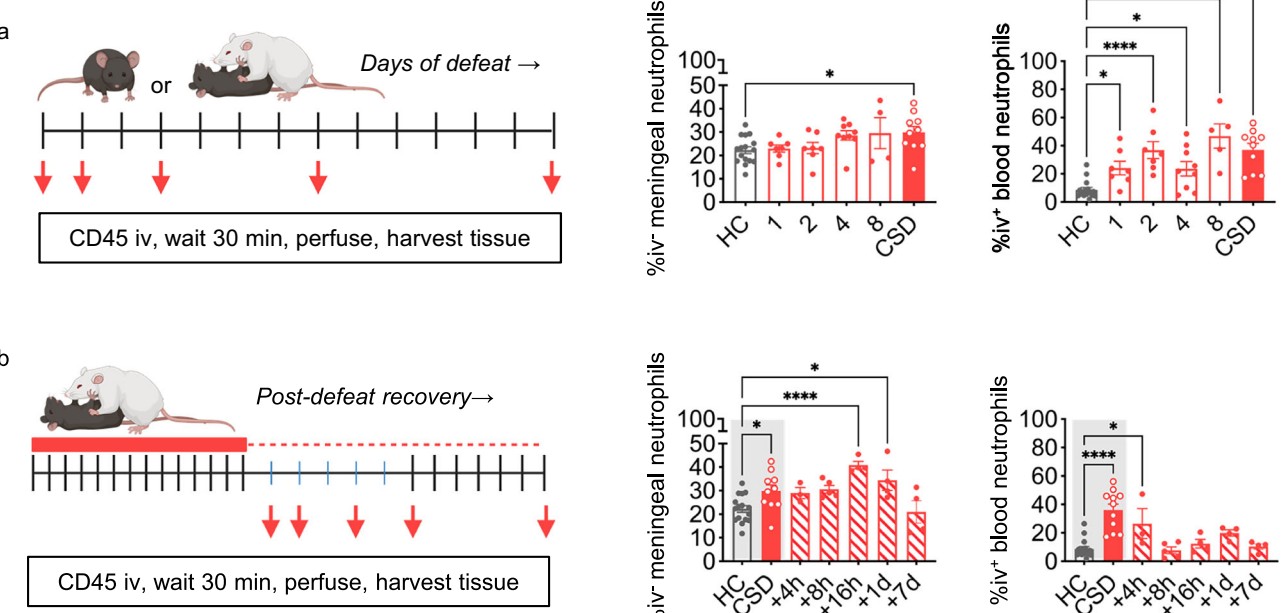

**Fig. 3 | Chronic, but not acute, social defeat stress elevates meningeal neutrophils; prolonged elevation of meningeal neutrophils relative to blood in recovery. a** *Left:* Schematic of acute vs chronic stress study; red arrows indicate time points in days at which mice were killed and tissue was harvested. *Right:* There was a main effect of the number of encounters for meningeal neutrophils (*$P$ = 0.030, $F_{(5,48)}$ = 2.7), but only the CSD day-14 group showed a significant increase by post hoc analysis. In contrast, there was a significant increase in blood neutrophils overall (****$P < 0.0001$, $F_{(5,48)}$ = 10.5) and at each time point when compared to HC (subscript indicates days of defeat: $n_{HC}$ = 8, $n_1$ = 7; $n_2$ = 7; $n_4$ = 9; $n_8$ = 4; $n_{14}$ = 10. N = 2 experiments). Results for other cell types are shown in Fig. S8A. **b** *Left:* Schematic of post-CSD recovery study; red arrows indicate time points in hours (blue ticks) or days (black ticks) at which mice were killed and tissue was harvested. Gray shading indicates shared HC and CSD animals with (**a**), as experiments were done contemporaneously. *Right:* There was a main effect of time post-CSD for meningeal neutrophils (***$P$ = 0.0001, $F_{(6,39)}$ = 6.1); 16 and 24 h post-CSD,

meningeal neutrophils were still significantly elevated relative to HC. In contrast, blood neutrophil levels showed an effect of time (****$P < 0.0001$, $F_{(6,40)}$ = 11.4) and remained significantly elevated for up to 4 h post-CSD, then recovered to HC-like levels thereafter (subscript indicates time post-CSD: $n_{4h}$ = 3, $n_{8h}$ = 5; $n_{16h}$ = 4; $n_{1d}$ = 4; $n_{7d}$ = 4. N = 1 experiment). See Tables S5–S8 for post-hoc statistics. Data points represent individual mice. All statistical tests were run as 1-way ANOVAs. Significance levels shown are from post-hoc testing, with adjustment for multiple comparisons. Results for other cell types are shown in Fig. S8B. All defeated groups are shown in red; open bars indicate fewer defeat encounters than the standard CSD paradigm, shaded bars indicate animals that underwent the complete CSD paradigm and were given time to recover. CSD = chronic social defeat stress, HC = home cage. Data shown as mean ± SEM. *$p < 0.05$, ****$p < 0.0001$. Source data are provided as a Source Data file. Schematics created in BioRender. Kigar, S. (2025) https://BioRender.com/djwvueb.

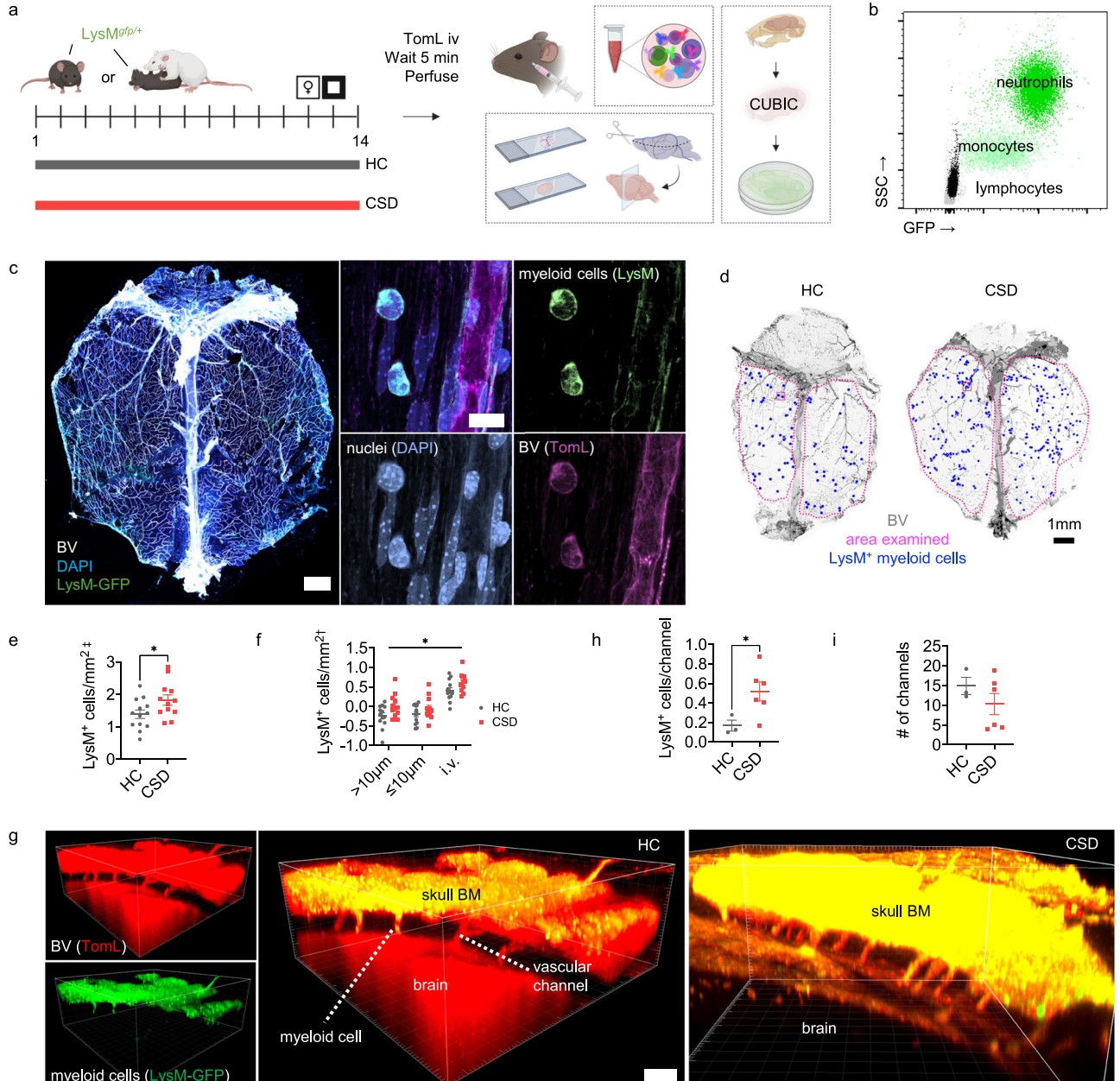

**Fig. 4 | CSD stress leads to increased numbers of LysM+ myeloid cells in vascular channels connecting skull BM to meninges. a** LysM$^{gfp/+}$ mice were subjected to CSD stress and behaviorally phenotyped (Fig. S8). Before TomL intravascular (iv) injection, blood was drawn for flow cytometry (Fig. S9). Dorsal meninges and brain (Fig. S10) were prepared for imaging. A separate cohort of mice was used for tissue clearing. **b** Flow cytometry data from blood shows high GFP expression in neutrophils compared to other cell types. **c** *Left*: Representative image for dorsal wholemount meningeal tissue. Scale bar = 1 mm. *Right*: LysM-GFP+ cells adjacent to a blood vessel; merge and individual channels. Scale bar = 10μm. **d** Representative images from HC and CSD mice showing hand-counted cells, normalized to indicated area. Inset box represents size of area shown in (**c**, *Right*). **e** Quantification of total meningeal LysM-GFP+ cells shows an increase with CSD stress (Unpaired t test, *$p$ = 0.046, $t$ = 2.2, $df$ = 23, 95% CI [0.019, 0.86], $n_{HC}$ = 13, $n_{CSD}$ = 12. N = 2 experiments). ‡natural log-transformed. **f** LysM-GFP+ cells from (**e**) were separated into three categories based on their location in the tissue: >10μm away from a blood vessel ("non-vascular"), ≤10μm away from a blood vessel ("abluminal"), and intravascular ("iv"). CSD led to an overall increase in LysM-GFP+ cells (RM-ANOVA: group, *$p$ = 0.043, $F_{(1,23)}$ = 4.6, $n_{HC}$ = 13, $n_{CSD}$ = 12); asterisk indicates main effect of CSD, but there were no significant post-hoc comparisons. †log$_{10}$ -transformed.
**g** Representative image from cleared skulls showing vascular channels between skull BM and the meninges. Scale bar = 100μm. *Left*: Individual channels. *Middle*: Merged image, HC mouse. *Right*: Merged image, CSD mouse. Quantification of LysM-GFP+ cells per channel (**h**) indicates increased egress from skull BM following CSD stress (Unpaired t test, *$p$ = 0.049, $t$ = 2.4, $df$ = 7, 95% CI [0.0028, 0.70]), and (**i**) no differences in the number of channels counted between groups ($n_{HC}$ = 3, $n_{CSD}$ = 6. N = 2 experiments). Data points represent individual mice. T tests were two-tailed. BM = bone marrow, BV = blood vessel, CSD = chronic social defeat stress, HC = home cage, iv = intravascular, SSC = side scatter complexity, TomL = tomato lectin. Data shown as mean ± SEM. *$p$ < 0.05. Source data are provided as a Source Data file. Schematic created in BioRender. Kigar, S. (2025) https://BioRender.com/p6bsp7f.

We also examined relationships between iv- meningeal monocyte and lymphocyte levels with the corresponding cells in other tissues but did not find any significant relationships (Fig. S11A, B). These data support a role for skull-to-meninges communication in CSD and highlight the unique composition of the meningeal immune environment compared to other tissues.

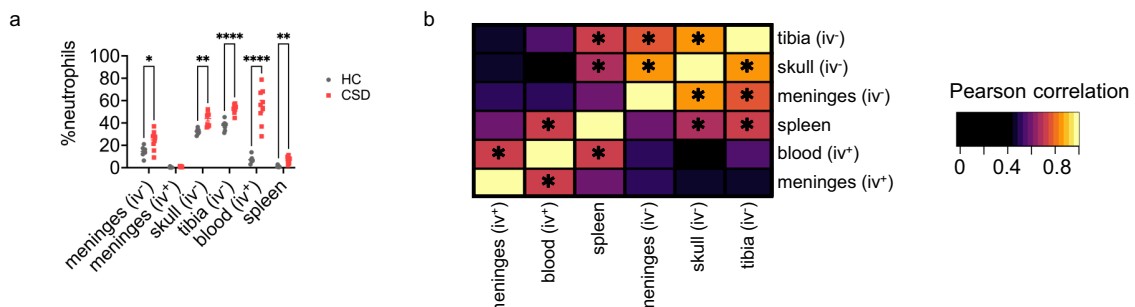

**Fig. 5 | CSD leads to widespread tissue increase in neutrophils, relationships present between skull and meningeal neutrophils support microscopy data. a** Flow cytometry shows increased neutrophil levels throughout the body following CSD stress in C57BL/6 J mice (REML model, main effect of group, ****$p < 0.0001$, $F_{(1,15)} = 112.8$, $n_{HC} = 8$, $n_{CSD} = 9$; for blood, $n_{HC} = 7$). See Table S9 for post-hoc statistics. **b** Unsupervised hierarchical clustering of data from (**a**) shows significant relationships between BM (*Skull*: ***FDR = 0.0005, r = 0.83, t(15) = 5.81. *Tibia*: **FDR = 0.009, r = 0.75, t(15) = 4.36) and iv⁻ meningeal neutrophils, but not with blood. Blood and iv⁺ meningeal neutrophils were significantly correlated (*FDR = 0.018, r = 0.72, t(15) = 3.97). Asterisks in heatmap indicate significant Pearson correlation after Bonferroni correction for multiple comparisons. See Table S10 for correlation coefficients and FDR values. Data points represent individual mice; data represent 4 independent experiments. Multiple tissues were collected from individuals; repeat measures were accounted for in (**a**). BM = bone marrow, CSD = chronic social defeat stress, HC = home cage, iv = intravascular. Data shown as mean ± SEM. *$p < 0.05$, **$p < 0.01$, ****$p < 0.0001$. Source data are provided as a Source Data file.

## Single cell RNA sequencing (scRNAseq) reveals increased neutrophils in CSD meninges, increased proinflammatory signaling, neutrophil heterogeneity

scRNAseq was used to profile meningeal immune cells from behaviorally stratified HC and CSD mice (Fig. 6a, b). Clustering identified 20 cell clusters which were manually annotated based on expression of canonical lineage genes (Fig. 6c, S12A–C). There was a large cluster of neutrophils, identified by high *Cxcr2* expression, which did not appear to be undergoing local cell proliferation (Fig. S12D, E). Consistently with our IHC and flow cytometric data, meninges from stressed mice showed a relatively greater proportion of neutrophils compared to HC (Fig. 6c, lavender bars).

Subclustering of neutrophils alone identified 6 subclusters with roughly equal proportions of each subtype in both HC and CSD meninges (Fig. S13A). Based on expression of genes related to primary, secondary, and tertiary granule formation, as well as suppression of transcriptional machinery (Fig. S13B), these subclusters likely correspond to states of increasing neutrophil maturation[30–32]. Pseudotime analysis of differential gene expression (DGE) across the subclusters supported this hypothesis (Fig. S13C).

We integrated our data with two publicly available multi-tissue datasets containing pre-neutrophils and neutrophils from blood and multiple BM locations[28,30]. When projected onto a shared PCA space, our meningeal neutrophils did not cluster with mature blood neutrophils but instead were most similar to immature BM neutrophils. In contrast, our meningeal pre-neutrophils clustered with the bulk and pseudobulked pre-neutrophil Evrard and Kolabas samples, suggesting successful integration of these three datasets (Fig. S14A, B).

### Validation of MHCII downregulation on CSD neutrophils

Examination of differential gene expression (DGE) in the pooled neutrophil cluster showed reduced transcription of *H2-Ab1* and *H2-Aa* (Fig. 6d)—2 of the 4 genes comprising the major histocompatibility complex, class II (MHCII). Our flow cytometry data showed this was also true at the level of protein expression: in CSD mice, there was a 2.3-fold reduction in the percentage of MHCII⁺ iv⁻ meningeal neutrophils (Fig. 6e).

### Effects of CSD stress on neutrophil cell size and actin polymerization

Gene set enrichment analysis (GSEA) of DEGs in the pooled neutrophil cluster uncovered 80 biological pathways associated with CSD stress (Fig. 6f, S15). 8 of the 41 positively enriched pathways indicated CSD

was associated with altered neutrophil cell size and/or actin polymerization, e.g. the gene ontology (GO) pathway "regulation of cellular component size" (Fig. S16A, B). Chronically high demand for neutrophils leads to the release from BM of immature cells[33,34], which are both volumetrically larger[35] and more rigid[36] than mature neutrophils. Given this and our GSEA results, we hypothesized there would be an increase in enlarged BM-derived neutrophils in the meninges.

To test this, we analyzed neutrophils on an Amnis ImageStream imaging flow cytometer (Figs. 6g, S16C). We verified the expected CSD increase in meningeal and blood neutrophils (Fig. S16D) and identified a subset of relatively enlarged neutrophils in both tissues. There was a significant, ~3x increase in the enlarged neutrophil subset in CSD meninges, but not in blood (Figs. 6g, S6D). There were no group differences in actin mean fluorescence intensity (MFI) for meningeal neutrophils, but CSD led to increased actin MFI in circulating neutrophils (Fig. S16E).

Pathway analysis also implicated increases in the GO pathway "cytokine production" in stressed animals, with increased expression of *Cxcr2* in CSD neutrophils (Fig. S17A, B). However, we did not detect changes in either the frequency (%CXCR2⁺) or protein expression (CXCR2 MFI) in any examined tissue (Fig. S17C, D). We did observe an increase in CSD CXCL1 plasma concentration, perhaps driven by increased mRNA expression in the liver, and decreased *Cxcl12* expression in CSD tibia (Fig. S17E). Collectively, this should promote egress of neutrophils into the blood stream, i.e. through CXCL1 binding to CXCR2 and decreased CXCL12 binding to CXCR4. Finally, detection of ROS by intracellular flow cytometry provided no evidence for increased neutrophil ROS production on a per cell basis, consistent with the GO pathway "cell redox homeostasis" (Fig. S18A–D).

### Enriched IFN-I signaling in meningeal neutrophils

Our GSEA results also identified enrichment of the GO pathway "response to type I interferons" (Fig. 6f, S19A), which we pursued further given extensive literature supporting a role for IFN-I in promoting depressive-like behavior[7–9]. Across all meningeal cell clusters, leading-edge genes *Interferon induced transmembrane proteins (Ifitm)-2* and *Ifitm3* showed greatest expression in neutrophils and monocytes (Fig. S19B). *Ifitm2* and *Ifitm3* expression was strongly increased in CSD-stressed meningeal neutrophils generally (Fig. 6d, S19C) and in several neutrophil subclusters (Fig. 7a, b). In contrast, monocyte expression of *Ifitm2* and *Ifitm3* did not show stress-associated changes, and monocytes did not show enrichment of interferon-mediated pathways (Fig. S19C, S20A, B).

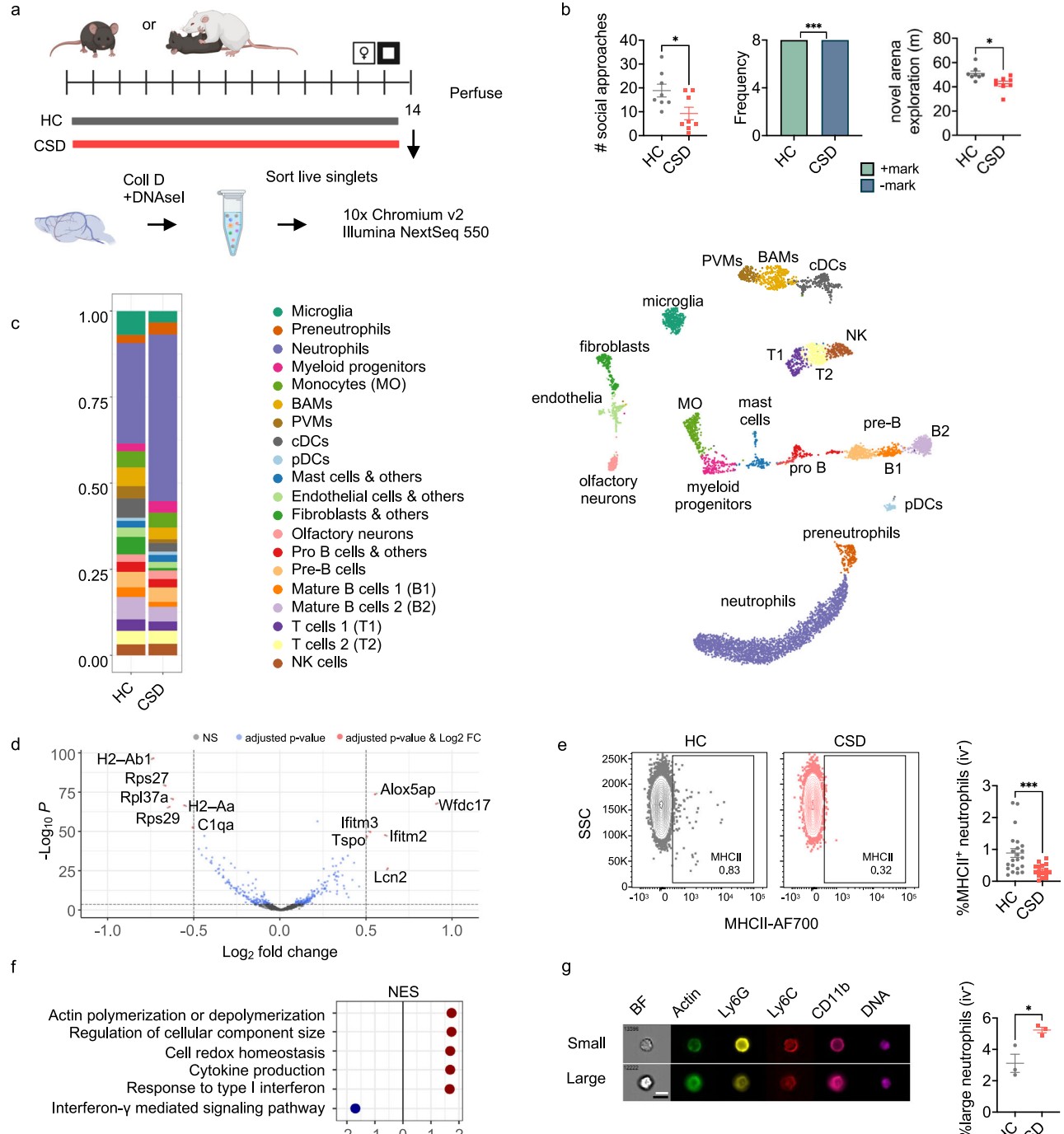

**Fig. 6 | Single-cell RNA sequencing (scRNAseq) of meningeal tissue validates CSD neutrophilia; biological assays support transcriptional findings.**
**a** Timeline for sample collection and processing for scRNAseq (N = 1 experiment).
**b** C57BL/6 J mice ($n_{HC}$ = 8, $n_{CSD}$ = 8) were behaviorally phenotyped prior to scRNA-seq; a subset with representative phenotypes for anhedonic- (*SI*: *p = 0.022, t = 2.6, df = 14, 95% CI [−1.78, −0.16]. *USM*: Fisher's exact test, ***p = 0.0002), and anxiety-like (*OF*: **p = 0.0022, t = 3.6, df = 17, 95% CI [−15.10, −3.93]) behavior were selected (shown in order from left to right). Data points represent individual mice. **c** *Left*: Visualization of recovered cells as a proportion of total cells recovered per group ($n_{HC}$=8, $n_{CSD}$ = 4). *Right:* Meningeal scRNAseq reveals 20 distinct immune cell clusters; data points represent individual cells. Plot shown previously[20]. **d** Volcano plot showing DEG between CSD and HC in the neutrophil cluster (excludes preneutrophil cluster). Indicated points represent DGE with LFC > 0.5 and FDR *p* < 0.001. **e** Flow cytometric validation of reduced iv⁻ meningeal neutrophil MHCII expression in (**d**). *Left*: Contour plots from representative samples; points represent cells from one mouse. *Right*: data points represent individual mice (Mann Whitney, ***p = 0.0006,

$U$ = 64.50, Hodges-Lehmann estimate = −0.40, $n_{HC}$ = 22, $n_{CSD}$ = 16. N = 5 experiments). **f** Gene set enrichment analysis (GSEA) revealed several enriched pathways related to cell size and the cytoskeleton (for complete list, see Fig. S15). **g** Imaging flow cytometry was used to visualize neutrophils. Representative images from individual meningeal samples. Two differently sized populations emerged, as depicted (white bar = 'small', black bar = 'large'). There were nearly 3x more enlarged neutrophils in CSD meninges compared to HC (*p = 0.025, t = 3.5, df = 4, 95% CI [0.44, 3.81], $n_{HC}$ = 3, $n_{CSD}$ = 3; N = 1 experiment), consistent with actin- and cytoskeleton-related pathway enrichment in (**f**). Data points represent individual mice; for more details, see Fig. S16. Two-tailed tests were used in (**b, d, g**); all tests were unpaired t tests unless otherwise indicated. BAM = border associated macrophage, BF = brightfield, cDC = conventional dendritic cell, CSD = chronic social defeat stress, HC = home cage, NES = normalized enrichment score, pDC = plasmacytoid dendritic cell, PVM = perivascular macrophage, SSC = side scatter complexity. Data shown as mean ± SEM. *p < 0.05, **p < 0.01, ***p < 0.001. Source data provided in Source Data file. Schematics from BioRender. Kigar, S. (2025) https://BioRender.com/v98waxl.

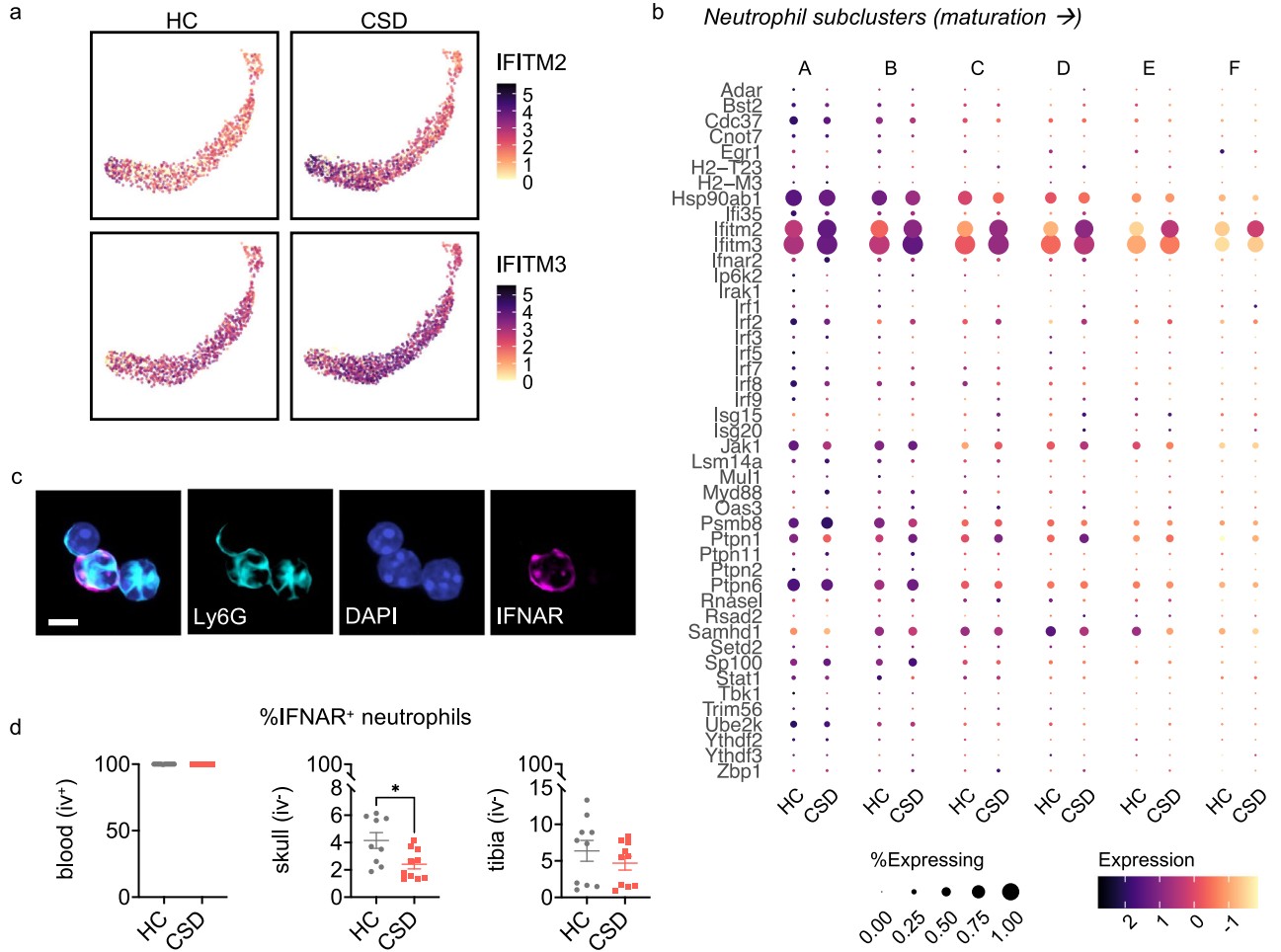

**Fig. 7 | Migration of IFNAR+ neutrophils from skull BM to the meninges may underlie the type I interferon neutrophil signature seen in CSD stressed mice.** **a** UMAP shows expression of *Ifitm2* and *Ifitm3*, the leading-edge genes for enrichment of the GO: "Response to type I interferon" pathway in neutrophils, in CSD compared to HC animals. Data points represent individual cells from the indicated group. **b** Dot plot showing gene expression in each neutrophil subcluster, which represents neutrophil maturation (see Fig. S13) for all genes comprising this pathway. Expression is scaled to mean ± standard deviation. **c** Confocal microscopy showing skull bone marrow neutrophils; IFNAR+ staining in more 'mature' neutrophil, based on nuclear morphology. Scale = 5μm. NB: these data are not quantified. **d** Flow cytometry data for IFNAR+ neutrophils, normalized to total neutrophils, for blood, skull, and tibia. The population of IFNAR+ skull BM neutrophils was decreased in CSD stressed mice (Unpaired t test, two-tailed, *$p = 0.017$, $t = 2.7$, $df = 17$, 95% CI [−3.12, −0.36]) and may represent a migration event to the meninges. Data points represent individual mice ($n_{HC} = 10$, $n_{CSD} = 10$; for skull, $n_{HC} = 9$. N = 2 experiments). See Fig. S21 for more detail. BM = bone marrow, CSD = chronic social defeat, HC = home cage, IFNAR = interferon-α/β receptor. Data shown as mean ± SEM. *$p < 0.05$. Source data are provided as a Source Data file.

## IFNAR+ neutrophil egress from skull BM

IFN-I sensing meningeal neutrophils could migrate to meninges from peripheral BM tissue (via the circulation), or directly from skull BM[26–28]. We would predict a decrease in IFNAR+ neutrophils in the source tissue as they exit to traffic to the meninges. IFNAR+ neutrophils were present in blood, skull and tibial BM (Fig. 7c, S21A). As predicted, comparison of HC vs CSD tissue revealed a 1.7-fold decrease of IFNAR+ neutrophils in CSD mice that was specific to skull BM (Fig. 7d). This result further suggests skull BM as the source of stress-induced meningeal neutrophils.

## IFNAR+ neutrophils are a relatively mature pool of BM neutrophils

We noted three distinct populations of neutrophils – IFNAR^neg, IFNAR^lo, and IFNAR^hi – with tissue-specific distributions and expression of activation markers (Fig. S21B, C). In blood, the proportion of IFNAR^lo and IFNAR^hi neutrophils was approximately equal, with no detectable IFNAR^neg neutrophils. BM IFNAR+ neutrophils had higher side scatter complexity (SSC)−a phenomenon corresponding to increased granule content and/or more complex nuclear morphology reflecting cell maturation. Conversely, the majority (>94%) of BM neutrophils were IFNAR^neg, consistent with an expected reservoir of immature cells in this niche. We also examined MFIs in each of the three IFNAR neutrophil subtypes. Ly6C is expressed in response to IFN-I signaling[37]. Consistently, Ly6C expression was highest in IFNAR^hi neutrophils from all three tissues (Fig. S21C).

## IFNAR blockade improves the behavioral response to CSD stress

We assessed whether peripheral blockade of IFN-I signaling rescues the negative behavioral sequelae associated with CSD stress by administering an IFNAR-blocking antibody to LysM^gfp/+ mice (Fig. 8a). As expected, in the USM test for anhedonia, more HC+control antibody (IgG) mice marked compared to the CSD+IgG group (Fig. S22A). Anti-IFNAR treatment rescued the CSD phenotype, with no difference in

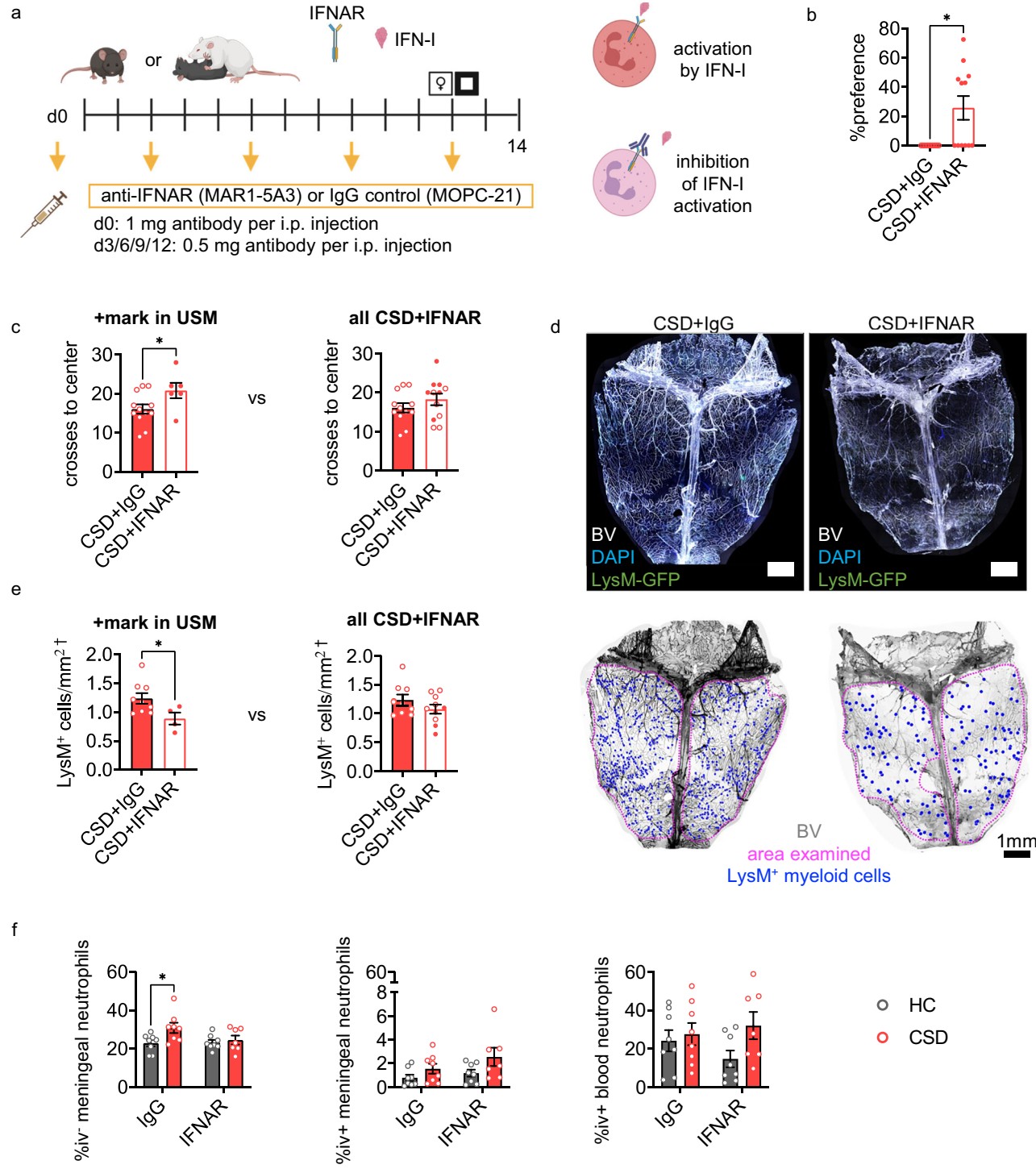

marking frequency between HC+IgG and CSD + IFNAR groups. For other examined behavioral parameters, HC animals appeared 'stressed', which we determined was influenced by repeated injections (Fig. S22B, C). We therefore examined the CSD group alone.

In the USM task, anti-IFNAR treatment protected mice from the depressive-like effects of CSD (Fig. 8b). However, USM data for the CSD+anti-IFNAR group was non-normally distributed, with distinct groups of animals that marked (+) or did not mark (-) in the test (Fig. S22D). We therefore treated these animals separately in our remaining analyses. There was a significant difference in the OF task between CSD+IgG and CSD+anti-IFNAR(+) mice, where CSD+IgG animals were more anxious (Fig. 8c). There were no other significant differences in behavior between groups (Fig. S22E).

**IFNAR blockade attenuates neutrophil migration into, but not B cell egress from, meningeal tissue in CSD stressed mice**

Hand counting of non-vascular meningeal GFP+ cells from these mice showed a decrease in the CSD+anti-IFNAR(+) group (Fig. 8d, e). No other subpopulations (i.e., vascular, abluminal) showed an effect of anti-IFNAR treatment (Fig. S22F). In WT mice, IFNAR blockade successfully attenuated iv- neutrophil accumulation in the meninges (Fig. 8f, Table S11) with the expected increase in control IgG antibody-treated CSD mice. Iv+ neutrophils in the meninges and blood were not impacted by anti-IFNAR treatment, though iv+ meningeal neutrophils were elevated by CSD. The effects of anti-IFNAR treatment appeared specific to iv- meningeal neutrophils. Notably, depletion of iv- meningeal B cells by CSD stress was unimproved. There were otherwise no

**Fig. 8 | Type I interferon receptor (IFNAR) blockade may improve CSD-stress related behavioral anhedonia and prevent meningeal neutrophil accumulation. a** Schematic showing drug-delivery schedule for IFNAR blocking antibodies. Mice were injected with either matched IgG control antibody or anti-IFNAR antibody on the days indicated with yellow arrows. **b** USM results from LysM$^{gfp/+}$ mice; see Fig. S22 for comparison with HC+IgG mice and for more detail. CSD+anti-IFNAR treated mice showed behavioral improvement compared to CSD+IgG mice (Mann Whitney, *$p = 0.014$, $U = 36$, Hodges-Lehmann estimate=21.35, $n_{CSD+IgG} = 12$, $n_{CSD+IFNAR} = 12$. N = 5 experiments). Hereafter we considered these LysM$^{gfp/+}$ mice as two separate groups – those that marked in the USM test (+, indicated with closed circles) and those that didn't (-, indicated with open circles). **c** *Left:* CSD+anti-IFNAR(+) mice showed improvement in the OF task for anxiety-like behavior compared to CSD+IgG-treated mice (*$p = 0.046$, $t = 2.2$, $df = 16$, 95% CI [0.10, 9.40]). *Right:* No improvement in OF behavior overall for CSF+anti-IFNAR-treated mice. $n_{CSD+IgG} = 12$, $n_{CSD+IFNAR} = 12$ (within IFNAR-treated, $n_{USM+} = 6$, $n_{USM-} = 6$). **d** *Top:* Representative images for dorsal whole-mount meningeal tissue from CSD+IgG and CSD+anti-IFNAR treated mice. Scale bar = 1 mm. *Bottom:* Images from HC and CSD mice showing hand-counted LysM-GFP$^+$ cells,

normalized to indicated area. **e** *Left:* Quantification of total nonvascular (>10μm away from a blood vessel) meningeal LysM-GFP$^+$ cells shows reduced myeloid cell accumulation in CSD+anti-IFNAR(+) treated mice (*$p = 0.048$, $t = 2.2$, $df = 11$, 95% CI [−0.69, −0.0043]). *Right:* No improvement in LysM-GFP$^+$ cells accumulation overall for CSF+anti-IFNAR-treated mice. $n_{CSD+IgG} = 9$, $n_{CSD+IFNAR} = 10$ (within IFNAR-treated, $n_{USM+} = 4$, $n_{USM-} = 6$). $^\dagger$square root transformed values. **f** In C57BL/6 J wild type mice there was an IFNAR-mediated rescue in meningeal neutrophil accumulation following CSD stress, with expected differences in IgG control groups (2-way ANOVA, *$P = 0.033$, $F_{(1,27)} = 5.0$, $n_{CSD+IFNAR} = 7$, otherwise n = 8. N = 2 experiments). See Table S11 for post-hoc statistics. Significance levels shown are from post-hoc testing adjusted for multiple comparisons. There was no effect of anti-IFNAR treatment on iv$^+$ meningeal or blood neutrophils or other cell types; see Fig. S23. Data points represent individual mice. Two-tailed tests were used in (**b, c, e**); all tests were unpaired t tests unless otherwise indicated. CSD = chronic social defeat, HC = home cage. Data shown as mean ± SEM. *$p < 0.05$, **$p < 0.01$. Source data provided in Source Data file. Schematics from BioRender. Kigar, S. (2025) https://BioRender.com/rvrmii4.

---

differences in other meningeal or blood immune cell populations (Fig. S23A).

Like LysM$^{gfp/+}$ mice, C57BL/6 J mice were behaviorally impacted by repeated injection stress (Fig. S23B, S24A). Unlike LysM$^{gfp/+}$ mice, the effects of injection stress had no impact on meningeal myeloid cell levels (Fig. S24B).

### Reduced expression of chemorepellent factors in CSD stress may permit neutrophil entry into the leptomeninges

As class 3 semaphorins (Sema3) are known to be involved in neutrophil trafficking from skull BM to meninges[38], we examined their expression in our previously published microarray data set from meningeal tissue[20]. *Sema3b* expression was significantly reduced in CSD mice compared to HC (Fig. S25A, B).

## Discussion

Our data robustly demonstrate increased neutrophils in the meninges following CSD stress, and that neutrophilia correlates with a negative behavioral response to chronic stress, summarized in Fig. 9. Our data lend further support to the extant literature suggesting direct trafficking of immune cells from the skull BM to the meninges[25–28], extending the evidence for this phenomenon to conditions of psychosocial stress.

Several independent lines of evidence from our study bolster support for this conclusion. First, in cleared skull tissue, we observed more LysM-GFP$^+$ myeloid cells present in skull-to-meninges vascular channels of CSD mice. While LysM-GFP expression is not restricted to neutrophils, unsupervised hierarchical clustering of flow cytometry data using the neutrophil-specific Ly6G antibody revealed that non-vascular meningeal neutrophil levels closely mirrored those of skull BM, but not blood. Further, integration with public datasets[28,30] showed that meningeal neutrophils have a transcriptional profile distinct from mature neutrophils in blood, and are more similar to immature BM neutrophils.

In this study we have replicated the blood neutrophilia described in other preclinical stress models[14,39] and in people with MDD[17], and find that meningeal neutrophilia outlasts the well-described peripheral effects of stress on neutrophils following cessation of stress. The distinctiveness of the responses and regulation of blood and meningeal neutrophils is further underscored by the different effects of time and stressor type on immune cell dynamics in these tissues. Although iv$^-$ meningeal neutrophil levels return to baseline within a week of recovery from CSD stress, iv$^-$ meningeal B cell levels are persistently reduced by about 30%. While a complete exploration on the immunology of defeat is beyond the scope of the current manuscript, future efforts to explore whether the persistent reduction in B cell numbers

'primes' the meninges for an exacerbated innate immune response, and how long B cell numbers remain low, is warranted.

We have previously shown that B cell depletion increases both IFN-I signaling and meningeal neutrophil levels in $Cd19^{-/-}$ mice[20], which are deficient in B cells[40]. Here, we replicated the depleting effect of CSD stress on meningeal B cells and demonstrated that CSD stress leads to an enriched IFN-I signature in neutrophils but not in other meningeal cell types. Our unbiased identification of this pathway in multiple CSD cohorts using multiple experimental techniques is strengthened by consistent evidence from humans, non-human primates, and rodents that IFN-I is sufficient to induce depression and depressive-like behavior[7–9].

It is not presently clear how meningeal neutrophils influence behavior given they do not appear to enter the brain parenchyma. Further exploration of the pathways by which neutrophils enter the meninges and exert their effects on brain tissue is merited. We observed a stress-associated reduction in meningeal *Sema3b* expression in bulk meningeal tissue; decreased expression of *Sema3a* by arachnoid barrier cells has been previously shown to regulate neutrophil entry into the subarachnoid space from skull[38], suggesting a potential role for Sema3 family members in this process. Future efforts to determine what stress-related factors influence *Sema3* expression will be important. differentially expressed gene (DEG) analysis of our single-cell RNA sequencing data indicated altered innate immune system activation across the neutrophil cluster. For example, the most strongly upregulated transcript in CSD meningeal neutrophils was *Wfdc17*, which inhibits NFκB-mediated microglial activation[41] and is proinflammatory in monocytes[42]. Decreased *C1qa* expression following stress may hamper phagocytosis and clearance of dying neutrophils by nearby macrophages, leading to sustained local inflammation[43,44].

Likewise, decreased meningeal neutrophil expression of MHCII genes *H2-Ab1* and *H2-Aa*, which we verified by flow cytometry, may inhibit antigen presentation to CD4$^+$ T cells[45,46]. B cells also present antigen to CD4$^+$ T cells in dural-associated lymphoid tissue (DALT) of the meninges[47]. Along with our previous finding that CSD leads to decreased levels of meningeal B cells[20]—reproduced in the current study –these data suggest CSD may be associated with reduced or disrupted antigen presentation to meningeal T cells by multiple cell types.

*Ifitm2* and *Ifitm3*, the genes driving the stress-associated upregulation of interferon response pathways in neutrophils, are known to be expressed in a mature, hyperinflammatory neutrophil subtype observed in human and mouse blood[32,48,49]. Our characterization of IFNAR expressing neutrophils in different tissue compartments revealed highest Ly6C expression in IFNAR$^{hi}$ neutrophils, which—in BM

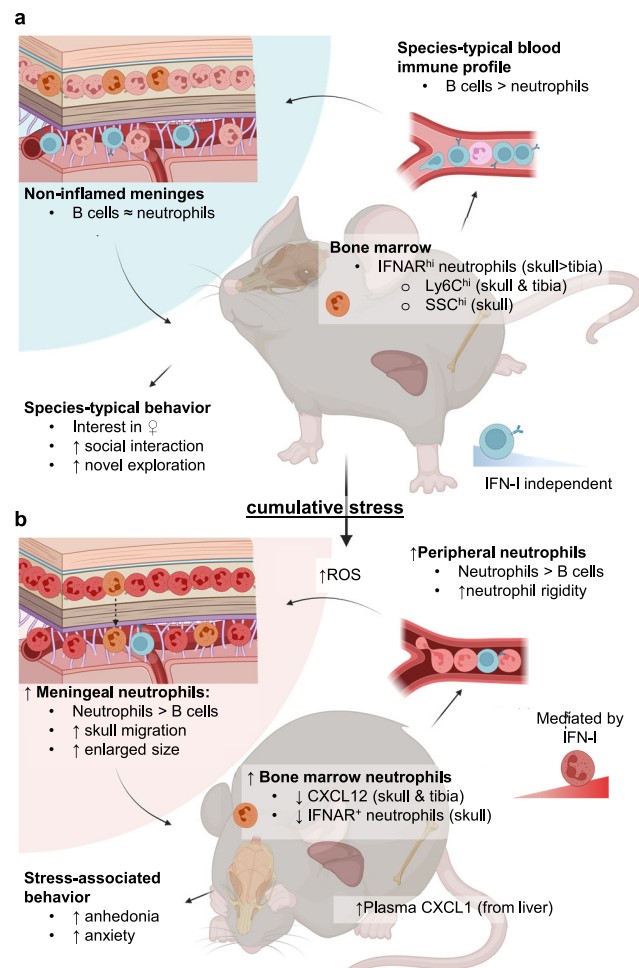

**Fig. 9 | Proposed model for how chronic, but not acute, stress leads to dysregulation of the meningeal environment. a** At baseline, both skull and tibia BM contain a small population of IFNAR⁺ neutrophils (orange), further divisible into IFNARˡᵒ and IFNARʰⁱ populations. Relatively mature IFNARʰⁱ cells express more Ly6C and may thus be more proinflammatory[37]. In blood, B cell numbers greatly outnumber neutrophils; in the meninges these two populations are roughly equal. **b** Acute stress exposure leads to repetitive release of neutrophils into the blood but is not sufficient for accumulation of neutrophils in the meninges (Figs. 3a, S6A, S7A). Prolonged exposure to psychosocial stress leads to a decline in meningeal B cells (Fig. S2C) that precedes an increase in meningeal neutrophils (Figs. S6A, S7A) and is prolonged for up to a week of recovery post-CSD (Fig. 3b, S6B, S7B). Whereas CSD-associated meningeal neutrophilia appears to be mediated by IFN-I signaling, meningeal B cell depletion does not (Figs. 8f, S23A). CSD stress causes random neurovascular damage[21]—potentially via stalled and rigid ROS-producing neutrophils in brain capillaries (Figs. S10, S16E, S18)—which amplifies neutrophil recruitment from adjacent skull bone marrow (Fig. 4g, h). Increased expression of the neutrophil chemoattractant *CXCL1* in the liver, increased CXCL1 protein in plasma, and reduced BM expression of CXCL12—which promotes retention of neutrophils—may contribute to the elevated circulating pool of neutrophils in blood (Fig. S17), though there were no detectable changes related to this pathway in the meninges. BBB = blood brain barrier, CSD = chronic social defeat, CXCL = C-X-C motif chemokine ligand, HC = home cage, IFN-I = type I interferon, IFNAR = interferon-α/β receptor, ROS = reactive oxygen species, SSC = side scatter complexity. Fig. created in BioRender. Kigar, S. (2025) https://BioRender.com/yf2zm8x.

—appeared to also be more mature. Consistently, qualitative assessment of IFNAR⁺ compared to IFNAR⁻ skull neutrophils showed more hyper-segmentation of the nucleus—a feature of neutrophil maturation. High Ly6C expression, as seen in IFNARʰⁱ neutrophils, is associated with greater proinflammatory capacity in monocytes[50]. This

suggests IFNAR⁺ neutrophils are a distinct, relatively mature BM subset that may also be more proinflammatory.

Although we did not assess IFNAR⁺ neutrophil staining in the meninges or in cleared skull tissue directly, our data are consistent with IFNAR-mediated trafficking of neutrophils from the skull BM to meninges. We demonstrated a skull BM-specific decrease in IFNAR⁺ neutrophils following CSD stress coupled with transcriptomic evidence of increased IFNAR signaling in meningeal neutrophils.

IFN-I depletion in C57BL/6 J mice normalized CSD-related meningeal neutrophil levels. While replication of these effects in LysMᵍᶠᵖ/⁺ mice presented additional complexity, CSD+anti-IFNAR–treated mice showed improved anhedonic behavior. We observed meningeal neutrophilia in otherwise non-stressed HC mice who failed to show preference for a sexually hedonic stimulus. This is consistent with reports suggesting individual differences in peripheral immunity influence stress susceptibility[51].

Given that the MAR1-5A3 IFNAR antibody clone we used is not thought to cross the blood brain barrier (BBB)[52] and improves BBB integrity[53], we assumed that the mediating effects of anti-IFNAR treatment on the negative sequelae of CSD stress did not involve brain parenchymal cells. However, other studies have highlighted a microglial IFN-I response during stress[54,55]. In agreement with this, we found upregulation of the IFN-I response genes *Ifitm2* and *Ifitm3* in microglia from CSD mice[21]. Future experiments using neutrophil-specific IFNAR knockout will be important to establish whether the beneficial effects of IFNAR depletion are restricted to peripheral neutrophils, or if microglia and other cell types are involved.

What processes drive neutrophil infiltration toward the meninges following CSD stress? While the cellular source of stress-related IFN-I is currently unclear, one possibility is that microglial production of IFN-I is responsible for neutrophil chemotaxis into the meninges. Using bulk transcriptomics, we have previously demonstrated enrichment of pathways related to IFN-I production in CSD microglia[21]. Microglia-specific depletion of IFN-I production, coupled with tissue clearing or in vivo imaging studies of fluorescently labeled neutrophils in mice given IFNAR antibody therapy, will be important to definitively corroborate this mechanism.

Alternatively, we have shown that CSD stress leads to randomly occurring microhemorrhages in the neurovasculature[21,24]. As neutrophils are typically 'first on the scene' to sites of brain injury[56], their active recruitment may facilitate progression, maintenance, or resolution of this stress-induced damage. Consistently, skull BM-to-meninges neutrophil trafficking has been demonstrated in hemorrhagic stroke, wherein BBB damage is present[57]—though, notably, we find no evidence for infiltration of neutrophils or other leukocytes into the brain parenchyma of CSD-stressed mice[21,22].

Another possibility is that chronic release of neutrophils into the bloodstream via repeated exposure to stress[14,15] drives formation of microhemorrhages[21,22,24], though we did not test this directly given our focus on the meninges. In neurological conditions, neutrophils are notorious drivers of bystander damage due to their release of toxic reactive oxygen species (ROS)[56–58]. We did not find evidence of increased neutrophil ROS production on a per-cell basis following stress, although the overall increase in ROS-producing neutrophils may exacerbate risk for bystander damage.

Blood-derived CSD neutrophils also showed signs of being both more rigid and proinflammatory. Specifically, we observed increased phalloidin staining intensity in CSD blood neutrophils, consistent with enhanced formation of filamentous actin (F-actin)[59]. Increased F-actin formation may cause neutrophil rigidity, thereby increasing the likelihood of neutrophil 'stalling' in narrow brain parenchymal capillaries. Indeed, we saw more LysM-GFP⁺ cells 'stuck' throughout the neurovasculature in CSD brains, and more neutrophil-like cells coprecipitating with neurovascular endothelia (NVE), consistent with other reported data[60]. Future efforts to characterize NVE-associated

neutrophils and how they differ from non-adherent blood and meningeal neutrophils will be important.

The apparently even distribution of LysM-GFP cells throughout the meninges and brain—consistent with the sporadic, stress-associated microhemorrhages we have previously investigated[21,24]—is in contrast to reports suggesting localized BBB damage preclinically and clinically[61]. Our scRNAseq data suggested increased expression of *Translocator protein* (*Tspo*) and related accessory proteins that mediate neutrophil chemotaxis[62,63]. We recently demonstrated a relationship between meningeal TSPO signal intensity and brain regions associated with clinical MDD[64]. TSPO signal intensity in microglia is related to duration (in years) of depressive symptoms[65]. This could suggest our observations represent a relatively early stage in the progression of depressive symptoms; future efforts to determine the precise nature of this myeloid 'sticking' behavior over the progression of MDD will be important.

Importantly, depressive symptoms are common in neurological disorders like stroke[66] and neurodegeneration[67], and in preclinical models of neurodegeneration, neutrophil capillary stalling has been linked to reduced brain perfusion and memory deficits[68]. This raises the possibility that altered neutrophil properties are a shared biological feature between neurological and psychiatric disorders; comparison of neutrophil phenotypes across brain disorders may suggest future transdiagnostic treatment targets. IFN-I signaling deleteriously impacts both neuronal injury after stroke[69] and synapse loss in AD[70]. Given the relationship between depressive-like behavior and IFN-I[7–9], and epidemiological evidence that depression doubles the risk for dementia late in life[71], future work in this area is warranted.

A major limitation of this and similar studies is the exclusive use of adult male mice. Future extension of this investigation to females will be crucial, especially as depression disproportionately affects women[72]. In addition, females may be more acutely susceptible to BBB permeability after stress induction[73–75] and exhibit a more robust antiviral IFN-I response[76,77]. These mechanisms could lead to sex-specific vulnerabilities to meningeal neutrophilia and associated behavioral outcomes.

Another limitation of our study is that social defeat stress is obligatorily associated with some degree of fight-inflicted wounding. Despite our efforts to mitigate this, we observed that higher wound scores were associated with greater levels of iv⁻ meningeal monocytes. This may require a more nuanced approach when translating results from social defeat stress models into a clinical setting, as has been discussed elsewhere[78].

Relatedly, systemic knockdown to test the impact of neutrophil depletion over the course of the stress paradigm was associated with animal welfare concerns. Specifically, wound-healing deficits resulting from neutrophil depletion are poorly compatible with the CSD model that results in fight-related wounds. Because of these welfare concerns, more localized strategies targeting the skull BM will be useful in future follow-up studies, which may also permit opportunities to demonstrate directionality of neutrophil migration, i.e., using intravital imaging.

Finally, depletion of IFN-I signaling showed pleiotropic, strain-dependent effects, with greater sensitivity to repeat injection in HC C57BL/6 Js compared to HC LysM$^{gfp/+}$ mice. In general, LysM$^{gfp/+}$ mice behaved similarly, but not identically, to C57BL/6 Js following CSD. Increased LysM$^{gfp/+}$ anxiety-like behavior in the OF test following social defeat has previously been reported[79]. Likewise, in a spinal cord injury model, LysM$^{gfp/+}$ mice showed an anhedonic behavioral phenotype[80]. Thus, while we cannot fully account for the behavioral differences between LysM$^{gfp/+}$ and C57BL/6 J mice, we are reasonably confident that CSD 'works' as expected in these animals.

Our strengths include our whole organism approach to neutrophil dynamics and our triangulation and replication of our findings via multiple experimental approaches. We have generated robust evidence for chronic stress-induced meningeal neutrophilia, and our data-led investigation of candidate mechanisms revealed IFN-I-mediated migration of neutrophils from skull BM to the meninges as a potential treatment target in stress-associated depression.

## Methods

### Animals
All procedures were approved by the National Institute of Mental Health Animal Care and Use Committee (protocol #LCMR06) and conducted in accordance with National Institutes of Health guidelines. Strains used were either C57BL/6J male mice purchased from Jackson Labs (Bar Harbor, ME) or male LysM$^{+/gfp}$ offspring from C57BL/6J mice bred in our facility with LysM$^{gfp/gfp}$ mice, which strongly express GFP in neutrophils[81] (obtained from Dr. Dorian McGavern, NINDS). Upon arrival into the facility, purchased mice were randomly pair-housed in divided cages and given one week of acclimation to the facility. Mice bred in-house were weaned at 3 weeks of age into same-sex cages of 2-4 littermates. Sex was assigned based on external anatomy. Animals were housed under a reverse light cycle (lights off 8:00 AM to 8:00 PM), with food and water *ad libitum*. Temperature and humidity were maintained between 18-23 °C and 40-60%, respectively, by NIMH animal care staff. At 8-10 weeks of age, mice were randomly assigned to an experimental group. At sacrifice, mice were deeply anesthetized with isoflurane, confirmed by tail and leg pinch. Anesthesia was maintained until diaphragm rupture, at which point the animals were transcardially perfused before decapitation and tissue collection.

### Chronic social defeat (CSD)
CSD stress was performed as previously reported[21]. Briefly, the 'intruder' test mouse was introduced into the home cage of an aggressive, CD-1 (Taconic; Rensselaer, New York) retired breeder. The two were separated by a perforated barrier and given 24 h to acclimate; the barrier allowed for olfactory, visual, and auditory communication, but not tactile contact. Each day for either 1, 2, 4, 8, or 14 consecutive days, depending on the experiment, the barrier was lifted, and antagonistic encounters were allowed to occur for 5 m. Interactions were monitored by a trained individual to ensure the test mouse exhibited submissive behavior and conversely that the CD1 exhibited dominant behavior. The C57/CD1 pairs were maintained throughout the study unless the CD1 failed to show dominant aggression toward the C57. When this occurred, the C57 was paired with a different CD1 mouse until submissive behavior in the C57 was evident, and the new pairing was maintained thereafter. Efforts to minimize physical damage were taken, i.e. CD1 mice were lightly anesthetized with isoflurane and incisors were trimmed prior to starting the social defeat paradigm, and on a weekly basis thereafter. Test mice were shaved and inspected for the presence of wounds at the end of the experiment; wounds were scored on a scale from 1-10 (see Table S12). Animals with wound scores of 10 were excluded. Unless otherwise noted, animals were sacrificed exactly 2 h after exposure to the final defeat stress on day 14. We did not go beyond 14 days of defeat out of concern for animal welfare.

### Behavioral phenotyping
All behavioral testing was done by both male and female experimenters during the dark phase of the light cycle, prior to a defeat encounter on that day. On the day of testing, mice were moved to a separate behavioral room with red lighting and acclimated for one hour prior to running the behavioral assays. Tests were usually run on separate days, but when multiple tests were run on a single day, the animals were given an hour to recover in between tests. Behavioral tests were performed in the following order (least to most aversive):

### Urine scent marking (USM)
As previously described[23], mice were placed into a novel arena (50 ×50 x 50 cm) that contained a thick sheet of paper; one corner of this paper

was 'spotted' with 50 μL of urine from multiple estrus females, and testing was performed with the lights off while the experimenter was outside the room. After testing, the sheets of paper were sprayed with ninhydrin and heated to indicate the presence of proteins, allowing for visualization of urine marking. Photos were taken of the sheets by an experimenter blind to group identity, then analyzed in ImageJ[82]. Reduced preference marking for the female scent is indicative of social anhedonia.

### Open field (OF)

Exploration of the novel open field arena, 50 ×50 x 50 cm in dimension, was performed under dim white lighting (~25 lux). Mice were placed in the middle of the arena; total distance moved, crosses to center, and time in center over a 10 min testing period whilst the experimenter was out of the room were later analyzed with automated tracking software (Clever Sys TopScan Suite) to eliminate potential human bias. Fewer crosses to center, reduced time in center, and reduced movement are all indicative of increased anxiety-like behavior.

### Light/dark box (LD)

The LD test uses an acrylic box (50 cm×25 cm with 30 cm walls) with aversive lighting (~40 lux). Approximately 1/3 of the box is enclosed and dark; an opening allows crossover between light and dark sections. Mice were placed in the light compartment and allowed to move freely for 10 min while the experimenter was out of the room. Time spent in the light compartment and number of crosses between the light and dark sides were scored from video recordings using TopScan. Low scores indicated anxiety-like behavior.

### Social Interaction (SI)

Two perforated acrylic cylinders, one containing an unfamiliar CD-1 mouse and the other empty, were placed in the OF arena with red lighting. The test mouse was placed in the middle of the arena and allowed to freely explore for 10 m with the experimenter outside of the room. TopScan was used on captured videos to track approaches to the social stimulus and time spent investigating social vs. non-social stimuli. Fewer social interactions indicated anhedonic behavior.

### Tissue collection

Unless otherwise indicated, mice were euthanized 2 h after their final exposure to the defeat stressor. Tissue collection occurred between ~8am and noon, with modifications as indicated below. Venous blood was collected into EDTA tubes via puncture of the submandibular vein and kept on ice until processing. Retro-orbital injections were administered while mice were under light isoflurane anesthesia. After, mice were deeply anesthetized with isoflurane prior to perfusion with 35 mL room temperature PBS. When indicated, hindlegs were collected for tibial BM extraction. The head was decapitated, and intact skull cleaned with a scalpel to remove muscle and connective tissue.

### Histology

LysM[+/gfp] mice received a retro-orbital intravascular injection of either DyLite 649- or DyLite 594-conjugated tomato lectin (TomL, Cat #DL-1178, Vector Labs), which was allowed to circulate for 5 min to label blood vessels before perfusion. In addition to PBS, mice were perfused with 10 mL of 4% cold paraformaldehyde. Skulls were post-fixed in 4% PFA for 24 h at 4 °C before transferring to 25% sucrose solution for cryoprotection, dehydration, and preparation for imaging.

### Single cell suspensions

WT mice were retro-orbitally injected with 4 μg CD45-FITC (Cat. #103108; Biolegend). Mice were then allowed to recover; after 25 min of circulation, mice were anesthetized lightly for venous blood collection. The decapitated skull was placed in cold HBSS + 0.1% BSA on ice until further processing.

### Tissue clearing and analysis of skull-to-meningeal vascular channels

We used the CUBIC tissue clearing method[83] on whole skulls from LysM[+/gfp] mice. Tissue was prepared as for histology except that samples were transferred to PBS instead of sucrose after 24 h fixation. Decalcification of bone prior to tissue clearing was achieved by incubation in decalcification solution (10% EDTA, 15% imidazole) for 5–7 days at 37 °C with shaking[84]. The decalcification solution was refreshed once on day 3. Following tissue clearing, the whole skull was inverted and placed in a glass-bottom dish filled with fresh Reagent 2[83] and imaged using a Zeiss 780 confocal microscope fitted with 10x objective. 2–5 images were collected at random locations for each skull and analyzed using IMARIS 9.7. For each image, both the number of vascular channels (labeled with TomL) and the number of discreet LysM-GFP[+] cells in a channel were counted. For each individual sample, the ratio of averaged LysM-GFP[+] cells normalized to the average number of channels is presented.

### Peripheral blood preparations

~500 μL of venous blood was lysed in 8 mL ACK Lysis Buffer (Cat. # 351-029-721; Quality Biological, Inc.) for 5 min at room temperature, and the reaction stopped by diluting with 7 mL cold HBSS + 0.1% BSA. Cells were pelleted, washed, and prepared for staining.

### Skull and tibial BM preparations

To prepare for skull BM extraction, the dorsal calvarium was trimmed to be relatively flat, and meninges were removed under a dissecting microscope. Next, the skull was cut into small bone pieces with scissors in cold HBSS + 0.1% BSA. This entire slurry was transferred to a 70 μm cell strainer and mashed with the rubber end of a 3 mL syringe for approximately 2 min per sample. Tibia were prepared by first stripping away all tissue from the bone, then cutting the very top such that a 23 g syringe needle could be inserted to flush out the BM into cold HBSS + 0.1% BSA. This was next transferred to a 70 μm cell strainer and mashed with the rubber end of a 3 mL syringe. For both BM tissues, the resulting cell suspensions were then pelleted and prepared for flow cytometry.

### Meningeal dissection and single cell suspension

Meninges samples were collected by first cutting around the lateral sutures of the skull; the dorsal skull and ventral skull were transferred to a fresh petri dish filled with cold HBSS + 0.1% BSA and kept on ice while pial and arachnoid meningeal membranes were gently picked off the entire outer surface of the brain into a second 'working dish' with Dumont #5 forceps (Cat. #RS-5058; Roboz, Gaithersburg, MD). Extra care was taken to avoid inclusion of choroid plexus from the 4th ventricle. Once finished with the brain, skull pieces were transferred as necessary into the working dish to remove attached meninges; we avoided leaving skull pieces in the 'working dish' to minimize contamination with cells from skull BM. Upon completion of the meningeal dissection, samples were transferred to a fresh tube and cells were pelleted by centrifugation, then resuspended in 2 mL of BSA-free HBSS supplemented with 2.5 mg/mL Collagenase D (Cat. #11088858001; Roche) and 12.5 μL of 0.5 mg/mL DNAseI (Cat. #L5002139; Worthington) for cell dissociation. The samples were incubated at 37 °C for 30 min, diluted with cold HBSS + 0.1% BSA, and mashed through a 70 μm cell strainer into single cell suspension.

### Meningeal whole mount preparations and staining

The dorsal skull was carefully removed to retain maximal attachment of the meningeal layers from tissue prepared for histology, and brains were returned to fresh 25% sucrose until sunk for further staining. Dorsal meninges were gently peeled from the skull as a single sheet, mounted ventral (brain) side up/dorsal (skull) side down onto slides and encircled with a Pap pen.

## Meningeal immunohistochemistry

Meningeal whole mount samples from LysM[+/gfp] mice were dried, washed with PBS, blocked for 1 h in 4% normal goat serum in 0.4% Triton-PBS, and incubated in a humidity chamber overnight at room temperature with chicken anti-GFP (1:1000, Cat #13970, Abcam), diluted in 0.2% Triton-PBS with 2% normal goat serum. Approximately 18 h later, the samples were washed 3 x 5 m with PBS and incubated for 2 h with Chicken IgY-Alexa Fluor 488 (1:500, Cat #150169, Abcam) in 0.4% Triton-PBS. Samples were washed 2 x 5 m with PBS, given a quick rinse in deionized water, and counterstained with DAPI for 5 min. They were rinsed briefly again with deionized water then cover-slipped with PVA-DABCO (made in-house).

## Meningeal image acquisition and analysis

Meningeal whole mounts were tile scanned using a 20x objective at 1024 × 1024 resolution and online stitching with a Zeiss 780 confocal microscope. Two independent investigators blind to treatment hand-counted confocal images of the meningeal whole mounts to quantify the density and location of LysM-GFP[+] cells, which we assumed were neutrophils (characterized by high GFP expression, irregularly shaped nuclei, and a semi-round shape) within the tissue using ImageJ software[82]. There was excellent agreement in their counts (Pearson correlation, ****$p < 0.0001$, $r = 0.937$). LysM-GFP[+] cells were also examined in relationship to blood vessels; LysM-GFP[+] cells > 10 μm from a blood vessel were provisionally called 'non-vascular,' whereas LysM-GFP[+] cells ≤ 10 μm were called 'abluminal.' LysM-GFP[+] cells were otherwise considered intravascular.

## Brain image acquisition and analysis

Brains were sliced into 30 μm sections on a freezing microtome and stored in an ethylene glycol solution at -20 °C until staining. Free-floating sections were washed in PBS, blocked with 4% normal goat serum, and incubated overnight at room temperature using the same staining protocol as that used for meningeal tissue. Mounted sections were tile scanned using a 20x objective at 1024×1024 resolution and online stitching with a Zeiss 780 confocal microscope. To obtain LysM[+] myeloid cell counts, two blinded experimenters counted strongly GFP[+] cells using ImageJ software[82].

## p-Selectin staining of brain tissue

A subset of samples was stained for p-Selectin as above with the addition of p-Selectin (Cat#148301, Biolegend) diluted at 1:1,000 in 0.4% Triton-PBS. An additional secondary antibody conjugated to Alexa Fluor 555 (Cat#A-31570, Invitrogen) in 0.4% Triton-PBS at a dilution of 1:500 was used. Volocity (PerkinElmer) was utilized to quantify the amount of TomL[+] blood vessels associated with P-selectin staining, which was used to calculate the percent P-selectin coverage.

## Neurovascular endothelial cell isolation

As previously[22], brains collected from saline-perfused mice were placed in a gentleMACS™ C tube (Cat#130-093-237, Miltenyi). 2.5 ml of 2.5 mg/ml Collagenase D plus 10 μL of Solution A and 20 μL Solution Y from a Neural Tissue Dissociation Kit (Cat#130-094-802, Miltenyi) in HBSS was added to each sample, which was minced with a gentleMACS Dissociator for 45 sec using brain protocol 1. Tubes were incubated for 30 min at 37 °C with rotation, triturated 100x with a p1000 pipette tip, and strained through a 70 μm nylon mesh cell strainer (Cat#352350, BD Biosciences). The filter was washed with 20 ml HBSS, and eluate pelleted in a swinging bucket centrifuge at 300 g for 5 min. The cell pellet was resuspended in 30% isotonic Percoll (Cat#P4937, Sigma-Aldrich) then centrifuged at 800 g for 30 min with no brake. The myelin layer was removed, and Percoll and cell pellet layers were diluted 4-fold with HBSS for washing. This was centrifuged at 300 g for 5 min; the resulting cell pellet was then stained for flow cytometry.

## Cell staining and flow cytometry

To exclude dead cells, samples were stained with either Fixable Viability Dye eFluor™ 780 (Cat. #65-0865; eBioscience) for 10 min at room temperature at 1:2400 dilution, or with Zombie AQUA (Cat # 423102, Biolegend) for 15 min at room temperature at 1:100 dilution. The cells were then washed and blocked with 1 μL of normal goat serum (Cat #G9023-10ML; Sigma) and 1 μL of Fc block (Cat #101302; Biolegend) for 10 m on ice. 25 μL Brilliant Violet stain buffer (Cat #563794; BD Horizon) was then added, followed by an antibody master mix (Table S13). Cells were incubated on ice for 25 min, washed with PBS, and fixed in 2% PFA at room temperature for 15 min for analysis on either a BD LSR Fortessa or a Beckman Coulter CytoFLEX flow analyzer. Compensation was performed for each session using UltraComp eBeads (eBioscience 01-2222-42) conjugated to antibodies used in the sample panels. Viability dye and GFP[+] controls used cells instead of beads. Data were analyzed using FlowJo (BD) software (multiple versions) with manual gating. Absolute cell counts were determined using CountBrite counting beads (ThermoFisher, catalog #C36950). See Supplementary Information for gating strategies used in individual experiments.

## Meningeal scRNAseq

Meningeal scRNAseq data were acquired from 8 non-stressed HC mice and 4 stress-susceptible CSD mice as described previously[20]. In brief, live, nucleated, singlet cells (DAPI-DRAQ5[+]) were sorted on a BD FACS Aria Fusion into HBSS + 10% FBS prior to droplet encapsulation using 10x Genomics' Drop-seq platform (Chromium v2). 4 pooled samples of 4 mice each were generated (two HC pools, two CSD pools) and run on the same 10x Chromium chip, though 1 CSD sample was lost due to errors with droplet encapsulation; libraries were sequenced on Illumina NextSeq 550 and feature counts generated using Cellranger V2 pipeline. Single cell sequencing data have been deposited in GEO under accession code GSE301684.

We performed the following steps to obtain N = 6694 quality-controlled single cells: cell calling using DropletUtils::emptyDrops[85]; exclusion of outlier cells based on mitochondrial reads (<8.3% of total) or total features per cell (range for included cells = 174 – 4548); doublet removal using scrublet[86], with doublet rates in the three samples estimated as 6.9%, 5.1% and 3.4%. Samples were normalized using scran deconvolution-based normalization[85] highly-variable genes (3599 genes) were selected as genes with biological variation across samples > 0 (using scran::decomposeVar). Batch correction across the three 10X lanes was performed using batchelor::multiBatchNorm and fastMNN (default 50 components used for dimensionality reduction)[87]. Clustering of MNN-corrected PCA components across all single cells was performed using the leidenalg clustering algorithm[88]. Marker genes were detected by two methods (a) scran findMarkers function, based on differential expression between clusters and (b) soupX quickMarkers function, which uses Term Frequency - Inverse Document Frequency (TF-IDF) to identify the genes most predictive of a cluster, based on the frequency with which a gene is expressed in a cluster. Clusters were manually annotated by comparing marker genes expressed with existing single cell datasets. differential gene expression (DGE) between CSD and HC cells was performed as follows: counts were renormalized within the cluster; genes differentially expressed in pseudobulk of empty droplets (i.e., likely representing ambient RNA) were removed as described elsewhere[89]; genes expressed in ≤ 15% of cell in the cluster were removed; then gene expression in CSD vs. HC cells was compared using a Mann-Whitney U test, with Benjamini-Hochberg FDR correction of p-values across all tested genes. Pre-ranked gene set enrichment analysis was performed using clusterProfiler[90] with genes ranked by -log10 (Mann-Whitney U test P-value) * sign(LFC). Cell cycle stage of each cell was estimated using scran::cyclone[91]. Results were plotted using Seurat, bespoke code, or ktplots[92] and pathway enrichment was performed using [R]

clusterProfile[90] GSEA function with signed -log10(p-value) as the ranking statistic and GO biological pathways[93] accessed via msigdbr[94].

## Comparison with public neutrophil transcriptomic datasets

To compare our neutrophil transcriptional data to neutrophils from other tissues, we reprocessed two public mouse RNAseq datasets that each included neutrophils acquired from multiple tissues[28,30]. For the public Kolabas dataset (scRNAseq), we summed raw cell counts per tissue and cell type to create pseudobulk profiles for neutrophils and preneutrophils across multiple tissues, retaining pseudobulk profiles including at least 25 cells, and excluding two outlier samples. Variance stabilizing transformation (VST) implemented in the DESeq2 package[95] was applied to the pseudobulked counts. Samples included bone marrow from femur, humerus, pelvis, scapula, skull, and vertebra. Our meningeal neutrophil and pre-neutrophil data were pseudobulked and transformed in the same way. For the public Evrard dataset (bulk sorted cell RNAseq), we selected control samples representing mature, immature and preneutrophils from femur, plus mature neutrophils from blood, then applied DESeq2 VST. PCA was performed on the Kolabas dataset using the intersect between the 20% most highly variable genes in the Kolabas dataset and the genes present in all three datasets. VST matrices from the stress and Evrard datasets were quantile-normalized and projected into the Kolabas PCA space to enable direct comparison (Fig. S14).

## Pseudotime trajectory analysis

We generated cluster centroids and a minimum spanning tree for the neutrophil subclusters using scran createClusterMST. We obtained a pseudotime ordering by mapping cells to this tree using TSCAN mapCellsToEdges[96]. Scran testPseudotime was used to find genes significantly associated with pseudotime.

## Meningeal microarray

Meningeal tissue was dissected as described above, but with the following modifications: no digestion was used, intact tissue was passed through a 70 μm nylon mesh cell strainer using a 1 ml syringe plunger. Cells were then centrifuged and stored in Trizol Reagent (Cat #15596026, ThermoFisher) at -80 °C. For RNA extraction, samples were thawed and triturated using syringe needles before using a Qiagen miRNeasy Mini kit (Cat#217004). Labelled probes were run on the Affymetrix GeneChip™ Mouse Gene 2.0 ST Array (Cat#902118) according to the manufacturer's guidelines. Data were RMA-normalized and used to test for DEGs in CSD vs HC conditions, as described previously[20]. Microarray data have been deposited in GEO under accession code GSE275966.

## ROS analysis

Single cell suspensions were generated as above, and relative ROS production was analyzed using the CellROX™ assay (Cat #C10492, ThermoFisher) following the manufacturer's instructions. After ROS labeling, cells were stained for flow cytometry and promptly run unfixed on a flow analyzer.

## Cortical actin analysis

Single cell suspensions of blood and meningeal cells were generated and stained for flow cytometry as described above. Cortical actin was stained by adapting previously published methods[59]. Specifically, prepared cells were washed in PBS, then fixed in a modified "superfix" solution: 7.75 ml custom buffer (100 mM KCL, 3 mM MgCL₂, 10 mM HEPES, 150 mM sucrose, pH 7.4), 1 ml freshly made 37% formaldehyde solution (from powder, made in PBS), 1 ml DMSO, 200 μl of 100 mM EGTA, and 10 μl of 50% glutaraldehyde (in PBS). To this, Alexa Fluor™488 phalloidin (Cat #A12379, ThermoFisher) was added at a final concentration of 2 U/ml for filamentous (F)-actin visualization, and Hoechst-33342 (Cat #H3570, ThermoFisher) was added at a

dilution of 1:10,000 to label nuclei. Imaging flow cytometry analyses was performed on an ImageStream MarkII (Cytek Bio) 2-laser, 12-channel instrument. Spectral compensation was performed with single color beads and cells as described above. Samples were acquired at 60x magnification and low speed. Data were analyzed using IDEAS 6.02 software (Amnis Corp.) and FlowJo.

## Enzyme-linked immunosorbent assay (ELISA)

Blood plasma was analyzed using ELISA kits for murine CXCL1 (Cat #900-K127, PreproTech), CXCL2 (Cat #900-K152, PreproTech), and CXCL12 (Cat #DY460, R&D Systems) according to manufacturer's instructions. To compensate for sample plasma effects, assay standards were diluted in the CXCL12 RD6Q mouse standard diluent. In all assays, EIA/RIA high binding polystyrene flat bottom well plates (Cat # 2580, Costar) were used for the attachment of capture antibody. Sample absorbance was detected at 450 nm using a Victor 3 plate reader. Data analysis was performed using a four-parameter fit of resultant absorbances in Prism 9.0.

## Isolation of RNA/real-time quantitative PCR (RT-qPCR)

Samples used for RT-qPCR were resuspended and stored in Trizol at -80 °C until further purification with the Qiagen RNeasy kit (Cat #74104, Qiagen) according to the manufacturer's instructions. Total RNA was quantified and 1 μg was converted into cDNA using Superscript II reverse transcriptase (Cat #18064014, ThermoFisher) and Oligo dT primers (Cat #18418020, ThermoFisher). RT-qPCR was performed using the below listed primer sets and 2xSYBR Green Master Mix (Cat #172570, Bio-Rad) in a Bio-Rad MyIQ iCycler. The cycling program used an initial denaturation for 3 m at 95 °C, followed by 40 cycles of the following: 95 °C for 15 s, 58 °C for 30 s, 72 °C for 30 s. Levels of the targeted gene expression were determined by comparison of the sample Ct values to standard curves of Ct vs. dose of the amplicon of interest, with normalization to *TATA binding protein* (*TBP*). The amplicons of each primer set for any target gene were validated by sequencing prior to these experiments. Primer sequences and GenBank accession numbers are: *CXCL1* (NM_008176. Fwd: GCTGGGATTCACCTCAAGAA, Rev: TGGGGACACCTTTTAGCATC), *CXCL2* (NM_009140. Fwd: AAGTCATAGCCATCTCAAGGG, Rev: CTTCCGTTGAGGGACAGCAG), *CXCL12* (NM_021704. Fwd: CTGCATCAGTGACGGTAAAC, Rev: TCCACTTTAATTTCGGGTCA), *TBP* (NM_013684. Fwd: GACCCACCAGCAGTTCAGTA, Rev: AAACACGTGGATAGGGAAGG).

## Cytospin

Skulls were obtained and prepared for single-cell suspension. The cells were then fixed, pelleted onto thin coverslips using funnel centrifugation (Cat #10-354, Fisher HealthCare), and imaged using a Zeiss 780 confocal microscope fitted with 10x objective.

## IFNAR-blocking assay

Anti-IFNAR (clone: MAR1-5A3, Cat # BE0241) and non-specific, IgG isotype control (clone: MOPC-21, Cat # BE0083) antibodies for repeated in vivo injections were purchased from BioXCell and diluted in *InVivo*Pure pH 7.0 Dilution Buffer (Cat # IP0070) to a concentration of 5 mg/mL. Mice were i.p. injected with 1 mg of antibody on day zero (d0), before the start of the defeat paradigm, and received 0.5 mg of antibody every third day thereafter, for a total of 5 injections per mouse. Defeats were done approximately 1-3 h after injections. LysM^+/gfp mice were ear-tagged and randomly assigned to one of three groups: HC+IgG, CSD+IgG, or CSD + IFNAR. WT mice were ear-tagged and randomly assigned to one of four groups: HC+IgG, HC + IFNAR, CSD+IgG, or CSD + IFNAR.

## Systemic neutrophil depletion

To further explore the mechanistic role of neutrophils in the response to chronic social defeat (CSD) stress, we first performed neutrophil

depletion experiments using methods optimized for prolonged treatment[97]. Briefly, 8 mice were randomly assigned to either the HC or CSD group. 2 mice from each group were then randomly assigned again to either treatment (Ly6G antibody depletion) or control (IgG antibody) conditions, comprising 4 groups in total: HC+IgG, HC+Ly6G, CSD+IgG, CSD+Ly6G ($n = 2$ per group). Primary (1º) rat anti-mouse Ly6G antibody (clone: 1A8, Cat # BE0075, Bio X Cell) and 50 µg secondary (2º) anti-rat antibody (clone: MAR18.5, Cat # BE0122) were administered on alternating days for the treatment group, and an equivalent amount of IgG antibody (clone: 2A3, Cat # BE0089) was administered to control condition mice, according to the timeline shown in Fig. S26A. In this pilot we noted (but did not quantify) ear tag holes in both HC and CSD groups that became enlarged over time, indicating possible wound healing deficits. Using our rubric for assessing fight-related wounds in CSD animals (Table S12), we determined that neutrophil depletion significantly impaired wound healing of the typically minor injuries sustained during the CSD protocol (Fig. S26B). We also noted massive splenomegaly (>300 mg) in two Ly6G-treated mice (one HC, one CSD). Due to concerns about animal welfare, we ceased using this method for CSD experiments.

We pursued a modified neutrophil depletion protocol with shortened timeline for defeat exposure (Fig. S26C) and reduced dosage of antibody compared to that typically used[98–102]–i.e., 100 µg compared to 200 µg. 14 mice were randomly assigned to either CSD+saline (n = 6) or CSD+Ly6G conditions (n = 8). After 10 days of defeat, wound scores were similar between groups, but massive splenomegaly was again observed in the neutrophil-depleted animals (Fig. S26D, E). Discussion of these data with animal care staff led us to cease pursuit of these studies due to a combination of animal welfare concerns and uncertainty about experimental confounding, as any immunological or behavioral differences observed could have been driven by gross immune abnormalities related to acute splenomegaly rather than effects on meningeal neutrophils in neutrophil-depleted CSD mice.

## Statistics

Prism 9.0.2 and 10.0 (GraphPad Software, LLC) was used for statistical testing and graphing of univariate analyses. Normality and equal variance were assessed, and appropriate tests were conducted thereafter. For simple, two-group comparisons, two-tailed tests were used. Samples with non-equal variance were analyzed using a Mann-Whitney U test, whereas samples with equal variance were analyzed using a Student's t test. In almost all instances, CSD-stressed mice failed to mark in the USM test; the behavioral output was thus better modeled statistically as a binary response (did or did not mark). We therefore used $\chi^2$ tests for univariate analyses of group in the USM test; posthoc testing was performed with the chisq.posthoc.test package in R, using the Benjamini-Hochberg method for multiple comparisons corrections. For multiple comparisons of normally distributed data, ordinary one- or two-way ANOVAs were used for analysis. When analyzing complete data from multiple tissues within the same mice, repeated measures with an interaction term were selected for the model; for data with missing values, Restricted Maximum Likelihood (REML) mixed-effects models were used. When a main effect was present, post-hoc analyses were conducted (Dunnett's, Tukey's, or Šídák, respectively). For multiple comparisons of non-parametric data, a Kruskal-Wallis test was used, and Dunn's test for multiple corrections was run for post-hoc analysis. Univariate comparisons are summarized as the mean ± SEM and considered statistically significant at $p < 0.05$. No power calculations were performed at the outset of the experiment.

## Additional statistics

Prior to debatching and for univariate analyses of meningeal histology data, predictor variables were assessed for outliers (using Grubb's test for outliers), skew, kurtosis, normality (using the Shapiro-Wilks test), and equal variance (using Bartlett's test) in R version 4.3.0[103].

Increasingly strict transformations were applied to individual variables until normal distributions and assumptions of equal variance were met (in order from least stringent to most stringent transformations: square root, $\log_{10}$, natural log). Ordinary least squares (OLS) regression was used to assess relationships between predictor variables (meningeal neutrophils, circulating white blood cell populations) and behavioral outcomes. Linear models were constructed in R with the following formula: [dependent variable] ~ [continuous predictor] + batch. Model fit was assessed via quantile-quantile plots and by examination of residual distributions. For multivariate analysis of USM, a generalized linear model was constructed with binomial family selection according to the formula USM_binary ~ [continuous predictor] + batch. Model fit was assessed via quantile-quantile plot. Coefficient estimates of the models were plotted using dot-whisker::dwplot in R[104]. The mixed linear regressions were treated as exploratory; thus, unadjusted $p$ values are presented in the Figures. However, For mixed linear regressions, Benjamini-Hochberg (FDR) corrections were performed to control the type I error rate; unadjusted p values are presented in Tables S1–S4.

Linear regression was used to assess wounding as a predictor for immune cell populations according to the formula [cell population] ~ [wound score]. To assess relationships between immune cells in different tissues, a Pearson correlation matrix was first generated with the function Hmisc::rcorr. Unsupervised hierarchical clustering was then performed using the Lance-Williams dissimilarity formula ("hclust" in R base stats) and the gplots::heatmap.2 function for plotting. The unsupervised clustering order for neutrophils was used to force clustering of monocytes and lymphocytes. Bonferroni corrections were applied to control the type I error rate; significant surviving associations were added to the heatmap manually, indicated with an asterisk.

## Reporting summary

Further information on research design is available in the Nature Portfolio Reporting Summary linked to this article.

## Data availability

Single cell sequencing data generated in this study have been deposited in the GEO database under accession code GSE301684. Publicly available single cell sequencing data analyzed in this study are available online under accession codes GSE192616 and GSE109467. Microarray data analyzed in this study have been deposited in the GEO database under accession code GSE275966. Other data generated in this study are provided in the Source Data and Supplementary Source Data files provided with this paper. Source data are provided with this paper.

## Code availability

All code used for analysis is available at https://doi.org/10.5281/zenodo.15774364.

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

## Acknowledgements

We thank Dorian McGavern for helpful commentary on early versions of the manuscript, for advice on studies, and for kindly providing LysM$^{gfp/gfp}$ mice. We thank Monica Manglani, Hannah Mason, Methma Udawatta, Donghyun Kim, and Chris Higham for technical assistance. We thank staff in the National Cancer Institute (NCI) Laboratory of Genome Integrity (LGI) Flow Cytometry Core, staff in the Microarrays and Single-Cell Genomics Core (National Human Genome Research Institute), and veterinary care staff at the National Institute of Mental Health for their help with data collection. This research was supported in part by the Intramural Research Program of the National Institutes of Health (NIH), award ZIA MH001090 (S.L.K., A.E.D., R.A., V.H.S., J.D.S., N.E.E., C.N.P., M.L.L., S.J.L., M.H.). The contributions of the NIH author(s) were made as part of their official duties as NIH federal employees, are in compliance with agency policy requirements, and are considered Works of the United States Government. The authors received additional support from the National Institute for Health and Care Research (NIHR) Cambridge Biomedical Research Centre (BRC), award NIHR203312 (S.L.K.); the UK Medical Research Council, award MR/S006257/1 (M.E.L.), the UKRI Mental Health Platform ImmunoMIND hub, award MR/Z50354X/1 (M.E.L.), NIH T32GM113896 (N.E.E.), NIH T32GM14532 (N.E.E.), NIH T32AG082661 (N.E.E.), and NIH T32NS082145 (N.E.E.). The NCI LGI Flow Cytometry Core is supported by funds from the Center for Cancer Research. All research at the Department of Psychiatry in the University of Cambridge is supported by the NIHR Cambridge BRC (NIHR203312) and the NIHR Applied Research Collaboration East of England. The findings and conclusions presented in this paper are those of the author(s) and do not necessarily reflect the views of the NIH, the U.S. Department of Health and Human Services, the NIHR, or the Department of Health and Social Care.

## Author contributions

S.L.K., A.E.D., R.A., J.D.S., N.E.E., C.N.P., M.L.L., S.J.L., F.L., A.G.E. participated in data collection. S.L.K., M.E.L., V.H.S., F.L., A.G.E. performed analyses. S.L.K. and M.E.L. conceived the study and wrote the manuscript. M.H., M.R.C., and E.T.B. provided supervision. M.R.C. contributed results the interpretation. A.E.D., R.A., J.D.S., V.H.S., C.N.P., M.L.L., S.J.L., F.L., M.R.C., E.T.B., M.H. reviewed and edited the manuscript.

## Competing interests

The authors declare the following competing interests: E.T.B. has recently consulted Boehringer Ingelheim, SR One, Novartis, GlaxoSmithKline, Sosei Heptares, and Monument Therapeutics. E.T.B. holds equity in and is a cofounder of Centile Bioscience Inc. M.L.L. is currently employed at AstraZeneca but was an employee at NIH at the time this work was conducted. The remaining authors declare no competing interests.
