## [Transparent Peer Review file · Nature Communications]

Chronic social defeat stress induces meningeal neutrophilia via type I interferon signaling in males

Corresponding Author: Dr Stacey Kigar

Version 0:

Reviewer comments:

Reviewer #1

(Remarks to the Author)

I enjoyed reading this interesting and novel manuscript which describes the migration of IFNAR+ neutrophils from the skull into meninges of mice exposed to chronic stress. The manuscript is clear and potentially important. The evidence for skull channel migration is somewhat circumstantial, but increasing the experimental proof is quite difficult and perhaps not necessary. The beneficial effects of IFNAR antibodies are very helpful but strictly speaking we do not know if this is neutrophil related. Overall the manuscript may be slightly premature. I did not spot any major other points of criticism, and given the high novelty of the work, perhaps the criticism above should not be weighed too much.

(Remarks on code availability)

Reviewer #2

(Remarks to the Author)

Kigar et al. evaluated the impact of social stress exposure on immune response in the meninges. The authors used the established chronic social defeat (CSD) stress paradigm to induce anxiety and anhedonia in mice and then evaluated immune changes with a particular focus on blood and meningeal neutrophils. Rationale is supported by previous work from this group and others indicating that blood neutrophil levels are elevated after acute stress exposure and in individuals with major depression. Here, various techniques such as flow cytometry, immunohistochemistry, single-cell sequencing and tissue clearing were elegantly combined to demonstrate that chronic social stress induces meningeal neutrophil accumulation and increases neutrophil trafficking through meningeal vascular channels. Transcriptional findings and drug rescue support the involvement of the IFN-I pathway and its receptor in the regulation of neutrophil recruitment to the meninges providing important mechanistic insights.

Overall, the topic of this study is novel, and the results reported interesting for scientists of various fields. Identification of the skull bone marrow as the source of stress-related meningeal neutrophil accumulation is very intriguing. I do believe that additional behavioral experiments are necessary to further characterize the transgenic mice, and IFNAR drug-delivery rescue. Some data discrepancies also deserve further explanations, but I am convinced that this team has the capacity to address these major points. Additional detailed comments and suggestions for consideration are listed below.

Major:

- The timeline of stress-induced neutrophil level changes (Fig.1g-h) is critical and should go beyond day 14 with a difference observed only at this time point for meningeal neutrophils the core focus of this manuscript. I would recommend adding at least two time points to confirm that the alteration is long lasting. In the same line, are neutrophil levels stable in control mice across time?
- How do the authors reconcile the results from LysM/gfp+ mice with intense staining in the brain (Fig.2g) vs no infiltration of neutrophils in the parenchyma with immunohistochemistry in regions linked to mood – medial prefrontal cortex, striatum, hippocampus (Fig.S4)?
- Other behavioral tests should be performed to confirm the behavioral phenotype of stressed LysM/gfp+ mice (Fig.3). Social interaction and sucrose preference tests would validate core features associated with a depressive phenotype. Number of crosses in the open field center is not sufficient to confirm anxiety and additional graphs (time in open vs closed arms,

number of entries, locomotion, etc.) should be provided in a supplementary Figure. Addition of an elevated plus maze test to confirm anxiety-like behaviors would further strengthen the mouse strain behavioral characterization following social stress exposure.

- Similar comment for the drug-delivery rescue (Fig.6) with IFNAR blocking antibodies. A full panel of behavioral tests (social interactions, forced swim, tail suspension, sucrose preference, etc.) could reveal domains with differential responses strengthening the study and impact of this work. It may also explain the non-normal distribution after treatment.
- A discussion on sex differences and data interpretation of the current findings beyond limitations would strengthen the manuscript with increasing evidence highlighting sex-specific mechanisms for brain barrier and immune function in the context of social stress responses.

Moderate:

- Sex of the mice (males only) should be mentioned in the abstract.
- It is not always clear in the introduction when the authors refer to preclinical or clinical studies.
- The CSD paradigm is unusual and additional details should be provided. Is the CD-1 aggressor always the same throughout the 1-14 consecutive days? Could aggression level affect immune cell level in the blood and meninges (as briefly mentioned in the limitation part for wounding)? On Fig.1 most of the stressed mice display % of meningeal neutrophils similar to the control group. Are those mice the ones without wounds?
- I wonder if it would be possible to perform a distribution analysis for brain areas involved in emotion regulation for the brain clearing-related data. The stress-induced increase in neutrophil accumulation seems to be evenly distributed (Fig.2F) how do the authors explain this vs human condition?
- Images should be provided for the drug-delivery rescue experiment (Fig.6).

Minor:

- It might be relevant to define neutrophilia in the abstract to reach a wider audience.
- Fig.5 the abbreviation neutrs has never been used before, neutrophils may be more appropriated.
- I would refrain from using strong words such as robustly.

(Remarks on code availability)

Reviewer #3

(Remarks to the Author)

(Remarks on code availability)

Reviewer #4

(Remarks to the Author)

In the paper by Kigar and colleague entitled "Chronic social defeat stress induces meningeal neutrophilia via type I interferon signaling", the authors describe and phenotype the accumulation of neutrophils in the meninges upon chronic social defeat using a combination of flow Cytometry, histology and single cell sequencing. They also implicate type I interferon in the neutrophilic phenomenon. The topic of research is of great interest given the regained interest in understanding how the immune cells of the brain borders can impact physiological and pathological brain function. The authors also use complementary approaches to convincingly demonstrate the infiltration of neutrophils in the meninges after chronic social defeat. Some conclusions put forward from the authors however are not substantiated by the presented data and would require additional experiment or alteration in interpretation:

Major points:

- The source of meningeal neutrophils: The authors use the measurement of the number of neutrophils in the skull bone marrow channel (Figure 2g-j) to claim that they come from the skull bone marrow, a statement that is repeated throughout the manuscript. The sole use of that data, however, does not exclude a potential large contribution of the more classical blood extravasation signaling. Particularly, the large accumulation of the meningeal neutrophils in or around the meningeal blood vasculature, and in the brain vasculature strongly suggest a potential consequential contribution of the blood as a source of meningeal neutrophils. Could the other validate the contribution from one versus the other via tagging of the skull bone marrow or blocking blood extravasation signaling?
- Neutrophils and type I interferon: The authors are using systemic blockade of Type I interferon, which shows reduction in behavioral effects of chronic social stress, and associate it with neutrophils recruitment in the meninges as a mechanism. Previous literature demonstrated that removal of IFNAR from microglia was sufficient to alleviate behavioral symptoms of chronic mild stress. While this reviewer understand that generation of cell specific KO animals may be outside of the scope of the current manuscript, the authors should, at least, demonstrated that depletion of meningeal neutrophils has an effect on chronic social stress associated behaviors. Does the single cell RNA seq provide any information as to why neutrophils

presence in the meninges may alter chronic social stress associated behavior? The authors are focusing on the migration related pathway in their analysis, but it feels like there is a missed opportunity to better understand what could be the functional consequences of meningeal neutrophilia. The authors show that blockade of Type I interferon normalize number of meningeal neutrophils. Have the authors looked if that decreased the migration of neutrophils through the skull bone marrow channel ?

- Chemorepellent factors: In figure 7C, the authors extract from microarray data the down regulation of Sema3b in the meninges and use that to conclude that chemorepellent are decreased which would increase neutrophils infiltration. These data comes out of nowhere in the progression of the paper. Either the authors should use that data, and investigate deeper, the circulation of neutrophils from the blood and the skull bone marrow, or that data should be removed from the manuscript.

Minor points:

- Throughout the manuscript, the authors are using the term parenchyma to talk about neutrophils that are "away" from blood vessels in the meninges. That nomenclature can be misleading as parenchyma usually refers to the inside of the brain for most neuroscientist. An change of nomenclature or a clearer definition of what the authors are referring to would help the readers appreciate the message that the authors are putting forward.

- It is unclear from the manuscript what considered a mouse to have "marked" or not. In some experiment (Type I interferon blockade), the mice are split depending on if they marked or not, while other experiments/group, the authors didn't. There would need to be some consistency to assess if the difference are solely associated with the "marked" status rather than the treatment itself. It is particularly visible in the Type I interferon blockade experiment where the treated mice almost completely segregate depending on their "marked" status. What do the authors hypothesis to have such a strong bimodal response in the treated group?

- In the introduction, the authors particularly emphasize on their previous work regarding meningeal B cells, and B cells are used throughout the manuscript as a control for their experiment. The authors also suggest a connection between B cells and neutrophils. However, no experiments are made to address this specifically. Refocusing the writing of the manuscript towards neutrophils may help straighten the message the authors are putting forward.

(Remarks on code availability)

Reviewer #5

(Remarks to the Author)

Animal models of stress and related disorders show increased blood neutrophilia, though its connection to symptoms or behavior is unclear. Using various techniques, the authors found that chronic, but not acute, social defeat (CSD) stress in mice causes neutrophil accumulation in the meninges, with potential increased trafficking from the skull bone marrow. Blocking type I interferon (IFN-I) signaling systemically protected against the behavioral effects of CSD, suggesting IFN-I as a potential therapeutic target for stress-related disorders.

The study is very interesting and deserves publication, but one experiment would be really interesting to confirm the causal role of neut in this model using neutrophil depletion. Also, here are some minor comments that would improve the manuscript.

Fig1. Authors should indicate the number of cells, not only the proportion, at least for iv- neutrophils at day 14 in HC versus CSD, to make sure that this population increases (as there is a relative decrease of other populations)

Fig1 could be summarized by fig 1f and 1g, and the rest could be supplemental data. Is timepoint 0 the HC? Is the loss of B cells significant? And the increase in monocytes? Stats should be done for each pop in the main figure, not only in supp data. Is the changes in iv+ B cells and neutrophils a reflection of what is in the blood? If so that would mean that there is a strong changes in bone marrow cells, that could be really interesting to analyze (femur versus skull).

Fig2defij: the graphs should display lysM+ cells, not neutrophils, as LysM is positive for neuts, mono, macs. How did they avoid including mono/meningeal macrophages, which are also LysM+?

Fig3: can the authors represent counts of neutrophils instead? Can the authors graph the time spent with female odor and time spent in OF in control and CSD mice, and do the stats of those 2 groups? This would assess more clearly whether CSD had an effect as expected. Then for the CSD group only, they should graph correlation between meningeal neuts, meningeal mono and meningeal B cells (and same for blood cells) as Y parameters and time spent (odor then OFT) as X parameter, and check correlation p value. Also, timepoint of test and timepoint of neuts % should be clearly indicated. The data on neuts and anxiety is also very interesting based on the literature between neuts and depression in humans.

Fig4c: adding ID number on the plot would help (as some colors are quite close). Where are the pre-neutrophils? What it the GO of up and down regulated genes (this should be in the main data, not only supp)? What about DEG in other clusters such as mono and B cells? Do we see difference in cell number too for mono and B cells? Neutrophils do not usually express MHC2, so it's surprising that they downregulate it.. was this checked by flow?

Fig5e: abcdef should be explained in the figure

Fig5d: could the huge difference in IFNAR+ cells in blood versus skull be a bias of the digestion in the skull? They would need controls (eg blood cells processed with digestion). They should quantify CD69 or IFITM in addition, as those are ISGs. And check levels of iv+ and iv- IFNAR+ neuts to make sure it's not a bias of preparation.

Fig6c: it seems not correct to stratify the CSD IFNAR group (+/-) as the HC are not stratified although they display similar

heterogeneity. The authors should do the stats of the overall CSD IFNAR group instead first. What is the source of IFNs? Do the authors have similar results on odor and OFT tests (ie restoration of normal behavior) upon neutrophil depletion (see main point above)?

Additional questions:

Is there BBB leakage (apparently yes based on previous studies from the team)? What attracts neutrophils in the meninges? IFNs? What about male versus female behavior?

'meningeal neutrophilia' is confusing, as neutrophilia refers to the blood.

If 'pathogenic' neutrophils come from the marrow, what do they author think the role/consequence/origin of blood neutrophils are? (cf 1st sentence of abstract) Also, directionality is not proven and accumulation of neutrophils derived from the vasculature could also traffick upwards to the skull.

What is the phenotype of neutrophils in the meninges (IFNAR+?) in HC and CSD mice?

What is the consequence of IFNAR blockade on blood neutrophils?

What could be the differential role of meningeal neutrophils versus 'stuck' neurovascular neutrophils?

The data on CXCL1 and CXCL12 should be indicated as highly hypothetical as no functional/blockade experiments have been performed.

Data on blood cells proportion upon IFNAR treatment should be indicated in the main text, as it's a nice way to decorrelate blood counts and restoration of cognitive functions.

Is IFN alone enough to shut down semaphoring expression?

The authors should only mention 'potential' migration from the bone marrow as now experiments is shown to prove it (parabiosis, skull graft, etc.)

(Remarks on code availability)

Version 1:

Reviewer comments:

Reviewer #1

(Remarks to the Author)

I find the revisions acceptable.

(Remarks on code availability)

NA

Reviewer #2

(Remarks to the Author)

The authors addressed all my criticisms and suggestions in a satisfactory manner. I can now recommend publication and congratulate them for this important work.

(Remarks on code availability)

Reviewer #3

(Remarks to the Author)

(Remarks on code availability)

Reviewer #4

(Remarks to the Author)

In the revised manuscript, the authors have addressed some of the comments from the prior submission. However some comments have not been addressed to the levels that it answers my concern.

- The specific effect of IFNAR1 on neutrophils migration to improve social defeat. The authors justified in their rebuttal that the anti-IFNAR1 antibody they use does not cross the blood brain barrier and that they can't measure neutrophils migration through the skull after IFNAR1 blockade. Regarding the first part of their answer, the manuscript they cite to claim that the

antibody does not cross the BBB state "MAR1-5A3 is not expected to cross the blood-brain-barrier". Without further evidence of such, it's a rather thin argument to exclude potential other effect of the IFNARI blockade, particularly on other cells like microglia and macrophages (shown previously to affect behavior). It is unclear with LysM GFP+ mice would not be able to be treated with anti-IFNARI to assess the skull channel quantification as presented in figure 3?

- The role of neutrophils in social defeat. Multiple reviewer highlighted that demonstrating the central role of neutrophils accumulation being a causal factor in social defeat would be important, even more so given the potential other effects of the anti-IFNARI treatment. The authors state that their attempt had detrimental effects. It is however unclear if the effects were limited to the stressed mice or the depletion itself. Boivin et al, (Nat Comm, 2020) demonstrated stable and sustainable depletion of neutrophils for up to 20 days. If the depletion of neutrophils is somehow detrimental in the CSD model, it would still be worth including in the manuscript for the community to be aware of.

(Remarks on code availability)

Reviewer #5

(Remarks to the Author)

The authors have addressed my concerns and have provided an exhaustive document, with associated experiments. The manuscript is suited for publication.

(Remarks on code availability)

N/A

Version 2:

Reviewer comments:

Reviewer #4

(Remarks to the Author)

First I would like to thank the authors for addressing my comments. With the addition of the new data, I have no further comments.

(Remarks on code availability)

Reviewer #1:

1/1) I enjoyed reading this interesting and novel manuscript which describes the migration of IFNAR+ neutrophils from the skull into meninges of mice exposed to chronic stress. The manuscript is clear and potentially important.

We thank the reviewer for their supportive remarks about the importance, novelty, and clarity of the manuscript.

1/2) The evidence for skull channel migration is somewhat circumstantial, but increasing the experimental proof is quite difficult and perhaps not necessary.

We agree that the skull trafficking data could be strengthened so we have reported two new lines of evidence in the revised paper.

First, we generated a correlation matrix for neutrophil levels (identified by flow cytometry) across several different tissues collected from individual C57/6J WT mice. Using unsupervised, hierarchical clustering of the data, we found that nonvascular—i.e., neutrophils negative for the presence of an intravascularly injected label, or iv⁻—meningeal neutrophil levels were significantly correlated with bone marrow neutrophil levels. Conversely, meningeal neutrophils positive for the intravascularly-injected dye (iv⁺) were significantly correlated with blood and spleen neutrophil levels.

This clustering pattern was distinct from monocytes, which derive from the same progenitor bone marrow cells as neutrophils, are also GFP⁺ in LysM mice, and show CSD stress-related increases across tissues. New sections have been added to the figures (**Figures 3J-K, S11A**), figure legends, methods, results, and discussion as follows, and accompanying code provided in the paper github repository:

Figure 3 ... “. j) Flow cytometry shows widespread increase in neutrophil levels throughout the body following CSD stress in C57BL/6J mice ($n_{HC} = 7-8$, $n_{CSD} = 9$). k) Unsupervised hierarchical clustering of data from (j) shows significant correlations between BM and iv⁻ meningeal neutrophils, but not with blood...”

“Figure S11: No evidence to support skull trafficking of monocytes... in C57BL/6J mice. **a)** Flow cytometry shows a main effect of stress on monocyte levels throughout the body, though only the spleen was significant with multiple comparisons corrections (main effect of group, $***p < 0.001$, $F_{(1,15)} = 24.7$. Šídák’s post hoc: *spleen iv⁺*, $*p < 0.05$, $t = 4.1$). Supervised hierarchical clustering, based on neutrophil clustering pattern, shows *iv⁺* meningeal, blood, and spleen monocytes significantly correlate with each other, but *iv⁻* meningeal and bone marrow monocyte levels do not. This is in contrast to the pattern seen in neutrophils, shown in **Figure 3K**...”

Results:

CSD increases neutrophil trafficking between skull BM and meninges.

“...We next tested the extent to which elevated meningeal neutrophil numbers reflect neutrophil levels in peripheral tissues, which were also increased by CSD stress in C57BL/6J mice (**Figure 3J**: $****p < 0.0001$, $F(1,78) = 62.7$. post hoc, $t = 7.9$, $****p < 0.0001$). Pearson correlations and unsupervised hierarchical clustering across tissues revealed that *iv⁻* meningeal neutrophil levels were most similar to skull ($***FDR < 0.001$, $r = 0.83$, $t(15) = 5.81$) and tibial ($**FDR < 0.01$, $r = 0.75$, $t(15) = 4.36$) BM neutrophils, and showed no relationship with blood neutrophils (**Figure 3K**). In contrast, *iv⁺* meningeal neutrophils correlated with blood neutrophils ($*FDR < 0.05$, $r = 0.72$, $t(15) = 3.97$).

We also examined relationships between *iv⁻* meningeal monocyte and lymphocyte levels with the corresponding cells in other tissues but did not find any significant relationship (**Figure S11**). These data support a role for skull-to-meninges communication in CSD, and highlight the unique composition of the meningeal immune environment compared to other tissues.”

Supplemental methods:

Additional statistics. ... “To assess relationships between immune cells in different tissues, a Pearson correlation matrix was first generated with the

function `Hmisc::rcorr`. Unsupervised hierarchical clustering was then performed using the Lance-Williams dissimilarity formula (“`hclust`” in R base stats) and the `gplots::heatmap.2` function for plotting. The unsupervised clustering order for neutrophils was used to force clustering of monocytes and lymphocytes. Bonferroni corrections were applied to control the type I error rate; significant surviving associations were added to the heatmap manually, indicated with an asterisk.”

Second, we compared our data to publicly available single-cell sequencing data from neutrophils in skull, blood, and other bone marrow locations (Evrard 2018, Kolabas 2023). This analysis showed that meningeal neutrophils are transcriptionally distinct from blood and share similarities with bone marrow. New sections have been added to the figures (**Figure S14**), figure legends, methods, results, and discussion as follows, and accompanying code provided in the paper github repository:

Figure S14 Principal Component Analysis (PCA) plots visualize the transcriptomic similarity of neutrophil and preneutrophil populations across tissues (blood, meninges and various bone marrow locations), cell types (neutrophils and preneutrophils), condition (HC, control and CSD, stress), and dataset. We integrated our meningeal dataset with two publicly available mouse RNA-seq datasets (Kolabas et al. and Evrard et al.) containing neutrophils and pre-neutrophils from blood and multiple bone marrow sites. Variance stabilising transform was applied to the bulk (Evrard) and pseudobulked (our meningeal data and Kolabas) samples (see **Supplementary Methods**). PCA was performed on Kolabas pseudobulk samples using the 20% most variable genes intersected with genes present in all three datasets. VST matrices from our data and Evrard's were quantile-normalized and projected into this PCA space. (a) PCA plot highlighting tissue origin (predominantly PC2) (b) PCA plot highlighting cell types (reflected by PC1). Meningeal pre-neutrophils cluster with pre-neutrophils from Kolabas and Evrard samples but meningeal neutrophils are most similar to bone marrow immature neutrophils. HC, control (white outlines), SD, social defeat (black outlines).

Results:

“Single cell RNA sequencing (scRNAseq) reveals increased neutrophils in CSD meninges, increased proinflammatory signaling, neutrophil heterogeneity.

“...We integrated our data with two publicly available multi-tissue datasets containing pre-neutrophils and neutrophils from blood and multiple BM locations^{29,31}. When projected onto a shared PCA space, our meningeal neutrophils did not cluster with mature blood neutrophils but instead were most similar to immature BM neutrophils. In contrast, our meningeal pre-neutrophils clustered with the bulk and pseudobulked pre-neutrophil Evrard and Kolabas samples, suggesting successful integration of these three datasets (**Figure S14**).”

Supplementary methods:

“Comparison with public neutrophil transcriptomic datasets. To compare our neutrophil transcriptional data to neutrophils from other tissues, we reprocessed two public mouse RNAseq datasets that each included neutrophils acquired from multiple tissues (Evrard et al., 2018; Kolabas et al., 2023). For the public Kolabas dataset (scRNAseq), we summed raw cell counts per tissue and cell type to create pseudobulk profiles for neutrophils and preneutrophils across multiple tissues, retaining pseudobulk profiles including at least 25 cells, and excluding two outlier samples. Variance stabilizing transformation (VST) implemented in the DESeq2 package (Love et al., 2014) was applied to the pseudobulked counts. Samples included bone marrow from femur, humerus, pelvis, scapula, skull, and vertebra. Our meningeal neutrophil and pre-neutrophil data were pseudobulked and transformed in the same way. For the public Evrard dataset (bulk sorted cell

RNAseq), we selected control samples representing mature, immature and preneutrophils from femur, plus mature neutrophils from blood, then applied DESeq2 VST. PCA was performed on the Kolabas dataset using the intersect between the 20% most highly variable genes in the Kolabas dataset and the genes present in all three datasets. VST matrices from the stress and Evrad datasets were quantile-normalized and projected into the Kolabas PCA space to enable direct comparison (**Figure S14**)."

Both of these new lines of evidence have been included in an updated version of the discussion:

Discussion:

"...Our data lend further support to the extant literature suggesting direct trafficking of immune cells from the skull BM to the meninges^{26–29}. To our knowledge, this is the first evidence for such a phenomenon under conditions of psychosocial stress.

Several independent lines of evidence from our study bolster support for this conclusion. First, in cleared skull tissue, we observed more LysM-GFP+ myeloid cells present in skull-to-meninges vascular channels of CSD mice. While LysM-GFP expression is not restricted to neutrophils, unsupervised hierarchical clustering of flow cytometry data using the neutrophil-specific Ly6G antibody revealed that nonvascular meningeal neutrophil levels closely mirrored those of skull BM, but not blood. Further, integration with public datasets^{29,31} showed that meningeal neutrophils have a transcriptional profile distinct from mature neutrophils in blood, and are more similar to immature BM neutrophils."

1/3) The beneficial effects of IFNAR antibodies are very helpful but strictly speaking we do not know if this is neutrophil related.

We agree with the reviewer that this novel work has generated important future directions for testing and that more work must be done to establish conclusively that IFNAR antibody treatment works specifically on neutrophils. We have modified the limitations section to bring forward these points :

"Other studies have highlighted IFN-I responses in brain cells during stress^{79,80}. Given that the MAR1-5A3 IFNAR antibody clone we used cannot cross the blood brain barrier (BBB)⁸¹, the mediating effects of anti-IFNAR treatment on the negative sequelae of CSD stress in our model likely do not involve cells within the brain parenchyma. The cellular source of stress-related IFN-I, and the relative contributions to behavior of IFNAR signaling in different cell types, requires further investigation. "

1/4) Overall the manuscript may be slightly premature.

We respectfully disagree with the reviewer that the manuscript is “premature’. While we recognize that the paper is substantively and technically novel, as the reviewer also recognised in 1/5, we consider that our findings, which have been extensively cross-checked and validated, are of sufficient general interest and robustness to merit publication in their revised form. We note that in our response to other comments by this reviewer, and three other reviewers, we have now reported substantial additional sensitivity analyses and other data to further consolidate our principal results.

1/5) I did not spot any major other points of criticism, and given the high novelty of the work, perhaps the criticism above should not be weighed too much.

Again, we thank the reviewer for their supportive comments, especially concerning the “high novelty of the work”.

Reviewer #2:

Kigar et al. evaluated the impact of social stress exposure on immune response in the meninges. The authors used the established chronic social defeat (CSD) stress paradigm to induce anxiety and anhedonia in mice and then evaluated immune changes with a particular focus on blood and meningeal neutrophils. Rationale is supported by previous work from this group and others indicating that blood neutrophil levels are elevated after acute stress exposure and in individuals with major depression. Here, various techniques such as flow cytometry, immunohistochemistry, single-cell sequencing and tissue clearing were elegantly combined to demonstrate that chronic social stress induces meningeal neutrophil accumulation and increases neutrophil trafficking through meningeal vascular channels. Transcriptional findings and drug rescue support the involvement of the IFN- γ pathway and its receptor in the regulation of neutrophil recruitment to the meninges providing important mechanistic insights.

2/1) Overall, the topic of this study is novel, and the results reported interesting for scientists of various fields. Identification of the skull bone marrow as the source of stress-related meningeal neutrophil accumulation is very intriguing. I do believe that additional behavioral experiments are necessary to further characterize the transgenic mice, and IFNAR drug-delivery rescue. Some data discrepancies also deserve further explanations, but I am convinced that this team has the capacity to address these major points. Additional detailed comments and suggestions for consideration are listed below.

We thank the reviewer for their supportive remarks on both the study design and the novelty of the work. We summarise our responses to their specific concerns below.

2/2) The timeline of stress-induced neutrophil level changes (Fig.1g-h) is critical and should go beyond day 14 with a difference observed only at this time point for meningeal neutrophils the core focus of this manuscript. I would recommend adding at least two time points to confirm that the alteration is long lasting.

While we agree it would be interesting in principle to explore further the stability of the phenotype, extending CSD beyond 14 days would, in our opinion, result in an unacceptably high risk for wounding that confounds interpretation of the observed immunological changes. We pick this idea back up in response to the reviewer's comments in point 2/9, below, and have made the following change to the methods section:

"Chronic social defeat: ... We did not go beyond 14 days of defeat out of concern for animal welfare."

However, to illustrate the dynamic nature of the meningeal environment to stress, we have performed substantial new analyses on recovery from CSD on the basis of existing experimental data. The key finding from our new analysis is that meningeal neutrophil levels remain elevated for at least 24 h after CSD, returning to control (home cage = HC) levels within a week of recovery. In contrast, blood neutrophil levels return to control levels within 8h of stress. Meningeal B cell depletion persists for up to a week following recovery from CSD. New figures have been added (Figures 2, S6, S7), with accompanying figure legends, methods, results, and discussion as follows:

“Figure 2: Chronic, but not acute, social defeat stress elevates meningeal neutrophils; prolonged elevation of meningeal neutrophils relative to blood in recovery. **a) Left:** Schematic of acute vs chronic stress study; red arrows indicate time points in days at which mice were killed and tissue was harvested. **Right:** There was a main effect of the number of encounters for meningeal neutrophils, but only the CSD day-14 group showed a significant increase by post hoc analysis. In contrast, there was a significant increase in

blood neutrophils overall and at each time point when compared to HC (subscript indicates days of defeat: $n_{HC} = 8$, $n_1 = 7$; $n_2 = 7$; $n_4 = 9$; $n_8 = 4$; $n_{14} = 10-11$). **b) Left:** Schematic of post-CSD recovery study; red arrows indicate time points in hours (blue ticks) or days (black ticks) at which mice were killed and tissue was harvested. Gray shading indicates shared HC and CSD animals with (a), as experiments were done contemporaneously. **Right:** There was a main effect of time post-CSD for meningeal neutrophils; 16 and 24 h post-CSD, meningeal neutrophils were still significantly elevated relative to HC. In contrast, blood neutrophil levels remained significantly elevated for up to 4h post-CSD, then recovered to HC-like levels thereafter (subscript indicates time post-CSD: $n_{4h} = 3$, $n_{8h} = 5$; $n_{16h} = 4$; $n_{1d} = 4$; $n_{7d} = 4$). Results for other cell types are shown in **Figures S6-S7**. Data shown as mean \pm SEM. * $p < 0.05$, **** $p < 0.0001$."

“Figure S6: Chronic and acute stress, as well as recovery from CSD, cause tissue-specific immune cell fluctuations over time. **a) Top:** Schematic of acute

vs chronic stress study; red arrows indicate time points in days at which mice were killed and tissue was harvested. *Bottom*: Visualization of different immune cell populations in iv^- meninges, iv^+ meninges, and blood following increasing exposure to social defeat stress (subscript indicates days of defeat: $n_{HC} = 8$, $n_1 = 7$; $n_2 = 7$; $n_4 = 9$; $n_8 = 4$; $n_{CSD} = 10-11$). **b) Top**: Schematic of post-CSD recovery study; red arrows indicate time points in hours (blue ticks) or days (black ticks) at which mice were killed and tissue was harvested. Gray shading indicates shared HC and CSD animals with (a), as experiments were done contemporaneously. *Bottom*: Visualization of different immune cell populations in iv^- meninges, iv^+ meninges, and blood for increasing intervals of CSD recovery (subscript indicates time post-CSD: $n_{4h} = 3$, $n_{8h} = 5$; $n_{16h} = 4$; $n_{1d} = 4$; $n_{7d} = 4$). See **Figure S7** for individual bar graphs and full statistics. Data shown as mean \pm SEM. * $p < 0.05$, ** $p < 0.01$, *** $p < 0.001$, **** $p < 0.0001$.

“Figure S7: Chronic and acute stress act on immune cell subsets in a temporal- and tissue-specific manner. Note that these data are the same sample preparations presented in **Figures 2** and **S6**. **a)** Like neutrophils, iv^-

monocytes accumulate in the meninges after 14 days of defeat stress, though they were also elevated after 4 days (Kruskal-Wallis test: $*P < 0.05$, $H = 14.0$; Dunn's test: $*p_4 < 0.05$, $Z_4 = 2.7$; $**p_{14} < 0.01$, $Z_{14} = 3.2$). Likewise, iv^+ neutrophils were elevated after 14 days (1-way ANOVA, $*p < 0.05$, $F_{(5,40)} = 3.2$. Dunnett's test, $q_{14} = 3.7$, $**p_{14} < 0.01$). iv^+ monocytes were significantly elevated after 8 days, but not 14 (1-way ANOVA, $***p < 0.001$, $F_{(5,48)} = 5.6$. Dunnett's test, $q_8 = 4.6$, $***p_8 < 0.001$). For blood monocytes, there was a trend for elevation at day 8 (Kruskal-Wallis test: $*P < 0.05$, $H = 12.6$; Dunn's test: $p_8 = 0.09$, $Z_8 = 2.4$). iv^- meningeal B cell levels first increased on day 1, then decreased by day 8 (1-way ANOVA: $****p < 0.0001$, $F_{(5,48)} = 10.1$; Dunnett's test: $*p_1 < 0.05$, $q_1 = 3.2$; $*p_8 < 0.05$, $q_8 = 2.8$; $**p_{14} < 0.01$, $q_{14} = 3.6$). iv^+ meningeal B cells were decreased after 2 defeat encounters and remained decreased (Kruskal-Wallis test: $**P < 0.01$, $H = 16.9$; Dunn's test: $**p_2 < 0.01$, $Z_2 = 3.3$; $**p_{14} < 0.01$, $Z_{14} = 3.1$). Blood B cell levels were reduced after the 1st defeat encounter (1-way ANOVA: $****P < 0.0001$, $F_{(5,49)} = 14.7$; . Dunnett's test: $**p_1 < 0.01$, $q_1 = 3.3$; $****p_2 < 0.0001$, $q_2 = 5.0$; $**p_4 < 0.01$, $q_4 = 3.5$; $****p_8 < 0.0001$, $q_8 = 7.5$; $**p_{14} < 0.01$, $q_{14} = 5.7$). Additionally, there was a significant decrease in iv^- meningeal DCs at day 8 (1-way ANOVA: $*P < 0.05$, $F_{(5,40)} = 2.7$; Dunnett's test: $*p_8 < 0.05$, $Z_8 = 3.1$). Conversely, iv^+ meningeal DCs increased at the same time point (1-way ANOVA: $*P < 0.05$, $F_{(5,40)} = 3.1$; Dunnett's test: $*p_8 < 0.05$, $Z_8 = 3.0$). No data are available for DCs in blood at day 8, but there was an increase in this population at day 4 (Kruskal-Wallis test: $P = 0.053$, $H = 9.2$; Dunn's test: $*p_4 < 0.05$, $Z_4 = 2.9$). **b) Top:** Like neutrophils, CSD meningeal iv^- monocyte levels showed a main effect of time following stress cessation (One-way ANOVA: $****P < 0.0001$, $F_{(6,39)} = 11.2$), remaining elevated for at least 24 h (post hoc, subscripts indicate hours post-defeat: $***p_8 < 0.001$, $q_8 = 4.4$; $****p_{16} < 0.0001$, $q_{16} = 6.9$; $**p_{24} < 0.01$, $q_{24} = 3.7$). iv^- meningeal B cell levels also showed a main effect of time (One-way ANOVA: $**P < 0.01$, $F_{(6,39)} = 3.7$), with reduced levels persisting for at least 7 d (post hoc, subscripts indicate hours post-defeat: $*p_{16} < 0.05$, $q_{16} = 3.2$; $*p_{168} < 0.05$, $q_{168} = 2.8$). iv^- meningeal dendritic cells levels fluctuate over time (One-way ANOVA: $****P < 0.0001$, $F_{(6,39)} = 7.0$), with reduced levels persisting for at least 7 d (post hoc, subscripts indicate hours post-defeat: $*p_4 < 0.05$, $q_4 = 2.9$; $**p_{168} < 0.01$, $q_{168} = 3.6$). **Middle:** iv^+ meningeal monocytes and neutrophils both showed a main effect of time (Monocytes, Kruskal-Wallis test: $**P < 0.01$, $H = 21.3$. Neutrophils, One-way ANOVA: $*P < 0.05$, $F_{(6,39)} = 2.5$), though neither the monocyte levels nor the neutrophil levels were statistically significant in the post-hoc. iv^+ meningeal B cell levels were persistently reduced (One-way ANOVA: $**P < 0.01$, $F_{(6,39)} = 4.1$) lasting at least 16 h ($*p_{16} < 0.05$, $q_{16} = 3.2$). **Bottom:** There was a main effect of time post-CSD on blood T cell levels (One-way ANOVA: $*P < 0.05$, $F_{(6,40)} = 2.7$), dropping 4 h into recovery ($*p_4 < 0.05$, $q_4 = 3.0$). Blood B cell levels were likewise reduced (One-way ANOVA: $****P < 0.0001$, $F_{(6,40)} = 16.7$), for up to 24h post-CSD ($**p_4 < 0.01$, $q_4 = 3.6$; $**p_{24} < 0.01$, $q_{24} = 3.4$). Blood dendritic cell levels fluctuate with time (Kruskal-Wallis test: $***P < 0.001$, $H = 25.3$), with elevated levels 16 h post-CSD ($*p_{16} < 0.05$, $Z_{16} = 2.9$). HC = home cage, CSD = chronic social defeat stress, na = not assessed. Data shown as mean \pm SEM.

Results:

“Tissue-specific recovery dynamics following cessation of CSD stress.

To assess the stability of the neutrophilia phenotype, mice were given varying amounts of time to recover from 14 days of CSD stress. We observed tissue- and immune cell-specific changes over the course of recovery (**Figures 2B, S6B, S7B**). Blood neutrophil levels had returned to control levels within 8h of stress cessation (subscript indicates hours post-defeat: $*p_4 < 0.05$, $t_4 = 3.0$; $p_8 = 0.94$, $t_8 = 0.20$). Meningeal neutrophil levels remained elevated up to 24 h after the final defeat encounter but decreased to control levels following 7 days of recovery (subscript indicates days post-defeat: $*p_1 < 0.01$, $t_1 = 3.3$; $p_7 = 0.73$, $t_7 = 0.35$).

Consistently, findings from data collected in LysM mice supported our findings from C57BL/6J mice. Specifically, we found a similar pattern of immune cell levels in tissue collected 16h post-defeat:

“Figure S8: LysM^{gfp/+} mice recapitulate expected behavioral phenotype following CSD stress. Note these animals were also used to generate data for **Figures 3C-F, S9, and S10A-D...** **c)** Quantification of total meningeal LysM-GFP⁺ cells 16 h post-CSD stress is consistent with the pattern seen in **Figure 2B** for C57BL/6J mice (One-way ANOVA: $P = 0.080$, $F_{(2,28)} = 2.8$; Tukey’s post-hoc: $p_{\text{CSD-2h v HC}} = 0.073$, $q_{\text{CSD-2h v HC}} = 3.2$; HC and CSD-2h mice are the same animals shown in **Figures 3C-F**). † natural log-transformed.

“Figure S9: LysM^{gfp/+} mice recapitulate blood immune phenotype of C57BL/6J mice following CSD stress. Note these animals were also used to generate data for **Figures 3C-F, S8, and S10A-D...** **b)** CSD increases blood monocytes (GFP^{int};CD11b⁺;Ly6G⁻;Ly6C^{hi}: Kruskal-Wallis test, $P = 0.053$, $H = 5.9$) and neutrophils. Blood neutrophil (GFP^{hi};CD11b⁺;Ly6G⁺;Ly6C^{int}) levels

return to control levels within 16h of CSD stress cessation (Kruskal-Wallis test, $**P < 0.01$, $H = 12.9$; Dunn's post-hoc: $**p_{\text{HCvCSD-2h}} < 0.01$, $Z_{\text{HCvCSD-2h}} = 3.3$; $*p_{\text{CSD-2h v CSD-16h}} < 0.05$, $Z_{\text{CSD-2h v CSD-16h}} = 2.8$). CSD stress led to a decreased percentage of B cells that recovered within 16h post-CSD (defined as $\text{GFP}^+;\text{CD11b}^-;\text{CD19}^+$: Kruskal-Wallis test, $**P < 0.01$, $H = 12.7$; Dunn's post-hoc: $*p_{\text{HCvCSD-2h}} < 0.05$, $Z_{\text{HCvCSD-2h}} = 2.9$; $**p_{\text{CSD-2h v CSD-16h}} < 0.01$, $Z_{\text{CSD-2h v CSD-16h}} = 3.2$). With 16 h of recovery from CSD, dendritic cell levels were elevated relatively to the CSD+2 h timepoint ($\text{CD11c}^+;\text{MHCII}^+;\text{GFP-mixed}$: Kruskal-Wallis test, $***P < 0.001$, $H = 16.0$; Dunn's post-hoc: $***p_{\text{CSD-2h v CSD-16h}} < 0.001$, $Z_{\text{CSD-2h v CSD-16h}} = 4.0$). $n_{\text{HC}} = 13$, $n_{\text{CSD-2h}} = 11$, $n_{\text{CSD-16h}} = 7$."

“Figure S10: CSD increases LysM^+ myeloid cells in neurovasculature, but not brain parenchyma.....**b)** Quantification of LysM^+ cells per mm^2 of tissue analyzed show that CSD stress was significantly associated with greater LysM^+ cell accumulation in blood vessels, irrespective of brain section examined. Within 16 h of stress cessation, LysM-GFP^+ cell levels had returned to control levels (main effect of group, $****p < 0.0001$, $F_{(2,64)} = 17.8$. Šidák's post hoc: $****p_{\text{HC v CSD-2h}} < 0.0001$, $q_{\text{HC v CSD-2h}} = 5.3$; $****p_{\text{CSD-2h v CSD-16h}} < 0.0001$, $q_{\text{CSD-2h v CSD-16h}} = 4.6$). $n_{\text{HC}} = 9-10$, $n_{\text{CSD-2h}} = 9-10$, $n_{\text{CSD-16h}} = 4-5$.”

Results:

“Replication of CSD-induced meningeal immune changes.

...The LysM-GFP^+ cell increase in CSD mice was observed across intravascular, abluminal ($\leq 10\mu\text{m}$ away from a blood vessel) and non-vascular ($> 10\mu\text{m}$ away from a blood vessel) meningeal tissue compartments (**Figure 3F**: $**p < 0.01$, $F_{(1,71)} = 9.4$; post hoc $*p < 0.05$, $q = 4.3$), and—like meningeal neutrophils in C57BL/6J mice—had not recovered within 16 h of recovery from CSD stress (**Figure S8C**).

$\text{LysM}^{\text{gfp}/+}$ mice were otherwise similar to C57BL/6J mice: flow cytometry analysis showed blood neutrophilia and decreased blood B cells with recovery to baseline 16 h post-CSD (**Figures S9A-B**). Immunohistochemical (IHC) examination of brain tissue showed no evidence of neutrophil infiltration into

brain parenchyma, though there were significantly more neutrophils ‘stuck’ in the neurovasculature of CSD mice (**Figures S10A-B**:**** $p < 0.0001$, $F_{(1,52)} = 22.1$. post hoc, $q = 6.7$, *** $p < 0.001$ in medial prefrontal cortex, striatum, and hippocampal sections). As in blood, neurovascular ‘sticking’ of LysM-GFP+ cells had returned to baseline within 16 h of recovery from CSD...”

Discussion:

“In this study we have replicated the blood neutrophilia described in other preclinical stress models^{15,40} and in people with MDD¹⁸, and find that meningeal neutrophilia outlasts the well-described peripheral effects of stress on neutrophils following cessation of stress. The distinctiveness of the responses and regulation of blood and meningeal neutrophils is further underscored by the different effects of time and stressor type on immune cell dynamics in these tissues. Interestingly, though iv- meningeal neutrophil levels return to baseline within a week of recovery from CSD stress, iv-meningeal B cell levels are persistently reduced by about 30%. While a complete exploration on the immunology of defeat is beyond the scope of the current manuscript, future efforts to explore whether the persistent reduction in B cell numbers ‘primes’ the meninges for an exacerbated innate immune response, and how long B cell numbers remain low, is warranted.”

Methods (new text in bold):

“Chronic social defeat (CSD).... **Unless otherwise noted, animals were sacrificed exactly 2 h after exposure to the final defeat stress on day 14.**”

2/3) In the same line, are neutrophil levels stable in control mice across time?

Yes, the meningeal phenotype in HC mice was quite consistent over time. For example, these extant data in the main figures—shown here for reference— each represent pooled data collected from different cohorts over the course of a year:

Figure 1F:

Figure 3E:

We have also added data from home cage animals which arrived in our colony at the same time and were killed within a week of each other (days indicate days in colony), providing additional evidence of the stability of meningeal neutrophil numbers. This has been added as new **Figure S2C**:

Figure S2:... “c) Flow cytometry data from mice that arrived in our colony on the same day but were sacrificed a week apart (14 days post-delivery from Jackson Labs vs. 21 days). Critically, for iv⁻ meningeal neutrophils there are no significant differences between mice housed in our facility for 14 days vs 21 days—though there is more variation in blood neutrophils for HC mice at the 21 day time point, potentially underscoring the non-specific nature of blood (compared to meningeal) neutrophils as a read-out for response”

2/4) How do the authors reconcile the results from LysM/gfp⁺ mice with intense staining in the brain (Fig.2g) vs no infiltration of neutrophils in the parenchyma with immunohistochemistry in regions linked to mood – medial prefrontal cortex, striatum, hippocampus (Fig.S4)?

We apologise for any lack of clarity in our presentation of the data in what was formerly Figure 2G (currently **Figure 3G**). We have now modified the labels for the figure to underscore that we **do not** see a large influx of GFP⁺ cells into the brain parenchyma with the tissue clearing experiment—which is consistent with the data presented in what was formerly Figures S4A-B (currently **Figure S10A-B**). We hope that the revised figure and legend make interpretation of the data clearer for readers.

New version:

“Figure 3: CSD stress leads to increased numbers of LysM+ myeloid cells in vascular channels connecting skull BM to meninges...Before TomL intravascular (iv) injection, blood was drawn for flow cytometry (**Figure S9**). Dorsal meninges and brain (**Figure S10**) were prepared for imaging. A separate cohort of mice was used for tissue clearing...**g**) Representative image from cleared skulls showing vascular channels between skull BM and the meninges. Scale bar = 100µm. *Left:* Individual channels. *Middle:* Merged image, HC mouse. *Right:* Merged image, CSD mouse...BV = blood vessel, BM = bone marrow, CSD = chronic social defeat stress, HC = home cage... TomL = tomato lectin. Data shown as mean ± SEM. * $p < 0.05$, **** $p < 0.0001$.”

2/5) Other behavioral tests should be performed to confirm the behavioral phenotype of stressed LysM/gfp+ mice (Fig.3). Social interaction and sucrose preference tests would validate core features associated with a depressive phenotype. Number of crosses in the open field center is not sufficient to confirm anxiety and additional graphs (time in open vs closed arms, number of entries, locomotion, etc.) should be provided in a supplementary Figure. Addition of an elevated plus maze test to confirm anxiety-like behaviors would further strengthen the mouse strain behavioral characterization following social stress exposure.

We agree with the reviewer that addition of more behavioral tests to confirm the behavioural phenotype of LysM/gfp+ mice would strengthen the manuscript. We have added additional behavioral analyses to the manuscript as new **Figure S8**. We have also added material to our results and to discussion of these issues as limitations of the experimental dataset, as follows:

“Figure S8: *LysM^{gfp/+}* mice recapitulate expected behavioral phenotype following CSD stress. Note these animals were also used to generate data for **Figures 3C-F, S9, and S10A-D**. Behavioral testing began after 10 days of defeat stress; results are shown with all CSD animals combined (a), or with CSD animals stratified into susceptible vs resilient based on their performance in the social interaction (SI) test (b). By our non-formalized metrics, *LysM* mice were ‘defeated’ normally, and exhibited other signs typical of ‘depressed’ mice: e.g., poor coat quality. However, the social interaction test gave opposite to expected results. *Top row:* CSD mice spent more time in proximity to the social stimulus overall [(a) Mann Whitney test: $***p < 0.001$, $U = 27$. $n_{HC} = 12$, $n_{CSD} = 18$], though this seems to be driven mostly by CSD-R mice—which represented 50% of the CSD group [(b) Kruskal-Wallis test: $***P < 0.001$, $H = 17.3$; Dunn’s post-hoc: $****p_{HCvCSD-R} < 0.0001$, $Z_{HCvCSD-R} = 4.2$. $n_{HC} = 12$, $n_{CSD-S} = 9$, $n_{CSD-R} = 9$]. Splitting mice into susceptible vs resilient based on SI quotient scores also did not generate the expected difference between HC and CSD-S [(b) Kruskal-Wallis test: $**P < 0.01$, $H = 12.2$; Dunn’s post-hoc: $*p_{HCvCSD-R} < 0.05$, $Z_{HCvCSD-R} = 2.8$; $**p_{CSD-SvCSD-R} < 0.01$, $Z_{CSD-SvCSD-R} = 3.3$], but the HC group appeared to be less social than expected (SI scores < 2). This may be influenced by group housing of *LysM^{gfp/+}* mice with their littermates post-weaning, as opposed to C57BL/6J mice that were purchased and pair-housed in cages with dividers upon arrival to our facility, though at this time we cannot explain their behavior further. *Second row:* *LysM^{gfp/+}* mice showed an expected ‘anhedonic’ response to CSD stress in the USM test: marking [(a) Fisher’s exact test, $***p < 0.001$. $n_{HC} = 18$, $n_{CSD} = 26$. (b) Fisher’s

exact test, $**p < 0.01$. $n_{HC} = 12$, $n_{CSD-S} = 9$, $n_{CSD-R} = 9$], total urine area [(a) Mann Whitney test: $***p < 0.001$, $U = 76$. $n_{HC} = 18$, $n_{CSD} = 26$. (b) Kruskal-Wallis test: $*P < 0.05$, $H = 8.9$; Dunn's post-hoc: $*p_{HCvCSD-S} < 0.05$, $Z_{HCvCSD-S} = 2.6$; $*p_{HCvCSD-R} < 0.05$, $Z_{HCvCSD-R} = 2.5$. $n_{HC} = 12$, $n_{CSD-S} = 9$, $n_{CSD-R} = 9$], urine area near female scent [(a) Mann Whitney test: $****p < 0.0001$, $U = 64$. $n_{HC} = 18$, $n_{CSD} = 26$. (b) Kruskal-Wallis test: $**P < 0.01$, $H = 9.8$; Dunn's post-hoc: $*p_{HCvCSD-S} < 0.05$, $Z_{HCvCSD-S} = 2.7$; $*p_{HCvCSD-R} < 0.05$, $Z_{HCvCSD-R} = 2.6$. $n_{HC} = 12$, $n_{CSD-S} = 9$, $n_{CSD-R} = 9$], and preference for female [(a) Mann Whitney test: $****p < 0.0001$, $U = 93$. $n_{HC} = 18$, $n_{CSD} = 26$. (b) Kruskal-Wallis test: $**P < 0.01$, $H = 10.7$; Dunn's post-hoc: $*p_{HCvCSD-S} < 0.05$, $Z_{HCvCSD-S} = 2.5$; $**p_{HCvCSD-R} < 0.01$, $Z_{HCvCSD-R} = 3.0$. $n_{HC} = 12$, $n_{CSD-S} = 9$, $n_{CSD-R} = 9$]. *Third row: LysM^{gfp/+} mice showed an expected anxiety-like behavioral response to CSD in the OF test: novel arena exploration [(a) Mann Whitney test: $*p < 0.05$, $U = 150$. $n_{HC} = 18$, $n_{CSD} = 26$], crosses to center [(a) Mann Whitney test: $**p < 0.01$, $U = 116$. $n_{HC} = 18$, $n_{CSD} = 26$. (b) Kruskal-Wallis test: $*P < 0.05$, $H = 7.2$; Dunn's post-hoc: $*p_{HCvCSD-R} < 0.05$, $Z_{HCvCSD-R} = 2.6$. $n_{HC} = 12$, $n_{CSD-S} = 9$, $n_{CSD-R} = 9$], and time spent in the center of the arena [(a) Mann Whitney test: $*p < 0.05$, $U = 150$. $n_{HC} = 18$, $n_{CSD} = 26$. (b) Kruskal-Wallis test: $P = 0.071$, $H = 5.3$. $n_{HC} = 12$, $n_{CSD-S} = 9$, $n_{CSD-R} = 9$]. *Bottom row: We did not test both L/D and SI in the same animals, so we cannot examine the effect of stress resiliency on this task. However, LysM^{gfp/+} behavior was also in an unexpected direction on this test. a) CSD mice spent significantly more time in the light (Mann Whitney test: $*p < 0.05$, $U = 2$. $n_{HC} = 4$, $n_{CSD} = 6$). This may represent freezing, as crosses to light seemed to be in the expected direction (fewer in CSD)..."**

Results:

"Replication of CSD-induced meningeal immune changes.

LysM^{gfp/+} mice, which strongly express GFP in neutrophils (**Figures 3A-B**), recapitulated the core behavioral response to CSD stress, e.g., decreased urine scent marking (USM) preference ($***p < 0.001$) and decreased open field (OF) crosses to center ($**p < 0.01$, $t = 3.27$, $df = 2$) (**Figure S8A-B**)..."

Limitations and strengths

".... In general, LysM^{gfp/+} mice behaved similarly, but not identically, to C57BL/6Js following CSD. Increased LysM^{gfp/+} anxiety-like behavior in the OF test following social defeat has previously been reported⁷⁶, though LD data were not reported. Likewise, in a spinal cord injury depression model, LysM^{gfp/+} mice showed the expected anhedonic behavioral phenotype⁷⁷. While we cannot fully account for the behavioral differences between LysM^{gfp/+} and C57BL/6J mice, we are reasonably confident that CSD 'works' as expected in these animals..."

2/6) Similar comment for the drug-delivery rescue (Fig.6) with IFNAR blocking

antibodies. A full panel of behavioral tests (social interactions, forced swim, tail suspension, sucrose preference, etc.) could reveal domains with differential responses strengthening the study and impact of this work. It may also explain the non-normal distribution after treatment.

We thank the reviewer for this suggestion, and have incorporated additional behavioral data for the anti-IFNAR antibody experiment from *LysM^{gfp/+}* and C57BL/6J mice, which were collected as part of the original study. This comprises **Figures S22, S23B, and S24**, with accompanying figure legends, results, and discussion of this important issue as a limitation of the study, which follow below. We agree it would be useful to have other behavioral task data for the anti-IFNAR antibody experiment, however the experiment was not designed in this way and it is beyond the scope of the current paper to conduct a new experimental study of the requisite scale and complexity.

To summarize the newly incorporated data, it is clear that repeated injections necessary for administration of anti-IFNAR treatment have a confounding, strain-dependent, effect on behavior. This prompted further examination of the effect of injection stress:

Figure 22 (emphasis added):

“Figure S22: Anti-IFNAR treatment in $LysM^{GFP/+}$ mice improves anhedonic behavior in the USM task but $LysM^{GFP/+}$ mice show strain-related effects of injection stress. a) Anti-IFNAR treatment protects against the anhedonic effects of stress. Control (IgG)-antibody treated mice exposed to CSD show expected behavioral deficits compared to HC-IgG mice. *Left:* See main text for stats. *Right:* preference for female scent (Kruskal-Wallis test: $P < 0.01$, $H = 9.5$; Dunn’s post-hoc: $*p_{HC-IgGvCSD-IgG} < 0.05$, $Z_{HC-IgGvCSD-IgG} = 2.9$; $*p_{CSD-IgGvCSD-IFNAR} < 0.05$, $Z_{CSD-IgGvCSD-IFNAR} = 2.4$; $n = 12$ per group). b) *Left: No effect of injection stress on USM behavior for HC $LysM^{GFP/+}$ mice* (Fisher’s exact text: $p = 0.39$; $n_{-inj} = 16$, $n_{+inj} = 11$). *Right: however, there was an increase in nonvascular, meningeal $LysM-GFP^+$ cells with injection stress* ($**p < 0.01$, $t = 3.2$, $df = 22$). † square root-transformed. c) Comparison of the effects of injection stress on strains. *Left:* Visualization of the percentage of either WT or $LysM^{GFP/+}$ mice that marked in the USM test; all animals shown here at HC. At baseline it appears that $LysM^{GFP/+}$ mice are more likely to mark than C57BL/6J mice (81% vs 62%). HC+IgG mice from both strains marked less frequently (64% of $LysM^{GFP/+}$ mice vs 0% of C57BL/6J mice); this was not a significant difference in $LysM^{GFP/+}$ mice (Fisher’s exact text: $p = 0.39$. C57: $n_{-inj} = 34$, $n_{+inj} = 10$; $LysM$: $n_{-inj} = 16$, $n_{+inj} = 11$). *Middle:* OF behavior not impacted by strain or injection status. *Right:* Significant interaction and status effects on the social interaction test; more anhedonic behavior in both strains with injection stress (2-way ANOVA test: interaction, $*P < 0.05$, $F_{(1,48)} = 4.2$; injection, $*P < 0.05$, $F_{(1,48)} = 7.0$. Tukey’s posthoc: C57 $_{-inj}$ vs $LysM_{-inj}$ $*p < 0.05$, $q = 4.1$; C57 $_{-inj}$ vs C57 $_{+inj}$ $**p < 0.01$, $q = 5.2$; C57 $_{-inj}$ vs $LysM_{+inj}$ $*p < 0.05$, $q = 4.2$. C57: $n_{-inj} = 22$, $n_{+inj} = 10$; $LysM$: $n_{-inj} = 12$, $n_{+inj} = 8$)... e) Anhedonic (SI) and anxiety-like (OF) behavior in $LysM^{GFP/+}$ mice that underwent CSD stress. The top line shows all behavior patterns for all CSD+anti-IFNAR treated mice. The bottom line shows the same, but only shows CSD+anti-IFNAR treated mice that marked in the USM task. Only OF crosses to center were improved by anti-IFNAR treatment; these data are included in **Figure 6C** and are shown here for comparison. SI: $n_{IgG} = 9$, $n_{IFNAR} = 3$ or 9. OF: $n_{IgG} = 12$, $n_{IFNAR} = 6$ or 12...”**

Figure 23B (emphasis added):

“Figure S23: C57BL/6J mice do not show behavioral rescue with anti-IFNAR treatment, despite meningeal ‘rescue’ of neutrophils...b) Top: Social interaction (SI). There was a trend for an effect of anti-IFNAR treatment

on social approaches (main effect of treatment, $P = 0.091$, $F_{(1,26)} = 3.1$), though nothing approached significance in post-hoc testing. No other statistical tests approached significance for SI. *Bottom*: Open Field (OF). There was a significant effect of anti-IFNAR treatment on OF time in center, where in both HC and CSD groups, time in the center increased, indicative of less anxious-like behavior (main effect of treatment, $**P < 0.01$, $F_{(1,27)} = 11.1$. Tukey's post-hoc: $*p_{\text{IgG-HC v IFNAR-CSD}} < 0.05$, $q_{\text{IgG-HC v IFNAR-CSD}} = 4.2$; $*p_{\text{IgG-CSD v IFNAR-CSD}} < 0.05$, $q_{\text{IgG-CSD v IFNAR-CSD}} = 4.7$). IFNAR treatment had no improving effect on novel arena exploration; both CSD groups explored less than the HC groups (main effect of group, $**P < 0.01$, $F_{(1,27)} = 11.7$. Tukey's post-hoc: $*p_{\text{IgG-HC v IgG-CSD}} < 0.05$, $q_{\text{IgG-HC v IgG-CSD}} = 2.6$; $**p_{\text{IgG-CSD v IFNAR-HC}} < 0.01$, $q_{\text{IgG-CSD v IFNAR-HC}} = 3.1$; $*p_{\text{IFNAR-HC v IFNAR-CSD}} < 0.05$, $q_{\text{IFNAR-HC v IFNAR-CSD}} = 2.2$). ...”

Figure 24 (emphasis added):

“Figure S24: Exploration of the effect of injection and strain on behavior and immune cell dynamics in C57BL/6J home cage (HC) mice. **a) Top:** Social interaction. **Repeated injections had an anhedonic effect on C57BL/6J mice for the SI quotient** (Mann Whitney test: $***p < 0.001$, $U = 29$. $n_{-inj} = 22$, $n_{+inj} = 10$). The number of social approaches was also reduced ($*p < 0.05$, $U = 55$). Time spent interacting with the non-social object increased ($****p < 0.0001$, $U = 21$). **Bottom:** Open field. **Repeated injections had an anxiogenic effect on C57BL/6J for exploration of a novel arena** ($**p < 0.01$, $U = 87$. $n_{-inj} = 19$, $n_{+inj} = 18$). **b) In C57BL/6J mice, there were no effects of injection stress on neutrophils.** However, injection stress increased the percentage of iv⁺ meningeal T cells ($****p < 0.0001$, $U = 48.5$. n.

inj = 29, n_{+inj} = 18), and decreased the percentage of iv⁻ meningeal dendritic cells (**p < 0.01, U = 112. n_{-inj} = 24, n_{+inj} = 18). There were no effects on iv⁺ meningeal cells. In blood, monocyte levels were decreased by injection stress (***p < 0.0001, U = 64. n_{-inj} = 32, n_{+inj} = 18). Neutrophil levels were not significantly elevated. Dendritic cell levels were reduced (**p < 0.01, U = 102. n_{-inj} = 24, n_{+inj} = 18). HC = home cage, CSD = chronic social defeat. Data shown as mean ± SEM.”

Results (emphasis added):

“IFNAR blockade improves the behavioral response to CSD stress.

We assessed whether peripheral blockade of IFN-I signaling rescues the negative behavioral sequelae associated with CSD stress by administering an IFNAR-blocking antibody to LysM^{gfp/+} mice (**Figure 6A**). As expected, in the USM test for anhedonia, more HC+control antibody (IgG) mice marked compared to the CSD+IgG group (**Figure S22A**: **p < 0.01, $\chi^2 = 9.5$; post hoc *p < 0.05). Anti-IFNAR treatment rescued the CSD phenotype, with no difference in marking frequency between HC+IgG and CSD+IFNAR groups. **For other examined parameters, HC animals appeared ‘stressed’, which we determined was influenced by repeated injections (Figures S22B-C, S23)**. We therefore examined the CSD group alone.

In the USM task, anti-IFNAR treatment protected mice from the depressive-like effects of CSD (**Figure 6B**: (*p < 0.05, U = 36). However, USM data for the CSD+anti-IFNAR group was non-normally distributed, with distinct groups of animals that marked (+) or did not mark (-) in the test (**Figure S22D**). We therefore treated these animals separately in our remaining analyses. There was a significant difference in the OF task between CSD+IgG and CSD+anti-IFNAR(+) mice, where CSD+IgG animals were more anxious (**Figure 6C**: *P < 0.05, H = 9.1; post hoc: *p < 0.05, Z = 2.8). There were no other significant differences in behavior between groups (**Figure S22E**).

IFNAR blockade attenuates neutrophil migration into, but not B cell egress from, meningeal tissue in CSD stressed mice.

...Like LysM^{gfp/+} mice, C57BL/6J mice were behaviorally impacted by repeated injection stress (**Figures S22C, S23B, S24A**). Unlike LysM^{gfp/+} mice, the effects of injection stress had no impact on meningeal myeloid cell levels (**Figures S24B**).”

Discussion (emphasis added):

“It is not presently clear how meningeal neutrophils influence behavior given they do not appear to enter the brain parenchyma. Further

exploration of the pathways by which neutrophils enter the meninges and exert their effects on brain tissue is merited...

... IFN-I depletion in C57BL/6J mice normalized CSD-related meningeal neutrophil levels. While replication of these effects in *LysM^{gfp/+}* mice presented additional complexity (see **Limitations**), CSD+anti-IFNAR–treated mice showed improved anhedonic behavior. Interestingly, we observed meningeal neutrophilia in otherwise non-stressed HC mice who failed to show preference for a sexually hedonic stimulus. This is consistent with reports suggesting individual differences in peripheral immunity influence stress susceptibility⁵³.

Limitations and strengths:

“Finally, depletion of IFN-I signaling showed pleiotropic, strain-dependent effects, with greater sensitivity to repeat injection in HC C57BL/6Js compared to HC *LysM^{gfp/+}* mice. In general, *LysM^{gfp/+}* mice behaved similarly, but not identically, to C57BL/6Js following CSD. Increased *LysM^{gfp/+}* anxiety-like behavior in the OF test following social defeat has previously been reported⁷⁷. Likewise, in a spinal cord injury model, *LysM^{gfp/+}* mice showed anhedonic behavioral phenotype⁷⁸. Thus, while we cannot fully account for the behavioral differences between *LysM^{gfp/+}* and C57BL/6J mice, we are reasonably confident that CSD ‘works’ as expected in these animals.

Other studies have highlighted IFN-I responses in brain cells during stress^{79,80}. Given that the MAR1-5A3 IFNAR antibody clone we used cannot cross the blood brain barrier (BBB)⁸¹, the mediating effects of anti-IFNAR treatment on the negative sequelae of CSD stress in our model likely do not involve cells within the brain parenchyma. The cellular source of stress-related IFN-I, and the relative contributions to behavior of IFNAR signaling in different cell types, requires further investigation.”

To address the reviewer’s more general question about the bimodal nature of USM marking, we examined further the heterogeneity in HC mice. We found that variability in marking behaviour was associated with meningeal neutrophil levels in HC mice (new **Figures S2D, S8D**):

“Figure S2:…d) HC mice that are ‘anhedonic’, i.e., do not show preference for female in the USM task, have elevated levels of neutrophils in the absence of other stressors (1-way ANOVA, $**p < 0.01$, $F_{(2,42)} = 6.3$. Dunnett’s posthoc test: $*p_{\text{HC+mark v HC-mark}} < 0.05$, $q_{\text{HC+mark v HC-mark}} = 2.7$; $**p_{\text{HC+mark v CSD}} < 0.01$, $q_{\text{HC+mark v CSD}} = 3.2$). ~53% of HC mice failed to mark; compare with **Figure S8D**…”

“Figure S8: … d) HC mice that are ‘anhedonic’, i.e., do not show preference for female in the USM task, do not have elevated levels of LysM-GFP cells in the absence of other stressors, in contrast to C57BL/6J mice (Kruskal-Wallis, $P = 0.063$, $H = 5.5$). ~72% (13/18) of LysM^{gfp/+} HC mice mark; compare with **Figure S2D**. †square root-transformed to improve normality.”

We have updated the discussion to include these findings:

“…Interestingly, we observed meningeal neutrophilia in otherwise non-stressed HC mice who failed to show preference for a sexually hedonic

stimulus. This is consistent with other reports suggesting individual differences in peripheral immunity influence stress susceptibility⁵³.”

2/7) A discussion on sex differences and data interpretation of the current findings beyond limitations would strengthen the manuscript with increasing evidence highlighting sex-specific mechanisms for brain barrier and immune function in the context of social stress responses.

We thank the reviewer for this valuable suggestion. We have added the following new text to the discussion:

“A major limitation of this and similar studies is the exclusive use of adult male mice. Future extension of this investigation to females will be crucial, especially as depression disproportionately affects women⁶⁹. In addition, females may be more acutely susceptible to BBB permeability after stress induction^{70–72} and exhibit a more robust anti-viral IFN-I response^{73,74}. These mechanisms could lead to sex-specific vulnerabilities to meningeal neutrophilia and associated behavioral outcomes.”

2/8) Sex of the mice (males only) should be mentioned in the abstract.

We agree and have modified the abstract as follows:

“Animal models of stress and stress-related disorders are associated with blood neutrophilia. The mechanistic relevance of this to symptoms or behavior is unclear. We used flow cytometry, immunohistochemistry, whole tissue clearing, and single-cell sequencing to characterize the meningeal immune response to chronic social defeat (CSD) stress in **male** mice.”

2/9) It is not always clear in the introduction when the authors refer to preclinical or clinical studies.

We apologise for any lack of clarity in our summary of the preclinical and clinical literature. We have provided more context in the introduction (relevant new text highlighted in bold) to avoid confusion:

“**In humans**, chronic inflammation has been linked to both psychosocial stress¹ and major depressive disorder (MDD)², highlighting a possible role for the immune system as an intermediary between psychological risk factors and the development of mood disorders. Proinflammatory cytokines produced by innate immune cells have attracted attention given strong evidence that they are associated with **MDD**³ and depressive-like behavior in animal models⁴. In particular, proinflammatory type I interferons (IFN-I) **are associated with development of MDD**^{5,6} and induce depressive symptoms in otherwise nondepressed **people**⁷ and **quasi-depressive behaviors in laboratory animals**^{8,9}...”

“...Mechanisms by which neutrophils contribute to depressive symptoms are currently unclear, however, we and others have demonstrated **in mice** that cells...”

2/10) The CSD paradigm is unusual and additional details should be provided.

We disagree that the CSD paradigm is “unusual”. This version of the social defeat paradigm is based on some of the earliest publications on social defeat stress (Kudryavtseva 1991), and we have been using it in its current format for more than 15 years (Brachman et al, 2015; Lehmann et al., 2022; 2020; 2019; 2018; 2017; 2016; 2013; 2012; Lehmann and Herkenham, 2011; Lynall et al., 2021; Samuels et al., 2023; Scheinert et al., 2016; Schloesser et al., 2010). We highlight here a line from the extant text that summarize this background context:

Introduction:

“Subjecting WT mice to chronic social defeat (CSD) stress—which reliably induces depressive- and anxiety-like behaviors^{22–25}...”

We also have provided additional data and analyses on the social interaction test, splitting mice into ‘susceptible’ vs ‘resilient’ based on social interaction quotient, given the widespread use of this metric in social defeat studies (Golden, et al., 2011). We furthermore provide data for the light/dark (LD) box, an independent test for anxiety-like behavior. These data are shown here for C57BL/6J mice and have been added as **Figures 1B-E** and **Figure S1**. Likewise, these data have been added for LysM mice as **Figure S7**, and are included in our response to **2/5**.

“Figure 1: Meningeal neutrophils are elevated following chronic social defeat (CSD) stress and are associated with depressive-like behavioral change. **a)** CSD mice were behaviorally tested on days 10-13... CSD results in expected sexual (**b**) and social (**c**) anhedonia in USM and SI tests, respectively. In (**b**), pink arrowheads indicate female urine, pipetted onto the blotting paper, and blue arrowheads point to representative urine scent marks from male test

mice. CSD leads to increased anxiety-like behavior in the (d) OF test and (e) LD box. Additional behavioral data shown in **Figure S1...**"

“Figure S1: Behavioral testing in C57BL/6J mice following chronic social defeat (CSD) stress shows expected phenotype (NB: these animals were used for flow cytometry analyses shown in **Figures 1F-H & S3-4**). Behavioral testing began after 10 days of defeat stress; results are shown with all CSD animals combined (**a**), or with CSD animals stratified into susceptible vs resilient based on their performance in the social interaction (SI) test (**b**). *Top row:* For the SI test—which assesses social anhedonia as a depressive-like phenotype—SI quotient was calculated based on time spent engaging with the social stimulus divided by time spent engaging with the non-social stimulus (see **Figure 1C** for visualization). **a**) Mann Whitney test: $p = 0.075$, $U = 378$. $n_{HC} = 34$, $n_{CSD} = 30$. **b**) Kruskal-Wallis test: $****P < 0.0001$, $H = 21$; Dunn’s post-hoc: $**p_{HCvCSD-S} < 0.01$, $Z_{HCvCSD-S} = 3.5$; $****p_{CSD-SvCSD-R} < 0.0001$, $Z_{CSD-SvCSD-R} = 4.2$. $n_{HC} = 34$, $n_{CSD-S} = 20$, $n_{CSD-R} = 10$. Social approaches represent the number of times the test mouse approached the social stimulus for sniffing. **a**) See **Figure 1C**. **b**) Kruskal-Wallis test: $**P < 0.01$, $H = 12$; Dunn’s post-hoc: $**p_{HCvCSD-S} < 0.01$, $Z_{HCvCSD-S} = 3.3$; $*p_{CSD-SvCSD-R} < 0.05$, $Z_{CSD-SvCSD-R} = 2.4$. $n_{HC} = 34$, $n_{CSD-S} = 20$, $n_{CSD-R} = 10$. Time spent engaging with the social stimulus [in (**b**), Kruskal-Wallis test: $***P < 0.001$, $H = 15$; Dunn’s post-hoc: $*p_{HCvCSD-S} < 0.05$, $Z_{HCvCSD-S} = 2.9$; $**p_{CSD-SvCSD-R} < 0.01$, $Z_{CSD-SvCSD-R} = 3.6$. $n_{HC} = 34$, $n_{CSD-S} = 20$, $n_{CSD-R} = 10$) and with the non-social stimulus, which comprise the SI quotient, are next. *Second row:* The urine scent marking

(USM) task was used to assess sexual anhedonia, or depressive-like behavior; marking indicates engagement with a hedonic (i.e., female scent) stimulus. Data were stratified by task response – marking (+mark), or no marks present (-mark— see **Figure 1B. a**) Fisher's exact test, **** $p < 0.0001$. $n_{HC} = 26$, $n_{CSD} = 15$. **b**) Fisher's exact test, * $p < 0.05$. $n_{HC} = 17$, $n_{CSD-S} = 2$, $n_{CSD-R} = 4$. Total area for test subject urine marking is shown [(**a**) Mann Whitney test: **** $p < 0.0001$, $U = 14.5$. $n_{HC} = 16$, $n_{CSD} = 13$], along with marks made in close proximity to the female scent [(**a**) Mann Whitney test: *** $p < 0.001$, $U = 29$. $n_{HC} = 16$, $n_{CSD} = 13$], %Preference indicates preference for female scent, which is derived by dividing the marking area near the female urine spot by total marking area (**Figure 1B**). *Third row:* The open field (OF) test was used to assess anxiety-like behavior (more willingness to explore the novel arena indicates less anxiousness). **a**) See **Figure 1D b**) Kruskal-Wallis test: **** $P < 0.0001$, $H = 20$; Dunn's post-hoc: **** $p_{HCvCSD-S} < 0.0001$, $Z_{HCvCSD-S} = 4.5$. $n_{HC} = 18$, $n_{CSD-S} = 12$, $n_{CSD-R} = 4$). We also tracked crosses into the center of the arena, time spent in the center of the arena, and number of fecal boli as a crude proxy for stress, though these were not significant. *Bottom row:* The light/dark (LD) box assesses anxiety-like behavior—time spent exploring a brightly lit area as opposed to a covered dark space is quantified. Additionally, the number of crosses made from the dark side to the light side shows significant differences between groups: **a**) See **Figure 1E. b**) Kruskal-Wallis test: ** $P < 0.01$, $H = 11$; Dunn's post-hoc: ** $p_{HCvCSD-S} < 0.01$, $Z_{HCvCSD-S} = 3.4$. $n_{HC} = 11$, $n_{CSD-S} = 10$, $n_{CSD-R} = 3$). Data shown as mean \pm SEM."

Results (emphasis added)

"CSD causes expected depressive- and anxiety-like behavioral changes.

We first confirmed that our well-established CSD paradigm^{22–25} elicits expected depressive- and anxiety-like behavior in C57BL/6J WT mice (Figures 1A-E, S1A). Male CSD-treated mice exhibited sexual and social anhedonia in the urine scent marking (USM) test²⁴ (reduced preference for female scent: **** $p < 0.0001$, $U = 30$) and social interaction (SI) test (reduced approaches to a novel CD-1 male: ** $p < 0.01$, $U = 367$), respectively. CSD stress also induced anxiety-like behavior in both the open field (OF) test (less exploration of the novel arena: **** $p < 0.0001$, $U = 75$) and the light/dark (LD) test (fewer crosses to light: *** $p < 0.001$, $U = 38$).

Approximately 30% of the CSD-treated mice were "resilient" to stress, as assessed using an SI quotient ≥ 2 (time socially investigating / time on non-social investigation; **Figure S1B**), comparable to similar versions of the social defeat stress model³⁰.

Methods

"Light/dark box (LD): The LD test uses an acrylic box (50 cm x 25 cm with 30 cm walls) with aversive lighting (~40 lux). Approximately 1/3 of the box is

enclosed and dark; an opening allows crossover between light and dark sections. Mice were placed in the light compartment and allowed to move freely for 10 min while the experimenter was out of the room. Time spent in the light compartment and number of crosses between the light and dark sides were scored from video recordings using TopScan. Low scores indicated anxiety-like behavior

Social Interaction (SI): Two perforated acrylic cylinders, one containing an unfamiliar CD-1 mouse and the other empty, were placed in the OF arena with red lighting. The test mouse was placed in the middle of the arena and allowed to freely explore for 10m with the experimenter outside of the room. TopScan was used on captured videos to track approaches to the social stimulus and time spent investigating social vs. non-social stimuli. Fewer social interactions indicated anhedonic behavior.”

We also performed a sensitivity analysis by reexamining our flow cytometry data, regrouping animals into susceptible and resilient, and found no major deviations from our original conclusions. These data are included here and in the paper as new **Figure S4**:

Figure S4: Stratifying mice into ‘susceptible’ (CSD-S) vs ‘resilient’ (CSD-R) has no effect on flow cytometry results shown in **Figure S3**. **a** *Top:* iv⁻ meningeal tissue results (proportion relative to total live CD45⁺ cells). Increased monocytes (Kruskal-Wallis test: * $P < 0.05$, $H = 7.4$; Dunn’s post-hoc: * $p_{HCvCSD-R} < 0.05$, $Z_{HCvCSD-R} = 2.7$). Increased neutrophils (Kruskal-Wallis test: * $P < 0.05$, $H = 9.1$; Dunn’s post-hoc: * $p_{HCvCSD-S} < 0.05$, $Z_{HCvCSD-S} = 2.8$). Reduced B cells (Kruskal-Wallis test: **** $P < 0.0001$, $H = 24$; Dunn’s post-hoc: **** $p_{HCvCSD-S} < 0.0001$, $Z_{HCvCSD-S} = 4.8$; * $p_{HCvCSD-R} < 0.05$, $Z_{HCvCSD-R} = 2.9$). $n_{HC} = 34$, $n_{CSD-S} = 20$, $n_{CSD-R} = 10$. *Middle:* iv⁺ meningeal tissue results (proportion relative to total live CD45⁺ cells). Increased monocytes (Kruskal-Wallis test: * $P < 0.05$, $H = 6.6$; Dunn’s post-hoc: * $p_{HCvCSD-S} < 0.05$, $Z_{HCvCSD-S} = 2.5$). Increased neutrophils (Kruskal-Wallis test: * $P < 0.05$, $H = 9.2$; Dunn’s post-hoc: ** $p_{HCvCSD-S} < 0.01$, $Z_{HCvCSD-S} = 3.0$). Reduced B cells (Kruskal-Wallis test: *** $P < 0.001$, $H = 16$; Dunn’s post-hoc: ** $p_{HCvCSD-S} < 0.01$, $Z_{HCvCSD-S} = 3.5$; * $p_{HCvCSD-R} < 0.05$, $Z_{HCvCSD-R} = 2.8$). $n_{HC} = 34$, $n_{CSD-S} = 20$, $n_{CSD-R} = 10$. *Bottom:* peripheral blood results (proportion relative to total live CD45⁺ cells). Increased neutrophils (Kruskal-Wallis test: **** $P < 0.0001$, $H = 29$; Dunn’s

post-hoc: **** $p_{HCvCSD-S} < 0.0001$, $Z_{HCvCSD-S} = 4.6$; *** $p_{HCvCSD-R} < 0.001$, $Z_{HCvCSD-R} = 3.9$). Reduced B cells (Kruskal-Wallis test: **** $P < 0.0001$, $H = 30$; Dunn's post-hoc: **** $p_{HCvCSD-S} < 0.0001$, $Z_{HCvCSD-S} = 4.9$; *** $p_{HCvCSD-R} < 0.001$, $Z_{HCvCSD-R} = 3.7$). $n_{HC} = 29$, $n_{CSD-S} = 14$, $n_{CSD-R} = 9$. **b)** Additional stratification of HC animals into those with $SI > 2$ or $SI < 2$. *Top:* iv⁻ meningeal tissue results (proportion relative to total live CD45⁺ cells). Increased monocytes (Kruskal-Wallis test: * $P < 0.05$, $H = 11$; Dunn's post-hoc: ** $p_{HC>2 v CSD>2} < 0.01$, $Z_{HC>2 v CSD>2} = 3.2$). Increased neutrophils (Kruskal-Wallis test: ** $P < 0.01$, $H = 15$; Dunn's post-hoc: * $p_{HC>2 v CSD>2} < 0.05$, $Z_{HC>2 v CSD>2} = 2.6$; *** $p_{HC>2 v CSD<2} < 0.001$, $Z_{HC>2 v CSD<2} = 3.7$). Reduced B cells (Kruskal-Wallis test: *** $P < 0.001$, $H = 21$; Dunn's post-hoc: ** $p_{HC>2 v CSD<2} < 0.01$, $Z_{HC>2 v CSD<2} = 3.4$). $n_{HC>2} = 18$, $n_{HC<2} = 18$, $n_{CSD>2} = 10$, $n_{CSD<2} = 22$. *Middle:* iv⁺ meningeal tissue results (proportion relative to total live CD45⁺ cells). Increased neutrophils (Kruskal-Wallis test: * $P < 0.05$, $H = 11$; Dunn's post-hoc: ** $p_{HC>2 v CSD<2} < 0.01$, $Z_{HC>2 v CSD<2} = 3.1$). Reduced B cells (Kruskal-Wallis test: *** $P < 0.001$, $H = 16$; Dunn's post-hoc: * $p_{HC>2 v CSD>2} < 0.05$, $Z_{HC>2 v CSD>2} = 2.8$; ** $p_{HC>2 v CSD<2} < 0.01$, $Z_{HC>2 v CSD<2} = 3.0$). $n_{HC>2} = 18$, $n_{HC<2} = 18$, $n_{CSD>2} = 10$, $n_{CSD<2} = 22$. *Bottom:* peripheral blood results (proportion relative to total live CD45⁺ cells). Increased neutrophils (Kruskal-Wallis test: **** $P < 0.0001$, $H = 30$; Dunn's post-hoc: *** $p_{HC>2 v CSD>2} < 0.001$, $Z_{HC>2 v CSD>2} = 3.7$; **** $p_{HC>2 v CSD<2} < 0.0001$, $Z_{HC>2 v CSD<2} = 4.3$). Reduced B cells (Kruskal-Wallis test: **** $P < 0.0001$, $H = 31$; Dunn's post-hoc: ** $p_{HC>2 v CSD>2} < 0.01$, $Z_{HC>2 v CSD>2} = 3.3$; **** $p_{HC>2 v CSD<2} < 0.0001$, $Z_{HC>2 v CSD<2} = 4.2$). $n_{HC>2} = 16$, $n_{HC<2} = 15$, $n_{CSD>2} = 9$, $n_{CSD<2} = 16$. Data shown as mean \pm SEM.

Finally, in the results, the following new text has been added under the subheading, "CSD increases meningeal and blood neutrophil abundance":

"For sensitivity analyses, we regrouped mice into susceptible ($SI < 2$) and resilient ($SI \geq 2$)³⁰; this did not alter the main effects of CSD on immune cell populations (**Figure S4**)."

2/11) Is the CD-1 aggressor always the same throughout the 1-14 consecutive days?

Yes, usually the CD-1 aggressor was the same throughout, unless the pairing did not result in antagonistic encounters, or if the C57 test mouse was overpowering the CD-1 mouse. In these cases, the C57 mouse was paired with a different CD-1 mouse until the desired subordination behavior was achieved, and kept with that CD-1 until the end of the study. We have added these extra details into the methods (new text highlighted here in bold):

"*Chronic social defeat (CSD)*: CSD stress was performed as previously reported²². Briefly, the 'intruder' test mouse was introduced into the home cage of an aggressive, CD-1 (Taconic; Rensselaer, New York) retired breeder. The two were separated by a perforated barrier and given 24 h to acclimate; the barrier allowed for olfactory, visual, and auditory

communication, but not tactile contact. Each day for either 1, 2, 4, 8, or 14 consecutive days, depending on the experiment, the barrier was lifted, and antagonistic encounters were allowed to occur for 5 m. Interactions were monitored by a trained individual to ensure the test mouse exhibited submissive behavior and conversely that the CD1 exhibited dominant behavior. **The C57/CD1 pairs were maintained throughout the study unless the CD1 failed to show dominant aggression toward the C57. When this occurred, the C57 was paired with a different CD1 mouse until submissive behavior in the C57 was evident, and the new pairing was maintained thereafter.** Efforts to minimize physical damage were taken, i.e. CD1 mice were lightly anesthetized with isoflurane and incisors were trimmed prior to starting the social defeat paradigm, and on a weekly basis thereafter. Test mice were shaved and inspected for the presence of wounds at the end of the experiment; wounds were scored on a scale from 1-10 (1 = no injuries, 5 = combination of old and new bite marks, 10 = severe wounds). Animals with wound scores of 10 were excluded.”

2/12) Could aggression level affect immune cell level in the blood and meninges (as briefly mentioned in the limitation part for wounding)? On Fig.1 most of the stressed mice display % of meningeal neutrophils similar to the control group. Are those mice the ones without wounds?

We agree there is overlap between the values from the two groups. We have replotted to make the spread of data and uncertainty on the mean clearer:

To further explore the more general issue raised by the reviewer about the relationship between meningeal neutrophils and wounding, we performed new analyses testing for correlations between the degree of wounding and levels of neutrophils and other immune cells. We did not observe significant associations between wounding and neutrophil levels in any tissues. There was a relationship between wounding and iv- meningeal monocytes:

Figure S5: No relationships between neutrophils and wounding scores in CSD mice. However, wounding severity was associated with iv⁻ meningeal monocyte levels. HC mice were excluded from this analysis as the distribution of their wound scores at 1 skews the analysis. **a) Monocytes.** iv⁻ meningeal (black circles): $\beta = 0.30$, Std error = 0.14, $*p < 0.05$, $n = 37$; iv⁺ meningeal (pink squares): $\beta = 0.0068$, Std error = 0.064, $p > 0.05$, $n = 37$; blood (purple diamonds): $\beta = 0.054$, Std error = 0.15, $p > 0.05$, $n = 36$. **b) Neutrophils.** iv⁻ meningeal (black circles): $\beta = 0.63$, Std error = 0.47, $p > 0.05$, $n = 39$; iv⁺ meningeal (pink squares): $\beta = 0.0068$, Std error = 0.47, $p > 0.05$, $n = 39$; blood (purple diamonds): $\beta = 1.5$, Std error = 0.99, $p > 0.05$, $n = 38$. **c) T cells.** iv⁻ meningeal (black circles): $\beta = 0.60034$, Std error = 0.13, $p > 0.05$, $n = 30$; iv⁺ meningeal (pink squares): $\beta = -0.024$, Std error = 0.019, $p > 0.05$, $n = 30$; blood (purple diamonds): $\beta = 0.15$, Std error = 0.36, $p > 0.05$, $n = 29$. **d) B cells.** iv⁻ meningeal (black circles): $\beta = -0.59$, Std error = 0.33, $p > 0.05$, $n = 30$; iv⁺ meningeal (pink squares): $\beta = -0.088$, Std error = 0.078, $p > 0.05$, $n = 30$; blood (purple diamonds): $\beta = -1.7$, Std error = 0.92, $p > 0.05$, $n = 29$. **e) Dendritic cells.** iv⁻ meningeal (black circles): $\beta = -0.017$, Std error = 0.14, $p > 0.05$, $n = 30$; iv⁺ meningeal (pink squares): $\beta = 0.0089$, Std error = 0.034, $p > 0.05$, $n = 30$; blood (purple diamonds): $\beta = -0.096$, Std error = 0.047, $p > 0.05$, $n = 29$. **f) Distribution of wound scores across both groups.** NA = not assessed.

New text has been added to the supplemental methods, results, and discussion as follows:

Results:

CSD increases meningeal and blood neutrophil abundance.

“...We considered the potential impact of fight-related wounding on immune cell dynamics (**Figure S5**). Of all cell types and tissues examined, only iv⁻ meningeal monocytes showed a significant relationship with wounding ($\beta = 0.30$, Std error = 0.14, * $p < 0.05$, $n = 37$).”

Limitations and strengths (new text in bold):

“Another limitation of our study is that social defeat stress is obligatorily associated with some degree of fight-inflicted wounding. **Despite our efforts to mitigate this (see Methods), we observed that** higher wound scores were associated with greater levels of iv⁻ meningeal monocytes. This may require a more nuanced approach when translating results from social defeat stress models into a clinical setting, as has been discussed elsewhere⁷⁵”

Supplemental Methods:

Additional statistics...“Linear regression was used to assess wounding as a predictor for immune cell populations according to the formula [cell population] ~ [wound score]...”

2/13) I wonder if it would be possible to perform a distribution analysis for brain areas involved in emotion regulation for the brain clearing-related data. The stress-induced increase in neutrophil accumulation seems to be evenly distributed (Fig.2F) how do the authors explain this vs human condition?

We agree this is an interesting topic for consideration and have added a new IHC analysis focused on the nucleus accumbens and prelimbic cortex as a panel for **Figure S10** (formerly Figure S4) with accompanying figure legend, shown here. We did not see evidence motivating further investigation given the even distribution of LysM-GFP cells, as noted by the reviewer. We have updated the discussion (below), highlighting this interesting conceptual point and the potential parallels with findings in depression and in neurodegenerative disorders.

“Figure S10: CSD increases LysM⁺ myeloid cells in neurovasculature, but not brain parenchyma... **d)** In a subset of samples, we looked at LysM-GFP⁺ myeloid cell sticking in the prelimbic cortex (PrL) and nucleus accumbens (NAc) given previous work highlighting a key role for these areas in stress-related anhedonic behavioral changes (Hodes et al., 2015; Menard et al., 2017; Dion-Albert et al., 2022). *Left:* representative section showing area of tissue examined for neutrophil counting. *Middle:* No differences between HC and CSD mice in LysM-GFP⁺ cell vascular sticking in either region of interest (ROI). GFP⁺ cell counts were normalized to the ROI investigated. † natural log-transformed values to improve normality. *Right:* no differences in the ROI areas examined between groups. $n_{HC} = 5$, $n_{CSD} = 7$...”

Discussion:

“The apparently even distribution of LysM-GFP cells throughout the meninges and brain—consistent with the sporadic, stress-associated microhemorrhages we have previously investigated^{22,25}—is in contrast to reports suggesting localized BBB damage preclinically and clinically⁶¹. Interestingly, our scRNAseq data suggested increased expression of *Translocator protein (Tspo)* and related accessory proteins that mediate neutrophil chemotaxis^{62,63}. We recently demonstrated a relationship between meningeal TSPO signal intensity and brain regions associated with clinical MDD⁶⁴. Interestingly, TSPO signal intensity in microglia is related to duration (in years) of depressive symptoms⁶⁵. This could suggest that our observations represent a relatively early stage in the progression of depressive symptoms; future efforts to determine the precise nature of this myeloid ‘sticking’ behavior over the progression of MDD will be important.

Importantly, depressive symptoms are common in neurological disorders like stroke⁶⁶ and neurodegeneration⁶⁷, and in preclinical models of neurodegeneration, neutrophil capillary stalling has been linked to reduced

brain perfusion and memory deficits⁶⁸. This raises the possibility that altered neutrophil properties are a shared biological feature between neurological and psychiatric disorders; comparison of neutrophil phenotypes across brain disorders may suggest novel transdiagnostic treatment targets.”

2/14) Images should be provided for the drug-delivery rescue experiment (Fig.6).

We thank the reviewer for this suggestion and have now included representative images of the analyzed meningeal samples, shown here and in **Figure 6D**:

“Figure 6: ... d) Left: Representative images for dorsal whole-mount meningeal tissue from CSD+IgG and CSD+anti-IFNAR treated mice. Scale bar = 1 mm. **Right:** Images from HC and CSD mice showing hand-counted LysM-GFP⁺ cells, normalized to indicated area...”

2/15) It might be relevant to define neutrophilia in the abstract to reach a wider audience.

We are happy to make this change to appeal to a broader audience. Modified text in abstract (new text highlighted in bold):

We find that chronic, but not acute, stress causes accumulation of neutrophils in the meninges —i.e., **“meningeal neutrophilia”**—and CSD increases neutrophil trafficking in vascular channels emanating from skull bone marrow (BM).

2/16) Fig.5 the abbreviation neut has never been used before, neutrophils may be more appropriated.

We apologize for the inconsistency in labeling. This has now been corrected as follows:

2/17) I would refrain from using strong words such as robustly.

We disagree that the evidence for increased neutrophils in the meninges following CSD stress is not robust. We consistently observed this phenomenon in multiple cohorts of mice, in independent studies, and over a span of several years. The evidence provided draws on three independent techniques (i.e., flow cytometry, microscopy, and single cell sequencing) from multiple strains (WT and LysM). Hence, we do feel “robustly” confident in this particular finding.

Reviewer #3 (Remarks to the Author):

3/1) I co-reviewed this manuscript with one of the reviewers who provided the listed reports. This is part of the Nature Communications initiative to facilitate training in peer review and to provide appropriate recognition for Early Career Researchers who co-review manuscripts.

We thank the reviewer for their contribution.

Reviewer #4 (Remarks to the Author):

In the paper by Kigar and colleague entitled "Chronic social defeat stress induces meningeal neutrophilia via type I interferon signaling", the authors describe and phenotype the accumulation of neutrophils in the meninges upon chronic social defeat using a combination of flow Cytometry, histology and single cell sequencing. They also implicate type I interferon in the neutrophilic phenomenon. The topic of research is of great interest given the regained interest in understanding how the immune cells of the brain borders can impact physiological and pathological brain function. The authors also use

complementary approaches to convincingly demonstrate the infiltration of neutrophils in the meninges after chronic social defeat. Some conclusions put forward from the authors however are not substantiated by the presented data and would require additional experiment or alteration in interpretation:

4/1) The authors use the measurement of the number of neutrophils in the skull bone marrow channel (Figure 2g-j) to claim that they come from the skull bone marrow, a statement that is repeated throughout the manuscript. The sole use of that data, however, does not exclude a potential large contribution of the more classical blood extravasation signaling. Particularly, the large accumulation of the meningeal neutrophils in or around the meningeal blood vasculature, and in the brain vasculature strongly suggest a potential consequential contribution of the blood as a source of meningeal neutrophils.

Like Reviewer 4, we expected there to be more of a relationship between blood neutrophils and meningeal neutrophils, but our evidence did not support this conclusion. However, we agree that the skull trafficking data could be strengthened so we have reported two new lines of evidence in the revised paper.

First, we compared our data to publicly available single-cell sequencing data from neutrophils in skull, blood, and other bone marrow locations (Evrard 2018, Kolabas 2023). This analysis showed that meningeal neutrophils are transcriptionally distinct from blood and share similarities with bone marrow. New sections have been added to the figures (**Figure S14**), figure legends, methods, results, and discussion as follows, and accompanying code provided in the paper github repository:

Figure S14 Principal Component Analysis (PCA) plots visualize the transcriptomic similarity of neutrophil and preneutrophil populations across tissues (blood, meninges and various bone marrow locations), cell types (neutrophils and preneutrophils), condition (HC, control and CSD, stress), and dataset. We integrated our meningeal dataset with two publicly available mouse RNA-seq datasets (Kolabas et al. and Evrard et al.) containing neutrophils and pre-neutrophils from blood and multiple bone marrow sites. Variance stabilising transform was applied to the bulk (Evrard) and pseudobulked (our meningeal data and Kolabas) samples (see **Supplementary Methods**). PCA was performed on Kolabas pseudobulk samples using the 20% most variable genes intersected with genes present in all three datasets. VST matrices from our data and Evrard's were quantile-normalized and projected into this PCA space. (a) PCA plot highlighting tissue origin (predominantly PC2) (b) PCA plot highlighting cell types (reflected by PC1). Meningeal pre-neutrophils cluster with pre-neutrophils from Kolabas

and Evrard samples but meningeal neutrophils are most similar to bone marrow immature neutrophils. HC, control (white outlines), SD, social defeat (black outlines).

Results:

“Single cell RNA sequencing (scRNAseq) reveals increased neutrophils in CSD meninges, increased proinflammatory signaling, neutrophil heterogeneity.

“...We integrated our data with two publicly available multi-tissue datasets containing pre-neutrophils and neutrophils from blood and multiple BM locations^{29,31}. When projected onto a shared PCA space, our meningeal neutrophils did not cluster with mature blood neutrophils but instead were most similar to immature BM neutrophils. In contrast, our meningeal pre-neutrophils clustered with the bulk and pseudobulked pre-neutrophil Evrard and Kolabas samples, suggesting successful integration of these three datasets (**Figure S14**).”

Supplementary methods:

“Comparison with public neutrophil transcriptomic datasets. To compare our neutrophil transcriptional data to neutrophils from other tissues, we reprocessed two public mouse RNAseq datasets that each included neutrophils acquired from multiple tissues (Evrard et al., 2018; Kolabas et al., 2023). For the public Kolabas dataset (scRNAseq), we summed raw cell counts per tissue and cell type to create pseudobulk profiles for neutrophils and preneutrophils across multiple tissues, retaining pseudobulk profiles including at least 25 cells, and excluding two outlier samples. Variance stabilizing transformation (VST) implemented in the DESeq2 package (Love et al., 2014) was applied to the pseudobulked counts. Samples included bone marrow from femur, humerus, pelvis, scapula, skull, and vertebra. Our meningeal neutrophil and pre-neutrophil data were pseudobulked and transformed in the same way. For the public Evrard dataset (bulk sorted cell RNAseq), we selected control samples representing mature, immature and preneutrophils from femur, plus mature neutrophils from blood, then applied DESeq2 VST. PCA was performed on the Kolabas dataset using the intersect between the 20% most highly variable genes in the Kolabas dataset and the genes present in all three datasets. VST matrices from the stress and Evrard datasets were quantile-normalized and projected into the Kolabas PCA space to enable direct comparison (**Figure S14**).”

Second, we generated a correlation matrix for neutrophil levels (identified by flow cytometry) across several different tissues collected from individual C57/6J WT mice. Using unsupervised, hierarchical clustering of the data, we found that nonvascular–i.e., neutrophils negative for the the presence of an intravascularly injected label, or iv--meningeal neutrophil levels were significantly correlated with bone marrow neutrophil levels. Conversely, meningeal neutrophils positive for the intravascularly-

injected dye (iv+) were significantly correlated with blood and spleen neutrophil levels.

This clustering pattern was distinct from monocytes, which derive from the same progenitor bone marrow cells as neutrophils, are also GFP+ in LysM mice, and show CSD stress-related increases across tissues. New sections have been added to the figures (**Figures 3J-K, S11A**), figure legends, methods, results, and discussion as follows, and accompanying code provided in the paper github repository:

Figure 3 ... “j) Flow cytometry shows widespread increase in neutrophil levels throughout the body following CSD stress in C57BL/6J mice ($n_{HC} = 7-8$, $n_{CSD} = 9$). k) Unsupervised hierarchical clustering of data from (j) shows significant correlations between BM and iv⁻ meningeal neutrophils, but not with blood...”

“Figure S11: No evidence to support skull trafficking of monocytes... in C57BL/6J mice. a) Flow cytometry shows a main effect of stress on monocyte levels throughout the body, though only the spleen was significant with multiple comparisons corrections (main effect of group, $***p < 0.001$, $F_{(1,15)} = 24.7$. Šídák’s post hoc: *spleen iv⁺*, $*p < 0.05$, $t = 4.1$). Supervised hierarchical clustering, based on neutrophil clustering pattern, shows iv⁺ meningeal, blood, and spleen monocytes significantly correlate with each other, but iv⁻ meningeal and bone marrow monocyte levels do not. This is in contrast to the pattern seen in neutrophils, shown in **Figure 3K...**”

Results:

CSD increases neutrophil trafficking between skull BM and meninges.

“...We next tested the extent to which elevated meningeal neutrophil numbers reflect neutrophil levels in peripheral tissues, which were also increased by CSD stress in C57BL/6J mice (**Figure 3J**: **** $p < 0.0001$, $F(1,78) = 62.7$. post hoc, $t = 7.9$, **** $p < 0.0001$). Pearson correlations and unsupervised hierarchical clustering across tissues revealed that iv-meningeal neutrophil levels were most similar to skull (**FDR < 0.001 , $r = 0.83$, $t(15) = 5.81$) and tibial (**FDR < 0.01 , $r = 0.75$, $t(15) = 4.36$) BM neutrophils, and showed no relationship with blood neutrophils (**Figure 3K**). In contrast, iv+ meningeal neutrophils correlated with blood neutrophils (*FDR < 0.05 , $r = 0.72$, $t(15) = 3.97$).

We also examined relationships between iv- meningeal monocyte and lymphocyte levels with the corresponding cells in other tissues but did not find any significant relationship (**Figure S11**). These data support a role for skull-to-meninges communication in CSD, and highlight the unique composition of the meningeal immune environment compared to other tissues.”

Supplemental methods:

Additional statistics. ... “To assess relationships between immune cells in different tissues, a Pearson correlation matrix was first generated with the function `Hmisc::rcorr`. Unsupervised hierarchical clustering was then performed using the Lance-Williams dissimilarity formula (“`hclust`” in R base stats) and the `gplots::heatmap.2` function for plotting. The unsupervised clustering order for neutrophils was used to force clustering of monocytes and lymphocytes. Bonferroni corrections were applied to control the type I error rate; significant surviving associations were added to the heatmap manually, indicated with an asterisk.”

Both of these new lines of evidence have been included in an updated version of the discussion:

Discussion:

“...Our data lend further support to the extant literature suggesting direct trafficking of immune cells from the skull BM to the meninges^{26–29}. To our knowledge, this is the first evidence for such a phenomenon under conditions of psychosocial stress.

Several independent lines of evidence from our study bolster support for this conclusion. First, in cleared skull tissue, we observed more LysM-GFP+ myeloid cells present in skull-to-meninges vascular channels of CSD mice. While LysM-GFP expression is not restricted to neutrophils, unsupervised hierarchical clustering of flow cytometry data using the

neutrophil-specific Ly6G antibody revealed that nonvascular meningeal neutrophil levels closely mirrored those of skull BM, but not blood. Further, integration with public datasets^{29,31} showed that meningeal neutrophils have a transcriptional profile distinct from mature neutrophils in blood, and are more similar to immature BM neutrophils.”

4/2) Could the other validate the contribution from one versus the other via tagging of the skull bone marrow or blocking blood extravasation signaling?

We agree it would be useful to perform tagging or blocking experiments of this nature. However, it is beyond the scope of the current paper to conduct a new experimental study of the requisite scale and complexity. We address this further in 4/3, below.

4/3) The authors are using systemic blockade of Type I interferon, which shows reduction in behavioral effects of chronic social stress, and associate it with neutrophils recruitment in the meninges as a mechanism. Previous literature demonstrated that removal of IFNAR from microglia was sufficient to alleviate behavioral symptoms of chronic mild stress. While this reviewer understand that generation of cell specific KO animals may be outside of the scope of the current manuscript, the authors should, at least, demonstrated that depletion of meningeal neutrophils has an effect on chronic social stress associated behaviors.

We appreciate the reviewer’s recognition that generating a neutrophil-specific IFNAR knockout lies beyond the scope of this manuscript. Based on current evidence, we do not believe the observed effects involve microglia, as the peripherally administered IFNAR antibody used in our study has been reported not to cross the blood–brain barrier (Pinto et al., 2011, doi: 10.1371/journal.ppat.1002407).

We agree neutrophil depletion experiments would be interesting. We performed these experiments using two depletion strategies: anti-GR1 (using a method suggested in personal communication by Dorian McGavern, NINDS) and anti-Ly6G (method reported in Faget et al., bioRxiv 2018).

The prolonged GR1 or Ly6G knockdown necessary to test the impact of neutrophil depletion over the course of the stress paradigm was associated with animal welfare concerns. We have updated the limitations section to address these important points:

We have highlighted these issues in the limitations sections as follows:

“Other studies have highlighted IFN-I responses in brain cells during stress^{79,80}. Given that the MAR1-5A3 IFNAR antibody clone we used cannot cross the blood brain barrier (BBB)⁸¹, the mediating effects of anti-IFNAR treatment on the negative sequelae of CSD stress in our model likely do not involve cells within the brain parenchyma. The cellular source of stress-

related IFN-I, and the relative contributions to behavior of IFNAR signaling in different cell types, requires further investigation.

Systemic knockdown to test the impact of neutrophil depletion over the course of the stress paradigm was associated with animal welfare concerns. Specifically, wound-healing deficits resulting from neutrophil depletion are poorly compatible with the CSD mode that results in fight-related wounds. Because of these welfare concerns, more localized strategies targeting the skull BM will be useful in future follow-up studies....”

4/4) Does the single cell RNA seq provide any information as to why neutrophils presence in the meninges may alter chronic social stress associated behavior?

We agree that the mechanism by which neutrophils affect neuronal function and stress-associated behaviour is vital to understand. However, without paired brain single cell data from the same stress model, it is not feasible to infer predicted cell-cell interactions that might mediate the immune-neural communication. Moreover, many of the secreted cytokines potentially mediating immune-neural communication are not detectable with 10X Chromium v2 technology. Given immune-neural communication was not the focus of our paper, and our data are consistent with multiple possible mechanisms, we do not put forward a specific hypothesized mechanism. We have added discussion of these points to the legend of the schematic **Figure S25** as below:

“Figure S25: Potential mechanisms of neutrophil trafficking to the meninges and neutrophil-brain communication. **a)** Class 3 semaphorins (SEMA3) have chemorepellant properties that prevent migration of neutrophils across dural lymphatics into the leptomeninges at arachnoid cuff exit (ACE) points (Smyth et al., 2024). Depletion of SEMA3 family expression leads to accumulation of neutrophils in subarachnoid space (SAS) where neutrophil-released factors could influence brain function. **b)** Microarray analysis of SEMA3 family members in bulk meningeal tissue. *Sema3b* expression was reduced in CSD

mice compared to HC (**Figure 7C**: LFC = -0.30, *unadjusted $P = 0.050$), which may permit neutrophil entry into the SAS ($n = 7$ per group). Communication between meningeal neutrophils and brain-resident cells may occur via direct cell–cell contact or through the secretion of bioactive molecules that influence neural, endothelial, or glial function. Notably, CSD meningeal neutrophils exhibited broad upregulation of secreted factors with potential neuroimmune activity, including *Lgals3* (Galectin-3), the neuroprotective factor *Sipi*, the alarmins *S100a8/S100a9*, and *Lcn2*. HC = home cage, CSD = chronic social defeat, DBC = dural border cell, ABC = arachnoid barrier cell.”

4/5) The authors are focusing on the migration related pathway in their analysis, but it feels like there is a missed opportunity to better understand what could be the functional consequences of meningeal neutrophilia. The authors show that blockade of Type I interferon normalize number of meningeal neutrophils. Have the authors looked if that decreased the migration of neutrophils through the skull bone marrow channel ?

We agree that these novel findings raise many intriguing questions for future investigation. While we do not have the dynamic imaging data necessary to address the reviewer’s question regarding neutrophil migration through bone marrow channels following anti-IFNAR treatment, we have emphasized this as a promising direction for future research in the Discussion section as follows:

“Although we did not assess IFNAR⁺ neutrophil staining in the meninges or in cleared skull tissue directly, our data are consistent with IFNAR-mediated trafficking of neutrophils from the skull BM to meninges. We demonstrated a skull BM-specific decrease in IFNAR⁺ neutrophils following CSD stress coupled with transcriptomic evidence of increased IFNAR signaling in meningeal neutrophils. IFN-I depletion in C57BL/6J mice normalized CSD-related meningeal neutrophil levels. While replication of these effects in *LysM^{gfp/+}* mice presented additional complexity (see **Limitations**), CSD+anti-IFNAR–treated mice showed improved anhedonic behaviour...”

4/6) Chemorepellent factors: In figure 7C, the authors extract from microarray data the down regulation of *Sema3b* in the meninges and use that to conclude that chemorepellent are decreased which would increase neutrophils infiltration. These data comes out of nowhere in the progression of the paper. Either the authors should use that data, and investigate deeper, the circulation of neutrophils form the blood and the skull bone marrow, or that data should be removed from the manuscript.

We agree that these results are not central to our main conclusions and have removed them to supplemental information where they may be of interest to readers seeking possible molecular mechanisms of neutrophil meningeal migration suggested by our results. Revised figure shown in response to **4/4**, above.

4/7) Throughout the manuscript, the authors are using the term **parenchyma** to talk about neutrophils that are "away" from blood vessels in the meninges. That nomenclature can be misleading as **parenchyma** usually refers to the inside of the brain for most neuroscientist. An change of nomenclature or a clearer definition of what the authors are referring to would help the readers appreciate the message that the authors are putting forward.

We have changed all instances of 'meningeal parenchyma', referring to meningeal tissue "away from blood vessels", to 'non-vascular meninges'. Instances of 'parenchyma' referring to brain tissue remain as-is.

4/8) It is unclear from the manuscript what considered a mouse to have "marked" or not. In some experiment (Type I interferon blockade), the mice are split depending on if they marked or not, while other experiments/group, the authors didn't. There would need to be some consistency to assess if the difference are solely associated with the "marked" status rather than the treatment itself. It is particularly visible in the Type I interferon blockade experiment where the treated mice almost completely segregate depending on their "marked" status. What do the authors hypothesis to have such a strong bimodal response in the treated group?

We agree that clarification of this phenomenon would improve the paper. We have now included in **Figure 1B** images that make more explicit what 'marking' looks like, along with an updated figure legend (relevant text in bold):

Figure 1... "CSD results in expected **sexual (b)** and social (c) **anhedonia in USM** and SI tests, respectively. **In (b), pink arrowheads indicate female urine, pipetted onto the blotting paper, and blue arrowheads point to representative urine scent marks from male test mice.**"

We have also provided more experimental data about the USM behavioral subscales (i.e., total area marked, area marked near female scent, preference for female vs general marking), and how this relates to 'resiliency' vs 'susceptibility' in the social interaction test, in new **Figures S1** and **S8**:

“Figure S1: Behavioral testing in C57BL/6J mice following chronic social defeat (CSD) stress shows expected phenotype (NB: these animals were used for flow cytometry analyses shown in **Figures 1F-H & S3-4**). Behavioral testing began after 10 days of defeat stress; results are shown with all CSD animals combined (**a**), or with CSD animals stratified into susceptible vs resilient based on their performance in the social interaction (SI) test (**b**). *Top row:* For the SI test—which assesses social anhedonia as a depressive-like phenotype—SI quotient was calculated based on time spent engaging with the social stimulus divided by time spent engaging with the non-social stimulus (see **Figure 1C** for visualization). **a**) Mann Whitney test: $p = 0.075$, $U = 378$. $n_{\text{HC}} = 34$, $n_{\text{CSD}} = 30$. **b**) Kruskal-Wallis test: **** $P < 0.0001$, $H = 21$; Dunn’s post-hoc: ** $p_{\text{HCvCSD-S}} < 0.01$, $Z_{\text{HCvCSD-S}} = 3.5$; **** $p_{\text{CSD-SvCSD-R}} < 0.0001$, $Z_{\text{CSD-SvCSD-R}} = 4.2$. $n_{\text{HC}} = 34$, $n_{\text{CSD-S}} = 20$, $n_{\text{CSD-R}} = 10$. Social approaches represent the number of times the test mouse approached the social stimulus for sniffing. **a**) See **Figure 1C**. **b**) Kruskal-Wallis test: ** $P < 0.01$, $H = 12$; Dunn’s post-hoc: ** $p_{\text{HCvCSD-S}} < 0.01$, $Z_{\text{HCvCSD-S}} = 3.3$; * $p_{\text{CSD-SvCSD-R}} < 0.05$, $Z_{\text{CSD-SvCSD-R}} = 2.4$. $n_{\text{HC}} = 34$, $n_{\text{CSD-S}} = 20$, $n_{\text{CSD-R}} = 10$. Time spent engaging with the social stimulus [in (**b**), Kruskal-Wallis test: *** $P < 0.001$, $H = 15$; Dunn’s post-hoc: * $p_{\text{HCvCSD-S}} < 0.05$, $Z_{\text{HCvCSD-S}} = 2.9$; ** $p_{\text{CSD-SvCSD-R}} < 0.01$, $Z_{\text{CSD-SvCSD-R}} = 3.6$. $n_{\text{HC}} = 34$, $n_{\text{CSD-S}} = 20$, $n_{\text{CSD-R}} = 10$) and with the non-social stimulus, which comprise the SI quotient, are next. *Second row:* The urine scent marking (USM) task was used to assess sexual anhedonia, or depressive-like behavior; marking indicates engagement with a hedonic (i.e., female scent) stimulus. Data were stratified by task response – marking (+mark), or no marks present (-mark— see **Figure 1B**). **a**) Fisher’s exact test, **** $p < 0.0001$. $n_{\text{HC}} = 26$, $n_{\text{CSD}} = 15$. **b**) Fisher’s exact test, * $p < 0.05$. $n_{\text{HC}} = 17$, $n_{\text{CSD-S}} = 2$, $n_{\text{CSD-R}} = 4$. Total area for test subject urine marking is shown [(**a**) Mann Whitney test: **** $p < 0.0001$, $U = 14.5$. $n_{\text{HC}} = 16$, $n_{\text{CSD}} = 13$], along with marks made in close proximity to the female scent [(**a**) Mann Whitney test: *** $p < 0.001$, $U = 29$. $n_{\text{HC}} = 16$, $n_{\text{CSD}} = 13$], %Preference indicates preference for female scent, which is derived by dividing the marking area near the female urine spot by total marking area (**Figure 1B**)...”

“Figure S8: *LysM^{gfp/+}* mice recapitulate expected behavioral phenotype following CSD stress. Note these animals were also used to generate data for **Figures 3C-F, S9, and S10A-D**. Behavioral testing began after 10 days of defeat stress; results are shown with all CSD animals combined (**a**), or with CSD animals stratified into susceptible vs resilient based on their performance in the social interaction (SI) test (**b**). By our non-formalized metrics, *LysM* mice were ‘defeated’ normally, and exhibited other signs typical of ‘depressed’ mice: e.g., poor coat quality. However, the social interaction test gave opposite to expected results. *Top row:* CSD mice spent more time in proximity to the social stimulus overall [(**a**) Mann Whitney test: *** $p < 0.001$, $U = 27$. $n_{HC} = 12$, $n_{CSD} = 18$], though this seems to be driven mostly by CSD-R mice—which represented 50% of the CSD group [(**b**) Kruskal-Wallis test: *** $P < 0.001$, $H = 17.3$; Dunn’s post-hoc: **** $p_{HCvCSD-R} < 0.0001$, $Z_{HCvCSD-R} = 4.2$. $n_{HC} = 12$, $n_{CSD-S} = 9$, $n_{CSD-R} = 9$]. Splitting mice into susceptible vs resilient based on SI quotient scores also did not generate the expected difference between HC and CSD-S [(**b**) Kruskal-Wallis test: ** $P < 0.01$, $H = 12.2$; Dunn’s post-hoc: * $p_{HCvCSD-R} < 0.05$, $Z_{HCvCSD-R} = 2.8$; ** $p_{CSD-SvCSD-R} < 0.01$, $Z_{CSD-SvCSD-R} = 3.3$], but the HC group appeared to be less social than expected (SI scores < 2). This may be influenced by group housing of *LysM^{gfp/+}* mice with their littermates post-weaning, as opposed to C57BL/6J mice that were purchased and pair-housed in cages with dividers upon arrival to our facility, though at this time we cannot explain their behavior further. *Second row:* *LysM^{gfp/+}* mice showed an expected ‘anhedonic’ response to CSD stress in the USM test: marking [(**a**) Fisher’s exact test, *** $p < 0.001$. $n_{HC} = 18$, $n_{CSD} = 26$. (**b**) Fisher’s exact test, ** $p < 0.01$. $n_{HC} = 12$, $n_{CSD-S} = 9$, $n_{CSD-R} = 9$], total urine area [(**a**) Mann Whitney test: *** $p < 0.001$, $U = 76$. $n_{HC} = 18$, $n_{CSD} = 26$. (**b**) Kruskal-Wallis test: * $P < 0.05$, $H = 8.9$; Dunn’s post-hoc: * $p_{HCvCSD-S} < 0.05$, $Z_{HCvCSD-S} = 2.6$; * $p_{HCvCSD-R} < 0.05$, $Z_{HCvCSD-R} = 2.5$. $n_{HC} = 12$, $n_{CSD-S} = 9$, $n_{CSD-R} = 9$], urine area near female scent [(**a**) Mann Whitney test: **** $p < 0.0001$, $U = 64$. $n_{HC} = 18$, $n_{CSD} = 26$. (**b**) Kruskal-Wallis test: ** $P < 0.01$, $H = 9.8$; Dunn’s post-hoc: * $p_{HCvCSD-S} < 0.05$, $Z_{HCvCSD-S} = 2.7$; * $p_{HCvCSD-R} < 0.05$, $Z_{HCvCSD-R} = 2.6$. $n_{HC} = 12$, $n_{CSD-S} = 9$, $n_{CSD-R} = 9$], and preference for female [(**a**) Mann Whitney test: **** $p < 0.0001$, $U = 93$. $n_{HC} = 18$, $n_{CSD} = 26$. (**b**) Kruskal-Wallis test: ** $P < 0.01$, $H = 10.7$; Dunn’s post-hoc: * $p_{HCvCSD-S} < 0.05$, $Z_{HCvCSD-S} = 2.5$; ** $p_{HCvCSD-R} < 0.01$, $Z_{HCvCSD-R} = 3.0$. $n_{HC} = 12$, $n_{CSD-S} = 9$, $n_{CSD-R} = 9$]....”

To address the reviewer's more general question about the bimodal nature of USM marking, we examined further the heterogeneity in HC mice. We found that variability in marking behaviour was associated with meningeal neutrophil levels in HC mice (new **Figures S2D, S8D**):

“Figure S2:…d) HC mice that are ‘anhedonic’, i.e., do not show preference for female in the USM task, have elevated levels of neutrophils in the absence of other stressors (1-way ANOVA, $**p < 0.01$, $F_{(2,42)} = 6.3$. Dunnett’s posthoc test: $*p_{\text{HC+mark v HC-mark}} < 0.05$, $q_{\text{HC+mark v HC-mark}} = 2.7$; $**p_{\text{HC+mark v CSD}} < 0.01$, $q_{\text{HC+mark v CSD}} = 3.2$). ~53% of HC mice failed to mark; compare with **Figure S8D**…”

“Figure S8: … d) HC mice that are ‘anhedonic’, i.e., do not show preference for female in the USM task, do not have elevated levels of LysM-GFP cells in the absence of other stressors, in contrast to C57BL/6J mice (Kruskal-Wallis, $P = 0.063$, $H = 5.5$). ~72% (13/18) of $\text{LysM}^{gfp/+}$ HC mice mark; compare with **Figure S2D**. †square root-transformed to improve normality.”

We have updated the discussion to include these findings:

“...Interestingly, we observed meningeal neutrophilia in otherwise non-stressed HC mice who failed to show preference for a sexually hedonic stimulus. This is consistent with other reports suggesting individual differences in peripheral immunity influence stress susceptibility⁵³.”

4/9) In the introduction, the authors particularly emphasize on their previous work regarding meningeal B cells, and B cells are used throughout the manuscript as a control for their experiment. The authors also suggest a connection between B cells and neutrophils. However, no experiments are made to address this specifically. Refocusing the writing of the manuscript towards neutrophils may help straighten the message the authors are putting forward.

We agree that in the original version of the figures, inclusion of data for other cell types may have distracted from the core focus of the paper on neutrophils. B cell related data has now been moved entirely to supplemental information. This involved reworking of Figures 1 and 6, which have been modified as follows:

Figure 2 (new figure that contains time-course study data that was included in previous version of Figure 1):

Supplemental Figure 6 (contains time course data that was originally presented in Figure 1 of previous version of manuscript):

Supplemental Figure 23 (contains B cell data that was originally presented in Figure 6 of previous version of manuscript):

Reviewer #5 (Remarks to the Author):

Animal models of stress and related disorders show increased blood neutrophilia, though its connection to symptoms or behavior is unclear. Using various techniques, the authors found that chronic, but not acute, social defeat (CSD) stress in mice causes neutrophil accumulation in the meninges, with potential increased trafficking from the skull bone marrow. Blocking type I interferon (IFN-I) signaling systemically protected against the behavioral effects of CSD, suggesting IFN-I as a potential therapeutic target for stress-related disorders.

5/1) The study is very interesting and deserves publication, but one experiment would be really interesting to confirm the causal role of neut in this model using neutrophil depletion.

We thank the reviewer for their positive feedback on the manuscript and support for publication.

We agree neutrophil depletion experiments would be interesting. We performed these experiments using two depletion strategies: anti-GR1 (using a method

suggested in personal communication by Dorian McGavern, NINDS) and anti-Ly6G (method reported in Faget et al., bioRxiv 2018).

The prolonged GR1 or Ly6G knockdown necessary to test the impact of neutrophil depletion over the course of the stress paradigm was associated with animal welfare concerns. We have updated the limitations section to address these important points:

“Systemic knockdown to test the impact of neutrophil depletion over the course of the stress paradigm was associated with animal welfare concerns. Specifically, wound-healing deficits resulting from neutrophil depletion are poorly compatible with the CSD mode that results in fight-related wounds. Because of these welfare concerns, more localized strategies targeting the skull BM will be useful in future follow-up studies, which may also permit opportunities to demonstrate directionality of neutrophil migration, i.e., using intravital imaging.”

Also, here are some minor comments that would improve the manuscript.

5/2) Fig1. Authors should indicate the number of cells, not only the proportion, at least for iv- neutrophils at day 14 in HC versus CSD, to make sure that this population increases (as there is a relative decrease of other populations)

We thank the reviewer for bringing to our attention that the cell count data in the original Figure 1 could be missed. We have edited the figure to highlight presentation of both proportional and absolute cell counts in meningeal tissue and blood. This new version has been added to the main paper as **Figures 1F-H**, replacing what was formerly Figure 1C-D.

5/3) Fig1 could be summarized by fig 1f and 1g, and the rest could be supplemental data.

These comments, in combination with concerns raised in 4/9, above, have prompted us to remove all references to cell types other than neutrophils in the main figures, as we agree their inclusion may detract from the core focus of the paper on neutrophils. B cell related data have now been moved entirely to supplemental information. The extracted part of original Figure 1 now appears in **Figures 2 & Figure S6**, as shown below:

Figure 2 (new figure that contains time-course study data from previous version of Figure 1):

Supplemental Figure 6 (contains time course data showing trends for all cell types that was originally presented in Figure 1):

5/4) Is timepoint 0 the HC?

Yes, timepoint 0 was meant to indicate HC. We agree this was potentially confusing and have edited the time course graphs to show 'HC' instead of '0'; it has been moved into supplemental information (**Figure S6**, shown above) for the reasons given in response to 5/2.

5/5) Is the loss of B cells significant? And the increase in monocytes? Stats should be done for each pop in the main figure, not only in supp data.

Yes, both of these cell types show significant changes with increasing defeat encounters. We thank the reviewer for pointing out the utility in including the full statistics for these graphs; these changes have been incorporated into the graphs for **Figure S6**, shown above in response to 5/2.

5/6) Is the changes in iv+ B cells and neutrophils a reflection of what is in the blood? If so that would mean that there is a strong changes in bone marrow cells, that could be really interesting to analyze (femur versus skull).

We thank the reviewer for this interesting suggestion. We addressed this by generating a correlation matrix for neutrophil, monocyte, or lymphocyte levels (identified by flow cytometry) across several different tissues collected from individual C57/6J WT mice.

Using unsupervised, hierarchical clustering of the data, we found that nonvascular—i.e., neutrophils negative for the the presence of an intravascularly injected label, or iv--meningeal neutrophil levels were significantly correlated with bone marrow neutrophil levels. Conversely, meningeal neutrophils positive for the intravascularly-injected dye (iv+) were significantly correlated with blood and spleen neutrophil levels.

This clustering pattern was distinct from both monocytes—which derive from the same progenitor bone marrow cells as neutrophils, are also GFP+ in LysM mice, and show CSD stress-related increases across tissues— and lymphocytes. New sections have been added to the figures (**Figures 3J-K, S11**), figure legends, methods, results, and discussion as follows, and accompanying code provided in the paper github repository:

Figure 3 ... “. j) Flow cytometry shows widespread increase in neutrophil levels throughout the body following CSD stress in C57BL/6J mice ($n_{HC} = 7-8$, $n_{CSD} = 9$). k) Unsupervised hierarchical clustering of data from (j) shows significant correlations between BM and iv⁻ meningeal neutrophils, but not with blood...”

“Figure S11: Cross-tissue effects of CSD on monocytes and lymphocytes. Unlike for neutrophils (**Figure 3K**), meningeal monocyte and lymphocyte populations do not correlate with their corresponding populations in skull bone marrow in C57BL/6J mice. **a)** Flow cytometry shows a main effect of stress on monocyte levels throughout the body, though only the spleen was significant with multiple comparisons corrections (main effect of group, $***p < 0.001$, $F_{(1,15)} = 24.7$. Šidák’s post hoc: *spleen iv⁺*, $*p < 0.05$, $t = 4.1$). Supervised hierarchical clustering, based on neutrophil clustering pattern, shows *iv⁺* meningeal, blood, and spleen monocytes significantly correlate with each other, but *iv⁻* meningeal and bone marrow monocyte levels do not. This is in contrast to the pattern seen in neutrophils, shown in **Figure 3K**. **b)** To examine inter-tissue relationships between lymphocytes in this data set, which did not include CD3 or CD19 markers, we gated on CD11b⁺;FSC^{lo}; SSC^{lo} cells. As with other studies, we saw decreased lymphocytes (likely

representing B cells) across multiple tissues (main effect of group, **** $p < 0.0001$, $F_{(1,15)} = 85.9$. Šídák's post hoc: *meninges iv*⁺, * $p < 0.05$, $t = 3.5$; *skull iv*⁺, ** $p < 0.01$, $t = 4.9$; *tibia iv*⁺, **** $p < 0.0001$, $t = 8.0$; *blood iv*⁺, ** $p < 0.01$, $t = 4.9$; *spleen iv*⁺, ** $p < 0.01$, $t = 4.5$). Supervised hierarchical clustering, based on neutrophil clustering pattern, shows *iv*⁺ meningeal relationships, but no *iv* meningeal relationships, in contrast to **Figure 3K**. $n_{HC} = 7-8$, $n_{CSD} = 9$. Data shown as mean \pm SEM.”

Results:

CSD increases neutrophil trafficking between skull BM and meninges.

“...We next tested the extent to which elevated meningeal neutrophil numbers reflect neutrophil levels in peripheral tissues, which were also increased by CSD stress in C57BL/6J mice (**Figure 3J**: **** $p < 0.0001$, $F(1,78) = 62.7$. post hoc, $t = 7.9$, **** $p < 0.0001$). Pearson correlations and unsupervised hierarchical clustering across tissues revealed that *iv*-meningeal neutrophil levels were most similar to skull (**FDR < 0.001 , $r = 0.83$, $t(15) = 5.81$) and tibial (**FDR < 0.01 , $r = 0.75$, $t(15) = 4.36$) BM neutrophils, and showed no relationship with blood neutrophils (**Figure 3K**). In contrast, *iv*⁺ meningeal neutrophils correlated with blood neutrophils (*FDR < 0.05 , $r = 0.72$, $t(15) = 3.97$).

We also examined relationships between *iv*-meningeal monocyte and lymphocyte levels with the corresponding cells in other tissues but did not find any significant relationship (**Figure S11**). These data support a role for skull-to-meninges communication in CSD, and highlight the unique composition of the meningeal immune environment compared to other tissues.”

Supplemental methods:

Additional statistics. ... “To assess relationships between immune cells in different tissues, a Pearson correlation matrix was first generated with the function `Hmisc::rcorr`. Unsupervised hierarchical clustering was then performed using the Lance-Williams dissimilarity formula (“`hclust`” in R base stats) and the `gplots::heatmap.2` function for plotting. The unsupervised clustering order for neutrophils was used to force clustering of monocytes and lymphocytes. Bonferroni corrections were applied to control the type I error rate; significant surviving associations were added to the heatmap manually, indicated with an asterisk.”

5/7) Fig2defij: the graphs should display lysM+ cells, not neutrophils, as LysM is positive for neuts, mono, macs. How did they avoid including mono/meningeal macrophages, which are also LysM+?

We have now changed the figures, figure legends and results to indicate the cells are “LysM-GFP”, “LysM+ myeloid cells”.

5/8) Fig3: can the authors represent counts of neutrophils instead?

We did a power analysis using the results from our proportion-based flow cytometry data, which was less variable than the absolute cell count data (a known consequence of the inherent variability in the proportion of total meningeal tissue obtained during meningeal dissections). The relationship between blood neutrophils and open field was 83% powered to detect an effect ($n = 14-15$), with Cohen's $f^2 = 0.67$ and $\alpha = 0.05$. Our absolute count sample size was significantly smaller than this, ($n_{\text{CSD}} = 7$ in blood), so we are not powered for this analysis.

5/9) Can the authors graph the time spent with female odor and time spent in OF in control and CSD mice, and do the stats of those 2 groups? This would assess more clearly whether CSD had an effect as expected. Then for the CSD group only, they should graph correlation between meningeal neuts, meningeal mono and meningeal B cells (and same for blood cells) as Y parameters and time spent (odor then OFT) as X parameter, and check correlation p value.

While we are unable to graph the time spent with female odor in the USM test due to the absence of video recordings, we used an alternative validated method to quantify interest in the social odor following our previously published methodology (Lehmann et al., 2013). Representative data are shown in revised **Figure 1B**, below.

In response to the broader concern about confirming the behavioral impact of CSD, we agree that additional behavioral validation would strengthen the manuscript. We have thus incorporated extensive new behavioral analyses across both mouse strains used in our study. These results are presented in **Figures 1B-E, S1**, and **S8**, with corresponding changes to results and discussion as below:

“Figure 1: Meningeal neutrophils are elevated following chronic social defeat (CSD) stress and are associated with depressive-like behavioral change. **a)** CSD mice were behaviorally tested on days 10-13... CSD results in expected sexual (**b**) and social (**c**) anhedonia in USM and SI tests, respectively. In (**b**), pink arrowheads indicate female urine, pipetted onto the blotting paper, and blue arrowheads point to representative urine scent marks from male test

mice. CSD leads to increased anxiety-like behavior in the (d) OF test and (e) LD box. Additional behavioral data shown in **Figure S1...**"

“Figure S1: Behavioral testing in C57BL/6J mice following chronic social defeat (CSD) stress shows expected phenotype (NB: these animals were used for flow cytometry analyses shown in **Figures 1F-H & S3-4**). Behavioral testing began after 10 days of defeat stress; results are shown with all CSD animals combined (**a**), or with CSD animals stratified into susceptible vs resilient based on their performance in the social interaction (SI) test (**b**). *Top row:* For the SI test—which assesses social anhedonia as a depressive-like phenotype—SI quotient was calculated based on time spent engaging with the social stimulus divided by time spent engaging with the non-social stimulus (see **Figure 1C** for visualization). **a**) Mann Whitney test: $p = 0.075$, $U = 378$. $n_{HC} = 34$, $n_{CSD} = 30$. **b**) Kruskal-Wallis test: $****P < 0.0001$, $H = 21$; Dunn’s post-hoc: $**p_{HCvCSD-S} < 0.01$, $Z_{HCvCSD-S} = 3.5$; $****p_{CSD-SvCSD-R} < 0.0001$, $Z_{CSD-SvCSD-R} = 4.2$. $n_{HC} = 34$, $n_{CSD-S} = 20$, $n_{CSD-R} = 10$. Social approaches represent the number of times the test mouse approached the social stimulus for sniffing. **a**) See **Figure 1C**. **b**) Kruskal-Wallis test: $**P < 0.01$, $H = 12$; Dunn’s post-hoc: $**p_{HCvCSD-S} < 0.01$, $Z_{HCvCSD-S} = 3.3$; $*p_{CSD-SvCSD-R} < 0.05$, $Z_{CSD-SvCSD-R} = 2.4$. $n_{HC} = 34$, $n_{CSD-S} = 20$, $n_{CSD-R} = 10$. Time spent engaging with the social stimulus [in (**b**), Kruskal-Wallis test: $***P < 0.001$, $H = 15$; Dunn’s post-hoc: $*p_{HCvCSD-S} < 0.05$, $Z_{HCvCSD-S} = 2.9$; $**p_{CSD-SvCSD-R} < 0.01$, $Z_{CSD-SvCSD-R} = 3.6$. $n_{HC} = 34$, $n_{CSD-S} = 20$, $n_{CSD-R} = 10$) and with the non-social stimulus, which comprise the SI quotient, are next. *Second row:* The urine scent marking

(USM) task was used to assess sexual anhedonia, or depressive-like behavior; marking indicates engagement with a hedonic (i.e., female scent) stimulus. Data were stratified by task response – marking (+mark), or no marks present (-mark— see **Figure 1B. a**) Fisher's exact test, **** $p < 0.0001$. $n_{HC} = 26$, $n_{CSD} = 15$. **b**) Fisher's exact test, * $p < 0.05$. $n_{HC} = 17$, $n_{CSD-S} = 2$, $n_{CSD-R} = 4$. Total area for test subject urine marking is shown [(**a**) Mann Whitney test: **** $p < 0.0001$, $U = 14.5$. $n_{HC} = 16$, $n_{CSD} = 13$], along with marks made in close proximity to the female scent [(**a**) Mann Whitney test: *** $p < 0.001$, $U = 29$. $n_{HC} = 16$, $n_{CSD} = 13$], %Preference indicates preference for female scent, which is derived by dividing the marking area near the female urine spot by total marking area (**Figure 1B**). *Third row:* The open field (OF) test was used to assess anxiety-like behavior (more willingness to explore the novel arena indicates less anxiousness). **a**) See **Figure 1D b**) Kruskal-Wallis test: **** $P < 0.0001$, $H = 20$; Dunn's post-hoc: **** $p_{HCvCSD-S} < 0.0001$, $Z_{HCvCSD-S} = 4.5$. $n_{HC} = 18$, $n_{CSD-S} = 12$, $n_{CSD-R} = 4$). We also tracked crosses into the center of the arena, time spent in the center of the arena, and number of fecal boli as a crude proxy for stress, though these were not significant. *Bottom row:* The light/dark (LD) box assesses anxiety-like behavior—time spent exploring a brightly lit area as opposed to a covered dark space is quantified. Additionally, the number of crosses made from the dark side to the light side shows significant differences between groups: **a**) See **Figure 1E. b**) Kruskal-Wallis test: ** $P < 0.01$, $H = 11$; Dunn's post-hoc: ** $p_{HCvCSD-S} < 0.01$, $Z_{HCvCSD-S} = 3.4$. $n_{HC} = 11$, $n_{CSD-S} = 10$, $n_{CSD-R} = 3$). Data shown as mean \pm SEM."

Results (emphasis added)

"CSD causes expected depressive- and anxiety-like behavioral changes.

We first confirmed that our well-established CSD paradigm^{22–25} elicits expected depressive- and anxiety-like behavior in C57BL/6J WT mice (Figures 1A-E, S1A). Male CSD-treated mice exhibited sexual and social anhedonia in the urine scent marking (USM) test²⁴ (reduced preference for female scent: **** $p < 0.0001$, $U = 30$) and social interaction (SI) test (reduced approaches to a novel CD-1 male: ** $p < 0.01$, $U = 367$), respectively. CSD stress also induced anxiety-like behavior in both the open field (OF) test (less exploration of the novel arena: **** $p < 0.0001$, $U = 75$) and the light/dark (LD) test (fewer crosses to light: *** $p < 0.001$, $U = 38$).

Approximately 30% of the CSD-treated mice were "resilient" to stress, as assessed using an SI quotient ≥ 2 (time socially investigating / time on non-social investigation; **Figure S1B**), comparable to similar versions of the social defeat stress model³⁰.

Methods

“Light/dark box (LD): The LD test uses an acrylic box (50 cm x 25 cm with 30 cm walls) with aversive lighting (~40 lux). Approximately 1/3 of the box is enclosed and dark; an opening allows crossover between light and dark sections. Mice were placed in the light compartment and allowed to move freely for 10 min while the experimenter was out of the room. Time spent in the light compartment and number of crosses between the light and dark sides were scored from video recordings using TopScan. Low scores indicated anxiety-like behavior

Social Interaction (SI): Two perforated acrylic cylinders, one containing an unfamiliar CD-1 mouse and the other empty, were placed in the OF arena with red lighting. The test mouse was placed in the middle of the arena and allowed to freely explore for 10m with the experimenter outside of the room. TopScan was used on captured videos to track approaches to the social stimulus and time spent investigating social vs. non-social stimuli. Fewer social interactions indicated anhedonic behavior.”

“Figure S8: *LysM^{gfp/+}* mice recapitulate expected behavioral phenotype following CSD stress. Note these animals were also used to generate data for **Figures 3C-F, S9, and S10A-D**. Behavioral testing began after 10 days of defeat stress; results are shown with all CSD animals combined (a), or with CSD animals stratified into susceptible vs resilient based on their performance

in the social interaction (SI) test **(b)**. By our non-formalized metrics, LysM mice were 'defeated' normally, and exhibited other signs typical of 'depressed' mice: e.g., poor coat quality. However, the social interaction test gave opposite to expected results. *Top row*: CSD mice spent more time in proximity to the social stimulus overall [(a) Mann Whitney test: $***p < 0.001$, $U = 27$. $n_{HC} = 12$, $n_{CSD} = 18$], though this seems to be driven mostly by CSD-R mice—which represented 50% of the CSD group [(b) Kruskal-Wallis test: $***P < 0.001$, $H = 17.3$; Dunn's post-hoc: $****p_{HCvCSD-R} < 0.0001$, $Z_{HCvCSD-R} = 4.2$. $n_{HC} = 12$, $n_{CSD-S} = 9$, $n_{CSD-R} = 9$]. Splitting mice into susceptible vs resilient based on SI quotient scores also did not generate the expected difference between HC and CSD-S [(b) Kruskal-Wallis test: $**P < 0.01$, $H = 12.2$; Dunn's post-hoc: $*p_{HCvCSD-R} < 0.05$, $Z_{HCvCSD-R} = 2.8$; $**p_{CSD-SvCSD-R} < 0.01$, $Z_{CSD-SvCSD-R} = 3.3$], but the HC group appeared to be less social than expected (SI scores < 2). This may be influenced by group housing of LysM^{gfp/+} mice with their littermates post-weaning, as opposed to C57BL/6J mice that were purchased and pair-housed in cages with dividers upon arrival to our facility, though at this time we cannot explain their behavior further. *Second row*: LysM^{gfp/+} mice showed an expected 'anhedonic' response to CSD stress in the USM test: marking [(a) Fisher's exact test, $***p < 0.001$. $n_{HC} = 18$, $n_{CSD} = 26$. (b) Fisher's exact test, $**p < 0.01$. $n_{HC} = 12$, $n_{CSD-S} = 9$, $n_{CSD-R} = 9$], total urine area [(a) Mann Whitney test: $***p < 0.001$, $U = 76$. $n_{HC} = 18$, $n_{CSD} = 26$. (b) Kruskal-Wallis test: $*P < 0.05$, $H = 8.9$; Dunn's post-hoc: $*p_{HCvCSD-S} < 0.05$, $Z_{HCvCSD-S} = 2.6$; $*p_{HCvCSD-R} < 0.05$, $Z_{HCvCSD-R} = 2.5$. $n_{HC} = 12$, $n_{CSD-S} = 9$, $n_{CSD-R} = 9$], urine area near female scent [(a) Mann Whitney test: $****p < 0.0001$, $U = 64$. $n_{HC} = 18$, $n_{CSD} = 26$. (b) Kruskal-Wallis test: $**P < 0.01$, $H = 9.8$; Dunn's post-hoc: $*p_{HCvCSD-S} < 0.05$, $Z_{HCvCSD-S} = 2.7$; $*p_{HCvCSD-R} < 0.05$, $Z_{HCvCSD-R} = 2.6$. $n_{HC} = 12$, $n_{CSD-S} = 9$, $n_{CSD-R} = 9$], and preference for female [(a) Mann Whitney test: $****p < 0.0001$, $U = 93$. $n_{HC} = 18$, $n_{CSD} = 26$. (b) Kruskal-Wallis test: $**P < 0.01$, $H = 10.7$; Dunn's post-hoc: $*p_{HCvCSD-S} < 0.05$, $Z_{HCvCSD-S} = 2.5$; $**p_{HCvCSD-R} < 0.01$, $Z_{HCvCSD-R} = 3.0$. $n_{HC} = 12$, $n_{CSD-S} = 9$, $n_{CSD-R} = 9$]. *Third row*: LysM^{gfp/+} mice showed an expected anxiety-like behavioral response to CSD in the OF test: novel arena exploration [(a) Mann Whitney test: $*p < 0.05$, $U = 150$. $n_{HC} = 18$, $n_{CSD} = 26$], crosses to center [(a) Mann Whitney test: $**p < 0.01$, $U = 116$. $n_{HC} = 18$, $n_{CSD} = 26$. (b) Kruskal-Wallis test: $*P < 0.05$, $H = 7.2$; Dunn's post-hoc: $*p_{HCvCSD-R} < 0.05$, $Z_{HCvCSD-R} = 2.6$. $n_{HC} = 12$, $n_{CSD-S} = 9$, $n_{CSD-R} = 9$], and time spent in the center of the arena [(a) Mann Whitney test: $*p < 0.05$, $U = 150$. $n_{HC} = 18$, $n_{CSD} = 26$. (b) Kruskal-Wallis test: $P = 0.071$, $H = 5.3$. $n_{HC} = 12$, $n_{CSD-S} = 9$, $n_{CSD-R} = 9$]. *Bottom row*: We did not test both L/D and SI in the same animals, so we cannot examine the effect of stress resiliency on this task. However, LysM^{gfp/+} behavior was also in an unexpected direction on this test. **a)** CSD mice spent significantly more time in the light (Mann Whitney test: $*p < 0.05$, $U = 2$. $n_{HC} = 4$, $n_{CSD} = 6$). This may represent freezing, as crosses to light seemed to be in the expected direction (fewer in CSD)..."

Results:

"Replication of CSD-induced meningeal immune changes.

LysM^{gfp/+} mice, which strongly express GFP in neutrophils (**Figures 3A-B**), recapitulated the core behavioral response to CSD stress, e.g., decreased urine scent marking (USM) preference ($***p < 0.001$) and decreased open field (OF) crosses to center ($**p < 0.01$, $t = 3.27$, $df = 2$) (**Figure S8A-B**)..."

Limitations/Strengths:

"...In general, LysM^{gfp/+} mice behaved similarly, but not identically, to C57BL/6Js following CSD. Increased LysM^{gfp/+} anxiety-like behavior in the OF test following social defeat has previously been reported⁸⁰, though LD data were not reported. Likewise, in a spinal cord injury depression model, LysM^{gfp/+} mice showed the expected anhedonic behavioral phenotype⁸¹. While we cannot fully account for the behavioral differences between LysM^{gfp/+} and C57BL/6J mice, we are reasonably confident that CSD 'works' as expected in these animals."

5/10) Also, timepoint of test and timepoint of neuts % should be clearly indicated.

We appreciate the reviewer raising the issue that what was originally Figure 3 did not include a timeline for behavioral testing and neutrophil collection. The revised **Figure 1** (panel I) now includes the data originally presented in Figure 3. The Figure 1 legend explicitly states days for behavioral testing and tissue collection (emphasis added):

“Figure 1: Meningeal neutrophils are elevated following chronic social defeat (CSD) stress and are associated with depressive-like behavioral change. a) CSD mice were behaviorally tested on days 10-13, and tissue harvested at day 14...i) Elevated neutrophil levels are associated with the negative behavioral sequelae of CSD stress. Plots show standardized effect sizes for neutrophils on behavioral outcomes. Negative relationships between blood and meningeal neutrophils and USM behavior ($n_{HC} = 16$, $n_{CSD} = 17$). CD45^{iv+} neutrophils are significantly associated with OF anxiety-like behavior ($n_{HC} = 21$, $n_{CSD} = 23$). No significant relationships between neutrophils and SI ($n_{HC} = 22$, $n_{CSD} = 26$) or LD ($n_{HC} = 15$, $n_{CSD} = 17$). See **Tables S1-S4 for full statistics; gating strategy shown in **Figure S2...**”**

5/11) The data on neuts and anxiety is also very interesting based on the literature between neuts and depression in humans.

We agree and thank the reviewer for this supportive comment.

5/12) ig4c: adding ID number on the plot would help (as some colors are quite close). Where are the pre-neutrophils?

We thank the reviewer for their suggestion. All populations are now directly labeled in the UMAP (updated **Figure 4C**):

“Figure 4... c) Left: Meningeal scRNAseq reveals 20 distinct immune cell clusters. **Right:** Visualization of recovered cells as a proportion of total cells recovered per group; lavender indicates neutrophils. ($n_{HC} = 8$, $n_{CSD} = 4$; see **Methods**). Plot shown previously²¹.”

5/13) What is the GO of up and down regulated genes (this should be in the main data, not only supp)?

We agree that the GO data should be more clearly visible in the main paper and we have now highlighted a subset of the GO pathways in a new main figure panel (**Figure 4F**):

Figure 4... f) Gene set enrichment analysis (GSEA) revealed several enriched pathways related to cell size and the cytoskeleton (for complete list, see **Figure S15).”**

5/14) What about DEG in other clusters such as mono and B cells?

We have previously published the differentially expressed genes and GO pathways in B cells from this dataset (Lynall and Kigar et al., 2022). The DEGs and GO pathways for monocytes have been added as **Figure S20**, shown below. In addition, we have made the preprocessed single cell data publicly available in a zenodo repository for exploration of DEGs and GO pathways in other cell types: <https://zenodo.org/records/13378961>

“Figure S20: Meningeal monocytes do not show CSD-associated enrichment of interferon signaling. **a)** Volcano plot showing differentially expressed genes (DGE) between CSD and HC in the single cell RNAseq monocyte cluster.

Indicated points represent DGE with LFC > 0.2 and FDR $p < 0.01$. **b)** Gene set enrichment analysis (GSEA) for biological pathways (top) and reactome (bottom).”

5/15) Do we see difference in cell number too for mono and B cells?

Our flow cytometry data demonstrate significant CSD-associated increases in meningeal monocytes and significant decreases in B cells, which we present in **Figure S3A**.

“Figure S3: Chronic stress has multifactorial effects on immune cell subsets in different tissues. Note that these data are the same sample preparations presented in **Figure 1F-H**. Neutrophil data is shown again to facilitate comparisons. **a) Top:** iv⁻ meningeal tissue results (proportion relative to total live CD45⁺ cells). CSD causes an increase in iv⁻ meningeal monocytes (Mann Whitney test: ** $p < 0.01$, $U = 758$, $n_{HC} = 52$, $n_{CSD} = 44$) and a decrease in iv⁻ B cells (Unpaired t test: *** $p < 0.001$, $t = 3.5$, $df = 77$, $n_{HC} = 44$, $n_{CSD} = 35$)...”

5/16) Neutrophils do not usually express MHC2, so it’s surprising that they downregulate it.. was this checked by flow?

We agree this is surprising. To validate this finding, we have now examined MHCII protein expression on iv⁻ meningeal neutrophils. We demonstrate decreased expression of MHCII at the protein level in CSD compared to HC animals, consistent with the single cell sequencing data. This has been added as a new **Figure 4G** with accompanying updates to results and discussion as below:

“Figure 4.... g) Validation of reduced MHCII expression seen in (e) by flow cytometry in CSD iv⁻ meningeal neutrophils ($n_{HC} = 22$, $n_{CSD} = 16$)...”

Results:

“Validation of MHCII downregulation on CSD neutrophils

Examination of differentially expressed genes (DEGs) in the pooled neutrophil cluster showed reduced transcription of *H2-Ab1* and *H2-Aa* (**Figure 4D**)—2 of the 4 genes comprising the major histocompatibility complex, class II (MHCII). Our flow cytometry data showed this was also true at the level of protein expression: in CSD mice, there was a 2.3-fold reduction in the percentage of MHCII⁺ iv⁻ meningeal neutrophils (**Figure 4E**: ****p* < 0.001, *U* = 64.5).”

Discussion:

“Likewise, decreased meningeal neutrophil expression of MHCII genes *H2-Ab1* and *H2-Aa*, which we verified by flow cytometry, may inhibit antigen presentation to CD4⁺ T cells^{46,47}. B cells also present antigen to CD4⁺ T cells in dural-associated lymphoid tissue (DALT) of the meninges⁴⁸. Along with our previous finding that CSD leads to decreased levels of meningeal B cells²¹—reproduced in the current study—these data suggest CSD may be associated with reduced or disrupted antigen presentation to meningeal T cells by multiple cell types.”

5/17) Fig5e: abcdef should be explained in the figure

We agree this required clarification, thank you. These letters refer to neutrophil subclusters: we have now labelled this on the updated figure, and updated the legend as follows:

“Figure 5:… b) Dot plot showing gene expression in each neutrophil subcluster, which represents neutrophil maturation (see Figure S13) for all

genes comprising this pathway. Expression is scaled to mean \pm standard deviation...”

5/18) Fig5d: could the huge difference in IFNAR+ cells in blood versus skull be a bias of the digestion in the skull? They would need controls (eg blood cells processed with digestion). They should quantify CD69 or IFITM in addition, as those are ISGs. And check levels of iv+ and iv- IFNAR+ neuts to make sure it's not a bias of preparation.

We did not use digestion to generate the bone marrow samples, so we are confident that is not the cause of the difference in IFNAR+ neutrophils. Here we provide relevant text in the methods section for reference:

“Peripheral blood preparations: ~500 μ L of venous blood was lysed in 8 mL ACK Lysis Buffer (Cat. # 351-029-721; Quality Biological, Inc.) for 5 min at room temperature, and the reaction stopped by diluting with 7 mL cold HBSS + 0.1% BSA. Cells were pelleted, washed, and prepared for staining.

Skull and tibial BM preparations: To prepare for skull BM extraction, the dorsal calvarium was trimmed to be relatively flat, and meninges were removed under a dissecting microscope. Next, the skull was cut into small bone pieces with scissors in cold HBSS + 0.1% BSA. This entire slurry was transferred to a 70 μ m cell strainer and mashed with the rubber end of a 3 mL syringe for approximately 2 min per sample. Tibia were prepared by first stripping away all tissue from the bone, then cutting the very top such that a 23g syringe needle could be inserted to flush out the BM into a tube of cold HBSS + 0.1% BSA. This was next transferred to a 70 μ m cell strainer and mashed with the rubber end of a 3 mL syringe. For both kinds of BM, the resulting cell suspensions were then pelleted and prepared for flow cytometry.”

5/19) Fig6c: it seems not correct to stratify the CSD IFNAR group (+/-) as the HC are not stratified although they display similar heterogeneity. The authors should do the stats of the overall CSD IFNAR group instead first.

In the updated manuscript, we have revised the relevant panels from **Figure 6**—which now includes data from both CSD+anti-IFNAR(+) and all CSD+anti-IFNAR mice combined—as shown below:

“Figure 6: Type I interferon receptor (IFNAR) blockade may improve CSD-stress related behavioral anhedonia and prevent meningeal neutrophil accumulation.. **b)** USM results from $LysM^{GFP/+}$ mice; see **Figure S22** for comparison with HC+IgG mice and for more detail. CSD+anti-IFNAR treated mice showed behavioral improvement compared to CSD+IgG mice ($n_{CSD+IgG} = 12$, $n_{CSD+IFNAR} = 12$). Hereafter we considered $LysM^{GFP/+}$ mice in two separate groups – those that marked in the USM test (+, indicated with closed circles) and those that didn’t (-, indicated with open circles). **c)** *Left:* CSD+anti-IFNAR(+) mice showed improvement in the OF task for anxiety-like behavior compared to CSD+IgG-treated mice. *Right:* No improvement in OF behavior overall for CSF+anti-IFNAR-treated mice. **d)** *Left:* Representative images for dorsal whole-mount meningeal tissue from CSD+IgG and CSD+anti-IFNAR treated mice. Scale bar = 1 mm. *Right:* Images from HC and CSD mice showing hand-counted LysM-GFP+ cells, normalized to indicated area. **e)** *Top:* Quantification of total nonvascular (>10 μ m away from a blood vessel) meningeal LysM-GFP+ cells shows reduced myeloid cell accumulation in CSD+anti-IFNAR(+) treated mice. *Bottom:* No improvement in LysM-GFP+ cells accumulation overall for CSF+anti-IFNAR-treated mice. †values were square root-transformed to improve normality...”

To address the reviewer’s more general question about the bimodal nature of USM marking, we examined further the heterogeneity in HC mice. We found that variability in marking behaviour was associated with meningeal neutrophil levels in HC mice (new **Figures S2D, S8D**):

“Figure S2:…d) HC mice that are ‘anhedonic’, i.e., do not show preference for female in the USM task, have elevated levels of neutrophils in the absence of other stressors (1-way ANOVA, $**p < 0.01$, $F_{(2,42)} = 6.3$. Dunnett’s posthoc test: $*p_{\text{HC+mark v HC-mark}} < 0.05$, $q_{\text{HC+mark v HC-mark}} = 2.7$; $**p_{\text{HC+mark v CSD}} < 0.01$, $q_{\text{HC+mark v CSD}} = 3.2$). Compare with **Figure S8D**…”

“Figure S8: … d) HC mice that are ‘anhedonic’, i.e., do not show preference for female in the USM task, do not have elevated levels of LysM-GFP cells in the absence of other stressors, in contrast to C57BL/6J mice. See Figure S2D, (Kruskal-Wallis, $P = 0.063$, $H = 5.5$). †square root-transformed to improve normality.”

These results suggest stratifying mice by marking status reveals meaningful differences in individual stress susceptibility. We have updated the discussion to include these findings:

“...Interestingly, we observed meningeal neutrophilia in otherwise non-stressed HC mice who failed to show preference for a sexually hedonic stimulus. This is consistent with other reports suggesting individual differences in peripheral immunity influence stress susceptibility⁵³.”

5/20) What is the source of IFNs?

As far as we are aware, this is an open question across the field. We have added text into the limitations section to reflect this important area for future consideration:

“Though the cellular source of IFN-Is mediating behavioral processes remain unclear, others have previously shown that systemic or cell-type specific knockout of IFNAR in the brain modulates stress-related behavioral outcomes^{79,80} ...”

5/21) Do the authors have similar results on odor and OFT tests (ie restoration of normal behavior) upon neutrophil depletion (see main point above)?

In addition to the expanded text discussing the important limitations of systemic neutrophil depletion shown in **5/1**, above, we provide additional behavioral data from both LysM and C57/6J mice for the anti-IFNAR rescue studies as new **Figures S22 & S23B**:

“Figure S22: Anti-IFNAR treatment in LysM^{gfp/+} mice improves anhedonic behavior in the USM task, but LysM^{gfp/+} mice show strain-related effects of injection stress. **a)** Anti-IFNAR treatment protects against the anhedonic effects of stress. Control (IgG)-antibody treated mice exposed to CSD show expected behavioral deficits compared to HC-IgG mice. *Left:* See main text for stats. *Right:* preference for female scent (Kruskal-Wallis test: $**P < 0.01$, $H = 9.5$; Dunn’s post-hoc: $*p_{HC-IgGvCSD-IgG} < 0.05$, $Z_{HC-IgGvCSD-IgG} = 2.9$; $*p_{CSD-IgGvCSD-IFNAR} < 0.05$, $Z_{CSD-IgGvCSD-IFNAR} = 2.4$; $n = 12$ per group). **b)** *Left:* No effect of injection stress on USM behavior for HC LysM^{gfp/+} mice (Fisher’s exact test: $p = 0.39$; $n_{-inj} = 16$, $n_{+inj} = 11$). *Right:* however, there was an increase in nonvascular, meningeal LysM-GFP⁺ cells with injection stress ($**p < 0.01$, $t = 3.2$, $df = 22$). †square root-transformed. **c)** Comparison of the effects of injection stress on strains. *Left:* Visualization of the percentage of either WT or LysM^{gfp/+} mice that marked in the USM test; all animals shown here at HC. At baseline it appears that LysM^{gfp/+} mice are more likely to mark than C57BL/6J

mice (81% vs 62%). HC+IgG mice from both strains marked less frequently (64% of $LysM^{gfp/+}$ mice vs 0% of C57BL/6J mice); this was not a significant difference in $LysM^{gfp/+}$ mice (Fisher's exact test: $p = 0.39$. C57: $n_{-inj} = 34$, $n_{+inj} = 10$; $LysM$: $n_{-inj} = 16$, $n_{+inj} = 11$). *Middle*: OF behavior not impacted by strain or injection status. *Right*: Significant interaction and status effects on the social interaction test; more anhedonic behavior in both strains with injection stress (2-way ANOVA test: interaction, $*P < 0.05$, $F_{(1,48)} = 4.2$; injection, $*P < 0.05$, $F_{(1,48)} = 7.0$. Tukey's posthoc: C57_{-inj} vs $LysM_{-inj}$ $*p < 0.05$, $q = 4.1$; C57_{-inj} vs C57_{+inj} $**p < 0.01$, $q = 5.2$; C57_{-inj} vs $LysM_{+inj}$ $*p < 0.05$, $q = 4.2$. C57: $n_{-inj} = 22$, $n_{+inj} = 10$; $LysM$: $n_{-inj} = 12$, $n_{+inj} = 8$). **d**) same data presented in **Figure 6B**, but binarized (Fisher's exact test: $*p < 0.05$; $n_{IgG} = 10$, $n_{IFNAR} = 11$). **e**) Anhedonic (SI) and anxiety-like (OF) behavior in $LysM^{gfp/+}$ mice that underwent CSD stress. The top line shows all behavior patterns for all CSD+anti-IFNAR treated mice. The bottom line shows the same, but only shows CSD+anti-IFNAR treated mice that marked in the USM task. Only OF crosses to center were improved by anti-IFNAR treatment; these data are included in **Figure 6C** and are shown here for comparison. SI: $n_{IgG} = 9$, $n_{IFNAR} = 3$ or 9. OF: $n_{IgG} = 12$, $n_{IFNAR} = 6$ or 12. **f**) Reduced nonvascular ($>10\mu m$ from blood vessel) meningeal $LysM$ -GFP+ cells in $LysM^{gfp/+}$ mice that underwent CSD stress but received anti-IFNAR treatment. The top line shows meningeal handcount data for all all CSD+anti-IFNAR treated mice. The bottom line shows the same but only CSD+anti-IFNAR treated mice that marked in the USM task are included. These data are included in **Figures 6D-E** and are shown here for comparison. No other meningeal subpopulations were significant. *Left to right*: total meningeal $LysM$ -GFP+ cells; nonvascular ($>10\mu m$ from blood vessel) meningeal $LysM$ -GFP+ cells; abluminal ($\leq 10\mu m$ away from a blood vessel) meningeal $LysM$ -GFP+ cells; intravascular meningeal $LysM$ -GFP+ cells. \dagger square root -transformed, \ddagger natural log-transformed values to improve normality. $n_{IgG} = 9$, $n_{IFNAR} = 4$ or 10. Data shown as mean \pm SEM."

b

No urine scent marking in any groups

“Figure S23: C57BL/6J mice do not show behavioral rescue with anti-IFNAR treatment, despite meningeal ‘rescue’ of neutrophils...**b) Top:** Social interaction (SI). There was a trend for an effect of anti-IFNAR treatment on social approaches (main effect of treatment, $P = 0.091$, $F_{(1,26)} = 3.1$), though nothing approached significance in post-hoc testing. No other statistical tests approached significance for SI. **Bottom:** Open Field (OF). There was a significant effect of anti-IFNAR treatment on OF time in center, where in both HC and CSD groups, time in the center increased, indicative of less anxious-like behavior (main effect of treatment, $**P < 0.01$, $F_{(1,27)} = 11.1$. Tukey’s post-hoc: $*p_{\text{IgG-HC v IFNAR-CSD}} < 0.05$, $q_{\text{IgG-HC v IFNAR-CSD}} = 4.2$; $*p_{\text{IgG-CSD v IFNAR-CSD}} < 0.05$, $q_{\text{IgG-CSD v IFNAR-CSD}} = 4.7$). IFNAR treatment had no improving effect on novel arena exploration; both CSD groups explored less than the HC groups (main effect of group, $**P < 0.01$, $F_{(1,27)} = 11.7$. Tukey’s post-hoc: $*p_{\text{IgG-HC v IgG-CSD}} < 0.05$, $q_{\text{IgG-HC v IgG-CSD}} = 2.6$; $**p_{\text{IgG-CSD v IFNAR-HC}} < 0.01$, $q_{\text{IgG-CSD v IFNAR-HC}} = 3.1$; $*p_{\text{IFNAR-HC v IFNAR-CSD}} < 0.05$, $q_{\text{IFNAR-HC v IFNAR-CSD}} = 2.2$)...”

Results (emphasis added):

“IFNAR blockade improves the behavioral response to CSD stress.

We assessed whether peripheral blockade of IFN-I signaling rescues the negative behavioral sequelae associated with CSD stress by administering an IFNAR-blocking antibody to $\text{LysM}^{\text{GFP/+}}$ mice (**Figure 6A**). As expected, in the USM test for anhedonia, more HC+control antibody (IgG) mice marked compared to the CSD+IgG group (**Figure S22A**: $**p < 0.01$, $\chi^2 = 9.5$; post hoc $*p < 0.05$). Anti-IFNAR treatment rescued the CSD phenotype, with no difference in marking frequency between HC+IgG and CSD+IFNAR groups. **For other examined parameters, HC animals appeared ‘stressed’, which we determined was influenced by repeated injections (Figures S22B-C, S23).** We therefore examined the CSD group alone.

In the USM task, anti-IFNAR treatment protected mice from the depressive-like effects of CSD (**Figure 6B**: (* $p < 0.05$, $U = 36$). However, USM data for the CSD+anti-IFNAR group was non-normally distributed, with distinct groups of animals that marked (+) or did not mark (-) in the test (**Figure S22D**). We therefore treated these animals separately in our remaining analyses. There was a significant difference in the OF task between CSD+IgG and CSD+anti-IFNAR(+) mice, where CSD+IgG animals were more anxious (**Figure 6C**: * $P < 0.05$, $H = 9.1$; post hoc: * $p < 0.05$, $Z = 2.8$). There were no other significant differences in behavior between groups (**Figure S22E**).

IFNAR blockade attenuates neutrophil migration into, but not B cell egress from, meningeal tissue in CSD stressed mice.

...Like $LysM^{gfp/+}$ mice, C57BL/6J mice were behaviorally impacted by repeated injection stress (**Figures S22C, S23B, S24A**). Unlike $LysM^{gfp/+}$ mice, the effects of injection stress had no impact on meningeal myeloid cell levels (**Figures S24B**)."

Discussion (emphasis added):

"It is not presently clear how meningeal neutrophils influence behavior given they do not appear to enter the brain parenchyma. Further exploration of the pathways by which neutrophils enter the meninges and exert their effects on brain tissue is merited...

... IFN-I depletion in C57BL/6J mice normalized CSD-related meningeal neutrophil levels. While replication of these effects in $LysM^{gfp/+}$ mice presented additional complexity (see **Limitations**), CSD+anti-IFNAR–treated mice showed improved anhedonic behavior. Interestingly, we observed meningeal neutrophilia in otherwise non-stressed HC mice who failed to show preference for a sexually hedonic stimulus. This is consistent with reports suggesting individual differences in peripheral immunity influence stress susceptibility⁵³."

Limitations and strengths:

"Other studies have highlighted IFN-I responses in brain cells during stress^{79,80}. Given that the MAR1-5A3 IFNAR antibody clone we used cannot cross the blood brain barrier (BBB)⁸¹, the mediating effects of anti-IFNAR treatment on the negative sequelae of CSD stress in our model likely do not involve cells within the brain parenchyma. The cellular source of stress-related IFN-I, and the relative contributions to behavior of IFNAR signaling in different cell types, requires further investigation."

5/22) Is there BBB leakage (apparently yes based on previous studies from the team)?

As noted by the reviewer, BBB leakage has previously been demonstrated by our lab in this model (Lehmann et al., Sci Rep 2018). The methods required for BBB leakage detection are incompatible with the methods required for meningeal flow cytometry, so we do not have data showing both BBB leakage and meningeal neutrophilia from the same animals. We have added some discussion of this point as a limitation of the study (new text in bold):

“Another possibility is that chronic release of neutrophils into the bloodstream via repeated exposure to stress^{15,16} drives formation of microhemorrhages, **though we did not test this directly given our focus on the meninges.**”

5/23) What attracts neutrophils in the meninges? IFNs?

Based on our single cell sequencing data, and evidence from the IFNAR antibody treatment, this is what we believe, yes. We did not see changes in the meningeal expression levels of CXCL1, CXCL2, or CXCL12, all of which influence neutrophil chemotaxis, in either the single cell data or by microarray analysis. Determining the cellular source of stress-associated IFNs is an open challenge for the field - see response to **5/20**, above.

5/24) What about male versus female behavior?

We agree this is an important question, and have added the following new text to the limitations section of the discussion:

“A major limitation of this and similar studies is the exclusive use of adult male mice. Future extension of this investigation to females will be crucial, especially as depression disproportionately affects women⁶⁹. In addition, females may be more acutely susceptible to BBB permeability after stress induction⁷⁰⁻⁷² and exhibit a more robust anti-viral IFN-I response^{73,74}. These mechanisms could lead to sex-specific vulnerabilities to meningeal neutrophilia and associated behavioral outcomes.”

5/25) ‘meningeal neutrophilia’ is confusing, as neutrophilia refers to the blood.

Thank you for pointing out the potential confusion regarding use of the word ‘neutrophilia’. We have modified the abstract accordingly:

“We find that chronic, but not acute, stress causes meningeal neutrophil accumulation—i.e., **“meningeal neutrophilia”**—and CSD increases neutrophil trafficking in vascular channels emanating from skull bone marrow (BM).”

5/26) If ‘pathogenic’ neutrophils come from the marrow, what do they author think the role/consequence/origin of blood neutrophils are? (cf 1st sentence of abstract) Also, directionality is not proven and accumulation of neutrophils derived from the vasculature could also traffick upwards to the skull.

The reviewer is correct that we have not directly demonstrated directionality—i.e., skull to meningeal trafficking—although our indirect evidence supports this conclusion. As noted in **5/1** we have updated the limitations section to address these important points (shown here again for reference, emphasis added):

“Systemic knockdown to test the impact of neutrophil depletion over the course of the stress paradigm was associated with animal welfare concerns. Specifically, wound-healing deficits resulting from neutrophil depletion are poorly compatible with the CSD mode that results in fight-related wounds. Because of these welfare concerns, more localized strategies targeting the skull BM will be useful in future follow-up studies, which may **also permit opportunities to demonstrate directionality of neutrophil migration, i.e., using intravital imaging.**”

In addition to the new data shown in response to **5/6**, we also provide new bioinformatics analyses comparing our single cell sequencing data with publicly available datasets from neutrophils in skull, blood, and other bone marrow locations (Evrard 2018, Kolabas 2023). This analysis showed that meningeal neutrophils are transcriptionally distinct from blood and share similarities with bone marrow. New sections have been added to the figures (**Figure S14**), figure legends, methods, results, and discussion as follows, and accompanying code provided in the paper github repository:

Figure S14 Principal Component Analysis (PCA) plots visualize the transcriptomic similarity of neutrophil and preneutrophil populations across tissues (blood, meninges and various bone marrow locations), cell types (neutrophils and preneutrophils), condition (HC, control and CSD, stress), and dataset. We integrated our meningeal dataset with two publicly available mouse RNA-seq datasets (Kolabas et al. and Evrard et al.) containing neutrophils and pre-neutrophils from blood and multiple bone marrow sites. Variance stabilising transform was applied to the bulk (Evrard) and pseudobulked (our meningeal data and Kolabas) samples (see **Supplementary Methods**). PCA was performed on Kolabas pseudobulk samples using the 20% most variable genes intersected with genes present in all three datasets. VST matrices from our data and Evrard’s were quantile-normalized and projected into this PCA space. (a) PCA plot highlighting tissue origin (predominantly PC2) (b) PCA plot highlighting cell types (reflected by PC1). Meningeal pre-neutrophils cluster with pre-neutrophils from Kolabas

and Evrard samples but meningeal neutrophils are most similar to bone marrow immature neutrophils. HC, control (white outlines), SD, social defeat (black outlines).

Results:

“Single cell RNA sequencing (scRNAseq) reveals increased neutrophils in CSD meninges, increased proinflammatory signaling, neutrophil heterogeneity.

“...We integrated our data with two publicly available multi-tissue datasets containing pre-neutrophils and neutrophils from blood and multiple BM locations^{29,31}. When projected onto a shared PCA space, our meningeal neutrophils did not cluster with mature blood neutrophils but instead were most similar to immature BM neutrophils. In contrast, our meningeal pre-neutrophils clustered with the bulk and pseudobulked pre-neutrophil Evrard and Kolabas samples, suggesting successful integration of these three datasets (**Figure S14**).”

Supplementary methods:

“Comparison with public neutrophil transcriptomic datasets. To compare our neutrophil transcriptional data to neutrophils from other tissues, we reprocessed two public mouse RNAseq datasets that each included neutrophils acquired from multiple tissues (Evrard et al., 2018; Kolabas et al., 2023). For the public Kolabas dataset (scRNAseq), we summed raw cell counts per tissue and cell type to create pseudobulk profiles for neutrophils and preneutrophils across multiple tissues, retaining pseudobulk profiles including at least 25 cells, and excluding two outlier samples. Variance stabilizing transformation (VST) implemented in the DESeq2 package (Love et al., 2014) was applied to the pseudobulked counts. Samples included bone marrow from femur, humerus, pelvis, scapula, skull, and vertebra. Our meningeal neutrophil and pre-neutrophil data were pseudobulked and transformed in the same way. For the public Evrard dataset (bulk sorted cell RNAseq), we selected control samples representing mature, immature and preneutrophils from femur, plus mature neutrophils from blood, then applied DESeq2 VST. PCA was performed on the Kolabas dataset using the intersect between the 20% most highly variable genes in the Kolabas dataset and the genes present in all three datasets. VST matrices from the stress and Evrard datasets were quantile-normalized and projected into the Kolabas PCA space to enable direct comparison (**Figure S14**).”

These new data, combined with the data shown in **5/6**, above, are summarized in the following updates to the discussion:

“...Our data lend further support to the extant literature suggesting direct trafficking of immune cells from the skull BM to the meninges^{26–29}. To our knowledge, this is the first evidence for such a phenomenon under conditions of psychosocial stress.

Several independent lines of evidence from our study bolster support for this conclusion. First, in cleared skull tissue, we observed more LysM-GFP⁺ myeloid cells present in skull-to-meninges vascular channels of CSD mice. While LysM-GFP expression is not restricted to neutrophils, unsupervised hierarchical clustering of flow cytometry data using the neutrophil-specific Ly6G antibody revealed that nonvascular meningeal neutrophil levels closely mirrored those of skull BM, but not blood. Further, integration with public datasets^{29,31} showed that meningeal neutrophils have a transcriptional profile distinct from mature neutrophils in blood, and are more similar to immature BM neutrophils.”

5/27) What is the phenotype of neuts in the meninges (IFNAR+?) in HC and CSD mice?

The reviewer raises an excellent question. We were unable to directly assay IFNAR⁺ staining on meningeal neutrophils and address this limitation in the revised discussion:

“Although we did not assess IFNAR⁺ neutrophil staining in the meninges or in cleared skull tissue directly, our data are consistent with IFNAR-mediated trafficking of neutrophils from the skull BM to meninges. We demonstrated a skull BM-specific decrease in IFNAR⁺ neutrophils following CSD stress coupled with transcriptomic evidence of increased IFNAR signaling in meningeal neutrophils.”

5/28) What is the consequence of IFNAR blockade on blood neuts?

We agree with the reviewer that the IFNAR blockade effect on neutrophils should be more clearly visible in the main paper and we have now highlighted these data in a new main figure panel (**Figure 6F**):

“**Figure 6:...**f) In C57BL/6J wild type mice there was an IFNAR-mediated rescue in meningeal neutrophil accumulation following CSD stress, with expected differences in IgG control groups ($n_{\text{CSD+IFNAR}} = 7$, otherwise $n = 8$). There was no effect of anti-IFNAR treatment on iv⁺ meningeal or blood neutrophils. For further details, see **Figure S23**.”

Results (new text in bold):

“In WT mice, IFNAR blockade successfully attenuated iv^- neutrophil accumulation in the meninges (**Figure 6F: $*P < 0.05$, $F_{(1,27)} = 5.0$**) with the expected increase in control IgG antibody-treated CSD mice ($*p < 0.05$, $q = 4.0$). **iv^+ neutrophils in the meninges and blood were not impacted by anti-IFNAR treatment, though iv^+ meningeal neutrophils were elevated by CSD ($*P < 0.05$, $F_{(1,27)} = 5.4$).** The effects of anti-IFNAR treatment appeared specific to iv^- meningeal neutrophils. Notably, depletion of iv^- meningeal B cells by CSD stress was unimproved ($***p < 0.001$, $F_{(1,27)} = 18.7$). There were otherwise no differences in other meningeal or blood immune cell populations (Figure S23A).”

5/29) What could be the differential role of meningeal neutrophils versus ‘stuck’ neurovascular neutrophils?

We agree this is an interesting question for future follow-up and have added text to the discussion:

“...Future efforts to characterize NVE-associated neutrophils and how they differ from non-adherent blood and meningeal neutrophils will be important...”

5/30) The data on CXCL1 and CXCL12 should be indicated as highly hypothetical as no functional/blocade experiments have been performed.

We acknowledge this point and have updated the legend for **Figure 7** to highlight the data we collected on neutrophil-related chemokines, and to emphasize that these may be involved in regulating peripheral neutrophil release (emphasis added):

“Figure 7:..Increased expression of the neutrophil chemoattractant CXCL1 in the liver, increased CXCL1 protein in plasma, and reduced BM expression of CXCL12—which promotes retention of neutrophils—may contribute to the elevated circulating pool of neutrophils in blood (Figure S17), though there were no detectable changes related to this pathway in the meninges.”

5/31) Data on blood cells proportion upon IFNAR treatment should be indicated in the main text, as it’s a nice way to decorrelate blood counts and restoration of cognitive functions.

This is now added to **Figure 6F** as per point **5/26** above.

5/32) Is IFN alone enough to shut down semaphoring expression?

This is a very interesting question; further exploration of the role of semaphorins in the stress response, and the regulators of semaphorins, lies outside the scope of our study, but we have added the following text to the discussion:

“Future efforts to determine what stress-related factors influence Sema3 expression will be important.”

5/33) The authors should only mention ‘potential’ migration from the bone marrow as now experiments is shown to prove it (parabiosis, skull graft, etc.)

Parabiosis and skull grafting studies are important mechanistic tools but come with their own caveats and issues related to host-graft rejection, BBB weakening by irradiation needed for transplantation studies, immune activation, general translatability concerns, etc.. We feel that with the addition of extensive new bioinformatic analyses, highlighted in response to **5/6** and **5/24**, above, the results and language stand as is. We have also added text into the limitations section highlighting the need for future work in this area, as shown in **5/1**.

Reviewer #1

I find the revisions acceptable.

Reviewer #2

The authors addressed all my criticisms and suggestions in a satisfactory manner. I can now recommend publication and congratulate them for this important work.

Reviewer #3

Reviewer #5

The authors have addressed my concerns and have provided an exhaustive document, with associated experiments. The manuscript is suited for publication.

We thank these 4 reviewers for their further review and their positive comments on our rebuttal, the importance of this work, and its readiness for publication.

Reviewer #4:

In the revised manuscript, the authors have addressed some of the comments from the prior submission. However some comments have not been addressed to the levels that it answers my concern.

4/1) The specific effect of IFNARI on neutrophils migration to improve social defeat. The authors justified in their rebuttal that the anti-IFNARI antibody they use does not cross the blood brain barrier and that they can't measure neutrophils migration through the skull after IFNARI blockade. Regarding the first part of their answer, the manuscript they cite to claim that the antibody does not cross the BBB state "MAR1-5A3 is not expected to cross the blood-brain-barrier". Without further evidence of such, it's a rather thin argument to

exclude potential other effect of the IFNARI blockade, particularly on other cells like microglia and macrophages (shown previously to affect behavior).

It is unclear with LysM GFP+ mice would not be able to be treated with anti-IFNARI to assess the skull channel quantification as presented in figure 3?

This comment raises two interesting but distinct questions: (i) how strong is the evidence for neutrophil-specificity of IFNARI effects in these data? (ii) why can we not treat LysM GFP+ mice with anti-IFNARI to quantify skull channel trafficking? We provide a detailed response to each of these points in turn:

(i) We agree that our data highlight promising directions for future studies to definitively establish the role of neutrophil-specific IFNAR signaling and trafficking. We have added additional text and moved this topic into the discussion section from the limitations section to underscore this point as follows (new text in bold):

“Given that the MAR1-5A3 IFNAR antibody clone we used is **not thought to cross** the blood brain barrier (BBB)⁵⁴ **and improves BBB integrity**⁵⁵, we **assumed** that the mediating effects of anti-IFNAR treatment on the negative sequelae of CSD stress did not involve brain parenchymal cells. **However, other studies have highlighted a microglial IFN-I response during stress**^{56,57}. **In agreement with this, we found upregulation of the IFN-I response genes *Ifitm2* and *Ifitm3* in microglia from CSD mice**²². **Future experiments using neutrophil-specific IFNAR knockout will be important to establish whether the beneficial effects of IFNAR depletion are restricted to peripheral neutrophils, or if microglia and other cell types are involved.**”

(ii) We agree that the suggested tissue-clearing studies in LysM mice would be of interest, and are technically feasible, and we have emphasized this point in our discussion of future directions (new text in bold):

“What processes drive neutrophil infiltration toward the meninges following CSD stress? **While the cellular source of stress-related IFN-I is currently unclear, one possibility is that microglial production of IFN-I is responsible for neutrophil chemotaxis into the meninges. Using bulk transcriptomics, we have previously demonstrated enrichment of pathways related to IFN-I production in CSD microglia**²². **Microglia-specific depletion of IFN-I production, coupled with tissue clearing or *in vivo* imaging studies of fluorescently labeled neutrophils in mice given IFNAR antibody therapy, will be important to definitively corroborate this mechanism.**”

However, these new experiments would require a minimum of 8 months to complete, accounting for time to reacquire LysM mice, establish breeding pairs, raise animals to adulthood, and conduct the experiments. We consider it beyond the scope (and resources) of the current study to undertake this significant follow-on study immediately, although as evidenced, we have flagged it as a possible future direction for the field. In agreement with other reviewers, we believe the current manuscript already reports large datasets with sufficient novelty and rigor to merit publication in its present form, without inclusion of major new datasets at this stage.

4/2) The role of neutrophils in social defeat. Multiple reviewer highlighted that demonstrating the central role of neutrophils accumulation being a causal factor in social defeat would be important, even more so given the potential other effects of the anti-IFNARI treatment. The authors state that their attempt had detrimental effects. It is however unclear if the effects were limited to the stressed mice or the depletion itself. Boivin et al, (Nat Comm, 2020) demonstrated stable and sustainable depletion of neutrophils for up to 20 days. If the depletion of neutrophils is somehow detrimental in the CSD model, it would still be worth including in the manuscript for the community to be aware of.

We agree that the detrimental effects of neutrophil depletion on mouse welfare in this model constitute important data to share with the scientific community. To address this, we have added a new supplemental figure, **Figure S26** (below), to highlight the adverse consequences of neutrophil depletion in the chronic social defeat (CSD) model with anti-Ly6G treatment. Accompanying figure legend, supplemental table, and changes to the text have been made in a revised version of the manuscript and are provided here.

Updated supplemental methods:

“Systemic neutrophil depletion. To further explore the mechanistic role of neutrophils in the response to chronic social CSD stress, we first performed neutrophil depletion experiments using methods optimized for prolonged treatment (Faget et al., 2018). Briefly, 8 mice were randomly assigned to either the HC or CSD group. 2 mice from each group were then randomly assigned again to either treatment (Ly6G antibody depletion) or control (IgG antibody) conditions, comprising 4 groups in total: HC+IgG, HC+Ly6G, CSD+IgG, CSD+Ly6G (n=2 per group). Primary (1^o) rat anti-mouse Ly6G antibody (clone: 1A8, Cat # BE0075, Bio X Cell) and 50 µg secondary (2^o) anti-rat antibody (clone: MAR18.5, Cat # BE0122) were administered on alternating days for the treatment group, and an equivalent amount of IgG antibody (clone: 2A3, Cat # BE0089) was administered to control condition

mice, according to the timeline shown in **Figure S26A**. In this pilot we noted (but did not quantify) ear tag holes in both HC and CSD groups that became enlarged over time, indicating possible wound healing deficits. We therefore developed a rubric for assessing fight-related wounds in CSD animals (**Table S5**). As anticipated, neutrophil depletion significantly impaired wound healing of the typically minor injuries sustained during the CSD protocol (**Figure S26B**). We also noted massive splenomegaly (>300mg) in two Ly6G-treated mice (one HC, one CSD). Due to concerns about animal welfare, we ceased using the Faget depletion model for CSD experiments.

In an attempt to mitigate these effects, we next pursued a modified neutrophil depletion protocol with a shortened timeline of defeat exposure (**Figure S26C**) and reduced dosage of antibody compared to that typically used (Daley et al., 2008; Zhang et al., 2009; Wojtasiak et al., 2010; Jaeger et al., 2012; Wozniak et al., 2012)—i.e., 100 µg compared to 200 µg. 14 mice were randomly assigned to either CSD+saline (n = 6) or CSD+Ly6G conditions (n = 8). After 10 days of defeat, wound scores were similar between groups, but massive splenomegaly was again observed in the neutrophil-depleted animals (**Figure S26D**). Discussion of these data with animal care staff led us to cease pursuit of these studies due to a combination of animal welfare concerns and uncertainty about experimental confounding, as any immunological or behavioral differences observed could have been driven by gross immune abnormalities related to acute splenomegaly rather than effects on meningeal neutrophils in neutrophil-depleted CSD mice.”

Figure S26: Systemic neutrophil depletion is poorly compatible with CSD stress. **a)** Neutrophil depletion paradigm illustrating schedule of anti-Ly6G and secondary anti-rat injections, based on (Faget et al., 2018). The 4 experimental groups were HC+IgG (control), HC+Ly6G, CSD+IgG (control) and CSD+Ly6G ($n = 2$ per group). **b)** Wound scores and spleen weights for mice used in **(a)**. Massive splenomegaly present in some animals, combined with informal observations that ear punches to track individuals did not heal properly—and on the contrary, expanded—and the concerning appearance of wound injuries in CSD mice led to early termination of the study. **c)** Modified neutrophil depletion paradigm illustrating reduced number of defeat encounters. All mice were exposed to defeat; saline injected mice served as a control ($n_{\text{saline}} = 6$, $n_{\text{Ly6G}} = 8$). **d)** Wound scores and spleen weights for mice used in **(c)**. There was no difference in wound severity with the modified paradigm, however splenomegaly was significantly increased in Ly6G-treated mice compared to saline (** $p < 0.01$, $t = -4.4$, $df = 7.3$). **e)** Wound scores and spleen weights for both Ly6G studies—Ly6G+2' **(a)** or Ly6G alone **(c)**—are

shown in comparison to mice in other studies shown throughout the manuscript.

New Table S5:

Wound score	Major criteria
1	No abrasions present
2-3	Abrasions
4	Abrasion + previous damage
5-6	Epidermal tears
7-8	Epidermal tears + previous damage
9-10	Laceration, deep tissue damage

Table S5: Criteria for assessing fight-related wounding.

In the limitations section, we have modified the text to reference the new material (new text in italics):

“...systemic knockdown to test the impact of neutrophil depletion over the course of the stress paradigm was associated with animal welfare concerns (see ***Supplemental Methods***)...”